



# Satellite retrieved sea ice concentration uncertainty and its effect on modelling wave evolution in marginal ice zones

Takehiko Nose[1], Takuji Waseda[1], Tsubasa Kodaira[1], and Jun Inoue[2]

[1]Graduate School of Frontier Sciences, The University of Tokyo, Kashiwa, Japan
[2]National Institute of Polar Research, Tachikawa, Japan

**Correspondence:** Takehiko Nose (tak.nose@k.u-tokyo.ac.jp)

**Abstract.** Satellite retrieved Sea Ice Concentration (SIC) uncertainty is studied with respect to its effect on spectral wave modelling of ice-covered water. Eight SIC products based on four algorithms applied to SSMIS and AMSR2 data were analysed. They were compared with sea-truth images captured from a 12 day fixed Marginal Ice Zone (MIZ) transect observation during the November 2018 R/V Mirai expedition in the Chukchi Sea. The analysis shows the refreezing sea ice field is highly variable

in time and space, and the uncertainty of SIC estimates is considerable. A wave hindcast experiment for the observation period using these SIC products as model forcing has shown that the SIC uncertainty translates to wave prediction discrepancies in ice cover. There is evidence that bivariate uncertainty data (model significant wave heights and SIC forcing) are correlated, although off-ice wave growth is more complicated due to the cumulative effect of SIC uncertainty and the model implementation of wave-ice interactions along an MIZ fetch. Analysis of significant wave height uncertainty distributions for SIC forcing

and wave-ice interaction source terms reveals that they are both sizeable; however, the study concludes the more dominant uncertainty source of modelling wave-ice interactions is the accuracy of satellite retrieved SIC estimates that are used as model forcing.

## 1 Introduction

Satellite remote sensing and in situ observations reveal the Arctic Ocean sea ice has been declining in extent and volume (Maslanik et al., 2007; Kwok and Rothrock, 2009; Stroeve et al., 2012). Stroeve and Notz (2018) have highlighted the emergence of consecutive monthly negative sea ice extent anomalies in recent years. From a practical view point, this downward trend of sea ice decline opens trans-Arctic shipping routes connecting Europe and Asia for longer times of the year; potential global economic benefits of non-ice breakers accessing routes like Northern Sea Route and North West Passage are substan-

tial (Stephenson et al., 2013; Bekkers et al., 2018). However, ships in polar waters can encounter hazards such as collision with perennial sea ice, sea-spray icing, and encountering high winds and waves. These hazards can be alleviated with reliable forecasts, but understanding model uncertainties, which are greater in the Arctic regions due to limited observations, is also





crucial. As such, the knowledge of limitations relating to modelling hazardous physical processes is a requisite for sustainable developments of the Arctic Ocean. This paper focuses on the ocean surface wind waves, in particular the effect of Sea Ice

Concentration (SIC) uncertainty on third-generation spectral wave model simulations in and near a Marginal Ice Zone (MIZ).

Documented academic work on wave-ice interactions has a long history dating back as far as Greenhill (1886) (V.A. Squire, 2007; Mosig et al., 2015). When wind waves propagate through/under sea ice cover, dispersion relation is modified and wave energy is attenuated due to non-conservative dissipation and a conservative scattering phenomenon. Standalone contemporary spectral wave models simulate wave-ice interactions using sea ice as forcing; in this space, the intensive field measurements

of the Arctic Sea State and Boundary Layer Physics Program (Thomson et al., 2018) have made a solid contribution to the recent advance of The WAVEWATCH III ® Development Group (WW3DG) (2019) wave-ice interaction source terms. Rogers et al. (2016); Cheng et al. (2017); Ardhuin et al. (2018); Boutin et al. (2018) describe the development, implementation, and optimisation of the latest WW3 parameterisations for wave evolution in sea ice cover. Despite the progress, Squire (2018); Thomson et al. (2018) qualify accurately quantifying the wave decay and connecting the associated mechanisms over a large

domain still remain a challenge because sea ice fields are notoriously heterogeneous; therefore, the wave-ice interaction source term is a source of uncertainty when simulating wave evolution in MIZs.

Besides the model interior, the $0^{th}$ order uncertainty pertains to sea ice forcing accuracy like SIC and sea ice thickness. In particular, SIC retrieved from satellite radiometers (or sea ice models that assimilate satellite observations) forms the most fundamental input into wave models simulating wave-ice physics and should have a profound effect on sea state predictions.

Spatial distributions of SIC in the Arctic Ocean can be mapped daily based on satellite microwave radiometry, and they are the primary source of sea ice trend and climate studies; however, discrepancies among retrieval algorithms have been a long-known issue, and numerous intercomparison studies have investigated the effects of retrieval algorithms, and to a lesser extent instruments, on SIC estimates (Comiso et al., 1997; Meier, 2005; Andersen et al., 2007; Notz, 2014; Ivanova et al., 2015; Comiso et al., 2017; Chevallier et al., 2017; Roach et al., 2018; Lavergne et al., 2019). To date, there is no robust validation

of any algorithm, so users are urged to understand strengths and weaknesses of the algorithms when using and interpreting the data (Ivanova et al., 2015; Comiso et al., 2017). The long-known SIC discrepancies imply there is uncertainty in the knowledge of true sea ice coverage (Notz, 2014). Because the satellite retrieved SIC has uncertainty, the choice of a SIC product a modeller and model developers select is an error source. Interestingly, the latest WW3 wave-ice parameterisation developments have all used different sea ice forcing products. Therefore, understanding the effect of SIC uncertainty on wave predictions is a relevant

contribution and is the primary objective of this paper.

The study motivation is introduced to close the preliminary section; it was inspired by an Arctic Ocean observational campaign on board R/V Mirai in the refreezing Chukchi Sea during November 2018 (JAMSTEC, 2018). A 12 day MIZ transect observation was conducted to capture daily changes in sea ice conditions at the same geographical locations, which revealed firsthand that the MIZ sea ice field is very dynamic in time and space; we began to inquire the accuracy of satellite retrieved

SIC compared to the sea-truth observation, which led to the subject of satellite retrieved SIC uncertainty. The ensuing Section 2 introduces the methods employed to analyse the wave model uncertainties associated with SIC forcing. Section 3 presents a comparison of eight satellite retrieved SIC products at the in situ measurement locations and the Chukchi Sea during the ob-





servation period. A wave hindcast experiment is conducted using these SIC products as sea ice forcing for which the model
results are compared with limited available in situ wave observations and two independent predictions as described in Sec-
tion 4. Analysis of bivariate uncertainty data (model significant wave heights and SIC forcing) from a physical view point of
modelling wave decay and growth are provided in the discussion Section 5, and Section 6 concludes the study.

## 2 Methods

### 2.1 R/V Mirai sea-truth and buoy observations

**R/V Mirai sea-truth observation**

Regions in the Arctic Ocean, like the Chukchi Sea, that were inaccessible in November are now open for navigation, even
for non-icebreakers, and R/V Mirai, a Japanese Polar Class 6 vessel (JAMSTEC, 2019), carried out a late autumn voyage in
2018. R/V Mirai arrived in the Chukchi Sea on 4 November; after other ship time commitments, it began a 12 day transect
observation that included an MIZ during daylight hours on 9 November. Daylight hours are limited at high latitudes during this
time of the year, so sea-truth observation was conducted generally 19:00–00:00 UTC each day. The transect spanned roughly
from 73.00° N, 198.00° E in the MIZ to 72.00° N, 194.00° E toward the central Chukchi Sea. Albeit the MIZ coverage being
less expansive, firsthand observation with a series of sea-truth photographic data of daily sea ice conditions taken at the same
geographical locations for an extend period requires exhaustive ship time and therefore is rare if not unique. The R/V Mirai
transect on 15 November is overlaid on the mosaic of Sentinel-1 A and B Synthetic Aperture Radar (SAR) Normalised Radar
Cross Section (NRCS) images captured on the same day in Figure 1. The NRCS data were obtained from the National Oceanic
and Atmospheric Administration (NOAA) Coastalwatch FTP site (NOAA, 2019). Shipboard measurements used in this study
include surface wind, Sea Surface Temperature (SST), air temperature, and wind waves (WM-2 and Piper-C#15). The details
of the measurement systems are provided in Appendix A.

**Drifting buoy wave measurement**

Two drifting type wave buoys were also deployed as part of the R/V Mirai observational campaign. One failed within hours, but
the other buoy (Piper#13) survived for 19 days after being deployed on 6 November 2018 22:18; it was remotely switched to a
sleep mode to preserve battery on 26 November, and the remote connection ceased on 5 December for some unknown reason.
Hardware and on-board data processing were mostly same as Nose et al. (2018) except Piper#13 produced bulk parameters
at 15 minute intervals, which were transmitted near real time via Iridium satellite communication. Piper#13 was deployed at
73.32° N, 201.09° E, and its track between 6 and 27 November is presented in Figure 1 overlaid on the NRCS mosaic. The wave
height is calculated as $H_{m0} = 4\sqrt{m_0}$ within the analysed range of a spectrum between the low and high cut-off frequencies,
$f\_low$ and $f\_high$, respectively. $m_0 = \int_{f\_low}^{f\_high} S(f)df$ where $S$ is the variance density spectrum.



## 2.2  Satellite retrieved sea ice concentration

SIC estimates from Earth-orbiting satellites are an indirect measurement translated from microwave brightness temperatures. Although passive microwave radiation has low energy, brightness temperatures between sea ice and open ocean are distinguishable due to difference in surface emissivity and physical temperatures. Microwave brightness temperatures measured from different frequency channels can account for the spatial and temporal variations of the ocean surface, so transfer functions, i.e., retrieval algorithms, can be applied to produce SIC field estimates (Comiso et al., 2017).

Since the 1970's, a number of multichannel passive microwave radiometers have been in operation, and the sensors currently in operation (that are most used) for sea ice analysis are SSMIS and AMSR2. The key difference between the two sensors to derive the SIC spatial distribution is footprint resolution as the latter instrument has around 3–4 times higher resolutions for frequencies near 19, 37, and 89 GHz. For these two sensors, an exhaustive number of SIC retrieval algorithms have been developed primarily because different algorithms produce considerably different SIC estimates. This is evidenced by a long list of intercomparison studies (Comiso et al., 1997; Meier, 2005; Andersen et al., 2007; Notz, 2014; Ivanova et al., 2015; Comiso et al., 2017; Chevallier et al., 2017; Roach et al., 2018; Lavergne et al., 2019).

A total of eight SIC products were selected for this uncertainty study based on four algorithms applied to SSMIS and AMSR2. The selected products are summarised in Table 1 where the product abbreviations and data references are provided. Four algorithms that appear most frequent in literature were considered as the leading algorithms. The selected algorithms are as follows: NASA-Team (Cavalieri et al., 1984), Bootstrap (Comiso, 1986), OSISAF, and ARTIST-sea-ice (Spreen et al., 2008).

A concise explanation for the selection of each product in Table 1 is provided below:

- NASA-Team—the algorithm has been used for sea ice trend and climate studies since the beginning of the satellite radiometry era. Data used here are the original NASA-Team algorithm applied to SSMIS and the enhanced NASA-Team2 algorithm applied to AMSR2.

- Bootstrap—like NASA-Team, the Bootstrap algorithm has been used for sea ice trend and climate studies for many years. Data used here are the Bootstrap algorithm applied to SSMIS and AMSR2.

- OSISAF—this algorithm was selected because the leading numerical weather prediction centre, European Centre for Medium-Range Weather Forecasts (ECMWF), uses SST and sea ice information based on the Operational Sea Surface Temperature and Sea Ice Analysis (OSTIA) system (Donlon et al., 2012). The ECMWF wave model (ECWAM)'s sea ice forcing uses the OSTIA system for which SIC is retrieved using the OSISAF algorithm applied to SSMIS. We also analyse data applied to AMSR2 in this study.

- ARTIST-Sea-Ice—this algorithm uses 89 GHz frequency signal to produce high resolution SIC estimates. This algorithm was selected as accurate higher resolution forcing is generally desirable for numerical models. For this product, we only use the data applied to AMSR2 but analyse two different grids. The ASI products are offered with various regional grids at the horizontal resolution of 3.125 km (The University of Bremen, 2019), so both the 6.250 km Arctic and 3.125 km





Chukchi-Beaufort grids are studied. There is a 3.125 km Arctic grid as well, but the Chukchi-Beaufort grid SIC appears subtly refined whereas the both Arctic grids are similar. As such, the Chukchi-Beaufort grid was selected.

The principle of all sea ice algorithms as described in Comiso (2007) is that measured radiative flux can be expressed as $T = T_i c_i + T_o c_o$ where $T_i$ and $T_o$ are the brightness temperatures normally observed from 100 % ice cover and 100 % open water, respectively. Then, SIC, $c_i$, can be expressed simply as $c_i = \frac{T_b - T_o}{T_i - T_o}$ where the subscript $b$ corresponds to observed ocean

surface, and the ice-free surface is $c_o = 1 - c_i$ (Comiso, 2007). The accuracy of SIC is then dependent on the closeness of tuning tie points to the ice-free and sea ice ocean surface brightness temperatures. The selection of frequency channels to derive polarisation ratios or differences (V and H) and gradient ratios (V polarisation) to retrieve SIC also dictates strengths and uncertainties of each algorithm. Technical details of the respective algorithms are described in the Table 1 data references.

### 2.3 Third-generation spectral wave model

The effect of SIC uncertainty on wave predictions was investigated by a hindcast experiment using The Arctic Ocean wave model developed at the University of Tokyo (TodaiWW3-ArCS) based on WW3, which was introduced in Nose et al. (2018). Third-generation spectral wave models simulate the numerical evolution of waves as energy budgets based on the action density balance equation,

$$\frac{\partial N}{\partial t} + \nabla \cdot \boldsymbol{c} N = \frac{s}{\sigma}. \tag{1}$$

The left hand side concerns wave kinematics where $N$ is the wave action density spectrum, which is a function of frequency ($\sigma$), direction ($\theta$), space ($x$ and $y$), and time ($t$), and $\boldsymbol{c}$ describes the propagation velocities in spatial and spectral coordinates. In deep water when neglecting currents, $\boldsymbol{c}$ is the group velocity, $\boldsymbol{c}_g$. The primary source terms are on the right hand side and consist of the following: the wind input term, $s_{\text{wind}}$, the wave dissipation term, $s_{\text{dissipation}}$, the non-linear interaction term, $s_{\text{non-linear interactions}}$, and the wave-ice interaction term, $s_{\text{ice}}$. The sum of these source terms, $s$, is expressed based on the following default scaling in

ice-covered waters:

$$s = (1 - c_i)(s_{\text{wind}} + s_{\text{dissipation}}) + c_i s_{\text{ice}} + s_{\text{non-linear interactions}}. \tag{2}$$

A curvilinear grid implemented by Rogers and Campbell (2009) is suitable at high latitudes and adopted here with a polar stereographic projection of 75° N latitude (produced by Mathworks Matlab's polarstereo_inv function). The TodaiWW3-ArCS grid used in this study has an increased horizontal resolution of 4 km, and its domain was modified to only cover most of the

Pacific side of the Arctic Ocean (which covers the following region of the polar stereographic grids: the easting extent between −1,800 km and 1,512 km, and the northing extent between 520 km and 2,904 km). The model domain was enclosed by ice cover from other seas during the November 2018 modelling period, so nesting was unnecessary. Similar to Rogers et al. (2016), we neglected swell penetration through the Bering Strait. The spectral grid has 36 directional and 35 frequency bins with the latter ranging from 0.041 Hz to 1.052 Hz. The grid was defined using the International Bathymetry Chart of the Arctic Ocean

bathymetry (Jakobsson et al., 2012) and the Global Self-consistent, Hierarchical, High-resolution Geography shoreline data (Wessel and Smith, 1996), and there are approximately 301,535 sea point cells.





During the version upgrade of TodaiWW3-ArCS, $s_{\text{wind}}$ and $s_{\text{dissipation}}$ parameterisations and wind forcing were sanity checked against the 2016 September storm when the model and observations agreed well (Nose et al., 2018). We compared the leading packages, ST4 (Ardhuin et al., 2010; Rascle and Ardhuin, 2013) and ST6 (Rogers et al., 2012; Zieger et al., 2015; Liu et al.,

2019), using ECMWF global reanalysis (ERA5) 10 m wind ($U_{10}$). The latter parameterisation showed marginally improved agreement using default parameters; so all simulations used the ST6 parameterisation and were forced with ERA5 wind fields.

The $s_{ice}$ term is composed of the following attenuation terms: dissipation due to basal friction, flexural dissipation, and visco-elastic dissipation, and these terms can be combined with an energy-conservative scattering attenuation term. Physics-based wave-ice parameterisations in WW3 include basal friction (IC2) (Ardhuin et al., 2018; Boutin et al., 2018), visco-elastic

dissipation (IC3) (Wang and Shen, 2010; Rogers et al., 2016; Cheng et al., 2017), IC5, which is an another type of a visco-elastic beam model (Mosig et al., 2015), and scattering including flexural dissipation (IS2) (Kohout and Meylan, 2008; Williams et al., 2013; Ardhuin et al., 2018; Boutin et al., 2018). The dominant floe size scale within a distribution should dictate the range of wavenumbers for which scattering is important. In field observations, a simple power law of floe size distributions appears to describe how wave attenuation varies with wave frequency (Squire, 2018). Numerically, attenuation using the IS2 switch based

on Kohout and Meylan (2008); Williams et al. (2013); Ardhuin et al. (2018); Boutin et al. (2018) has been implemented in WW3, which adopts an assumed power law derived from the estimated maximum floe size due to ice break up. In refreezing ocean near the ice edge where the R/V Mirai transect and Piper#13 observations were conducted, the MIZ mostly comprised of pancake and ice cake sizes well below the local wavelengths; scattering attenuation is less dominant for these waters (Squire, 2018). As such, scattering was not considered here, and the wave hindcast experiment used the IS0 switch. The underlying

principle of sea ice models is that sea ice is treated a continuum. In the field, sea ice floes are not a continuous sheet, so the selection of an appropriate wave-ice interaction parameterisation does remain an open question. We do know, however, these source terms are sensitive to the data for which they are tuned, so the experiment adopted the IC3 package with the default parameters as it has been developed for grease, frazil, and pancake ice (Rogers et al., 2016; Cheng et al., 2017) using the refreezing Beaufort Sea data of Thomson et al. (2018). Regarding sea thickness forcing, a homogeneous input option with a

value of 10 cm was applied to avoid ambiguity between different SIC products; the treatment of independent SIC and sea ice thickness data sets is not a trivial matter (Chevallier et al., 2017). 10 cm was chosen because the MIZ transect observation was mostly characterised by new and young ice like grease, nilas, and pancakes, of which the upper bound of sea ice thickness is of a similar order (Canadian Ice Service-Environment Canada, 2005).

For the hindcast experiment, ASI-3km and OSISAF-AMSR2 data were excluded: the former high resolution regional data

are invaluable for the SIC uncertainty analysis, but its domain is too small for the hindcast experiment. The OSISAF-AMSR2 data have noise in the open ocean, which yield erroneous wave simulation results when they are used as model forcing (as described in Appendix B). The remaining six satellite retrieved SIC products in Table 1 are used as sea ice forcing to examine wave modelling uncertainty, so the wave hindcast experiment dataset has $\{H_{m0\,sic1}, ..., H_{m0\,sic6}\}$ where the subscripts denote the SIC forcing. By doing so, we recognise sea ice information becomes independent of the ERA5 atmospheric conditions

(which produced the wind forcing), which implies moment flux is distorted. SIC uncertainty would cause an atmosphere-ocean discrepancy and likely affects the lower atmospheric conditions. For example, Inoue et al. (2011) evaluated surface heat transfer





from three reanalysis products by focusing on how the models treat sea ice and found the accuracy of SIC is a key variable for estimating surface turbulent heat fluxes. The authors are not aware of studies that have quantified the SIC uncertainty effect on the accuracy of surface wind in atmospheric models. Since WW3 is a standalone wave model that has consistent numerical

approaches for both ice-free and ice-covered waters, numerical stability is unaffected, so the wave hindcast experiment used consistent wind forcing with various SIC forcing. Step-like changes of SIC from daily intervals are not ideal as forcing, so the SIC data were interpolated to match the model output frequency of hourly intervals. Unless specified otherwise, all other settings were default. The modelling period covers both R/V Mirai and Piper#13 observations and is from 5 to 25 November 2018 with a 5 day spin up.

## 195 3 Sea ice concentration comparison

### 3.1 R/V Mirai sea-truth observation

Coinciding with the ship schedule, the transect waters began to refreeze and became consolidated ice cover at the start and (several days after) end of the observation period, respectively. The sea ice observation is grouped in four phases as distinct ocean surface features were captured from four meteorological conditions the ship encountered. The shipboard wind and SST

time series is provided in Figure A3 of Appendix C. During the first few days between 9 and 13 November 2018 (Phase 1), gradual sea ice growth was observed both in extent and ice cake/floe sizes under generally calm surface conditions. On 14 November, the most significant on-ice wind event during the transect period occurred. The peak wind speed measured by R/V Mirai was 18 ms$^{-1}$, and $> 10$ ms$^{-1}$ winds persisted for roughly 18 hours at the ship location. WM-2 and Piper-C#15 $H_{m0}$ both peaked $> 2.00$ m in ice cover indicating energetic sea state of this event. The MIZ was mostly broken up ice fields on the

following day, which was followed by the most apparent sea ice advance on 16 November as a seemingly dense ice field was encountered. We've grouped this period, Phase 2, as the on-ice event and aftermath. During 17 and 18 November, sea-truth observation was mostly open water; our conjecture is the sea ice disappeared by horizontal advection, but there is insufficient evidence to simply discard rapid melting or other processes. This analysis is still ongoing work. This period of minimal ice sighting is referred as Phase 3. In the final Phase 4, SSTs along the transect waters began to warm despite the persistent and

strengthening cold off-ice winds. Air temperatures along the MIZ transect on 18—20 November were $< -10$ °C, but the shipboard SSTs elevated $> 0$ °C for the entire MIZ transect waters the ship traversed on 20 November.

   Images of the ocean surface were captured by GoPro and hand-held cameras when R/V Mirai was in the MIZs. Sea-truth images are invaluable because they connect satellite retrieved SIC estimates and the ocean surface conditions. For our analysis, SIC was linearly interpolated in time to hourly intervals and bilinearly interpolated in space from respective native grids to the

R/V Mirai positions using the Python Scipy interpolation package (Jones et al., 2001–). The connection between the satellite estimates and sea-truth images during 14 and 16 November 2018 are respectively depicted in Figures 2 and 3. Following the gradual sea growth in Phase 1, R/V Mirai encountered a dense ice cover comprising of pancake ice and slurry as it can be seen in the images captured at 20:08 and 21:05 shown in Figure 2. Despite the foreseeable ocean being more or less completely covered in ice, satellite SIC estimates are $< 0.40$. Farther into the MIZ near the peak of the SIC estimates, R/V Mirai was located in





an expanse of open ocean. As the ship began its return leg along the transect, similar dense ice cover was encountered where the sea state was energetic. A similar remark about the spatial variability of sea ice field can be made for the 16 November observation. The sea-truth observation reveals that MIZ sea ice fields during the refreezing period appear to form in strips and patches of new and young ice within and in between the expanse of open water. The sea ice tends to cluster, so it appears dense where the ice exists. In other words, the spatial distribution is highly variable and not uniform; therefore, SIC would vary
depending on the footprint for which the quantity is being estimated.

Uncertainty of a physical quantity, $X$, from herein is defined as follows:

$$uncertainty(X) = max(X_{sic1}, ..., X_{sic8}) - min(X_{sic1}, ..., X_{sic8}), \tag{3}$$

where the subscripts are the respective satellite retrieved SIC products (six SIC forcing for the wave hindcast data). So SIC and $H_{m0}$ uncertainties are denoted respectively as follows: $\Delta$SIC $= uncertainty($SIC$)$ and $\Delta H_{m0} = uncertainty(H_{m0})$.

Figures 2 and 3 depict the SIC interpolated at the R/V Mirai positions has considerable uncertainty. The entire time series of SIC comparison for all phases during the transect observation are provided in Figures A4 to A6 of Appendix C. A noticeable feature of the satellite estimates is the $\Delta$SIC is effectively equal to the maximum SIC except for Phase 2. It implies for all other phases, at least one data product has near-zero SIC while other estimates indicate higher values.

## 3.2 SIC uncertainty in the refreezing Chukchi Sea

A comparison of gridded SIC data to Lagrangian observation requires interpolation. A direct-qualitative comparison of SIC estimates in the Chukchi Sea is discussed to further examine the uncertainty. Figure 4 compares the SIC fields on 15 November 2018 from three products: OSISAF-SSMIS, BST-AMSR2, and ASI-3km. The difference among these estimates in the MIZs is striking regardless of their footprint resolutions; the sea ice fields around Wrangel Island (71.00° N, 280.00° E) and the waters between north east of it to the central north Chukchi Sea have completely different characteristics depending on the data. A
distinctly noticeable trait of OSISAF-SSMIS is that SIC is generally characterised by a monotonic function between the open ocean and pack ice. Accordingly, the spatial variability is not captured; coarser footprint resolution is likely a cause, however, it is speculated that other factors such as aerial filtering may also be affecting the OSISAF-SSMIS SIC estimates.

To quantitatively analyse the spatial variability shown in Figure 4, 0.15 and 0.85 SIC contours were overlaid on the NRCS mosaic of Sentinel-1 images acquired on 15 November 2018 (Figure 5). The NRCSs are an useful medium to overlay the
SIC contours because they provide indication of true sea ice fields. The mosaic depicts sea ice edges have a wavy, but highly nonlinear, jagged form. For most of the ice edges, Figure 5 shows OSISAF-SSMIS derived 0.15 contours are smoother whereas the BST-AMSR2 and ASI-3km contours appear to follow the sea ice edge concavity and convexity with a varying degree of closeness; the latter appears to be qualitatively more consistent with the ice edges detected in the NRCS data. Figure 5 also shows the 0.85 SIC contours are somewhat qualitatively similar between OSISAF-SSMIS and ASI-3km; however, BST-
AMSR2 and OSISAF-SSMIS 0.85 contours can be some 200 km apart, for example between 73.00° N, 190.00° E and 74.00° N, 185.00° E. Regions of low radar intensity that appears dark in the NRCSs are apparent in the disparate 0.85 SIC contours, which indicate the waters in this area were not high SIC. As such, it does imply BST-AMSR2 overestimated the SIC for this date.



Although not shown here, 0.50 SIC contours are inconsistent among all three products. Qualitatively, spatial variability in the region of the greatest discrepancies appears to be best captured by ASI-3km for this particular date.

The analysis here suggests ASI-3km generally captures the SIC spatial variability shown in the NRCS data; however, there is insufficient evidence to determine its accuracy. A more straightforward conclusion is that there is considerable uncertainty among various satellite retrieved SIC products derived from different algorithms and sensors.

## 4    Wave model predictions at the observation sites

The wave hindcast experiment as described in Section 2.3 is conducted to examine how the SIC uncertainty translates to wave
predictions in MIZs. Because we lack a sufficient duration of robust in situ wave measurements, two independent numerical wave models that produce predictions in the Arctic Ocean, namely ERA5 ECWAM and Arctic Monitoring and Forecasting Centre (ARCMFC) wave model, both based on WAM, are also included in the analysis. Our goal is to understand SIC uncertainty effects on waves, but adequate model accuracy, at least qualitatively, is needed for a meaningful uncertainty analysis. These models provide a guide on the TodaiWW3-ArCS performance against the best practice wave models. The ERA5 ECWAM
data are made available on a 0.5° resolution grid on the Climate Data Store (Copernicus, 2019), and the model treats grid points with SIC $> 0.30$ as land using an ice mask. The ARCMFC wave model has a horizontal resolution of (a bit over) 8 km and simulates waves under sea ice cover based on Sutherland et al. (2019)'s two-layer sea ice model. The ARCMFC TOPAZ model provides the concentration and thickness forcing, which are kept constant from the initial state. Daily ARCMFC wave analysis data remapped to a 6.25 km polar stereographic grid are made available on ARCMFC (2019). The available data for
the analysis period of November 2018, however, have a large number of grids with no values, and the sea ice cut-off criterion is not clear in the documentation. Nevertheless, it is assumed wave attenuation is simulated where grid cell values exist.

**R/V Mirai transect observation**

Figure 6 presents time series of $H_{m0}$ interpolated at the R/V Mirai positions for all models. The figure also includes shipboard wave measurements from the WM-2 integrated analog system (TSK Tsurumi Seiki Co., 2019) when the ship speed was
$< 2$ ms$^{-1}$ (refer to Appendix A for further explanation), but the Piper#15 data on board R/V Mirai are not shown as measured waves were mostly wind seas, for which the data generally did not reflect the sea state. The time series figure depicts the effect of sea ice on waves each time R/V Mirai sailed in the ice cover as uncertainty generally increased in the MIZs. In the case of waves propagating toward the ice edge, waves decay with varying attenuation rates depending on the sea ice forcing used, and the representation of the ice edge affects the simulations of wave growth as they propagate toward open water. The
model $\Delta H_{m0} > 1.50$ m occurred on 20 November 2018 during the off-ice wind condition, which is $> 50$ % of the open water $H_{m0}$ (of the TodaiWW3-ArCS and ARCMFC models). Because R/V Mirai slowed down in the MIZs, at least one ice-cover measurement was obtained each day, and they generally lie within $[min(H_{m0\,sic1}, ..., H_{m0\,sic6}), max(H_{m0\,sic1}, ..., H_{m0\,sic6})]$. Furthermore, when the open ocean sea state is energetic, daily peak $\Delta H_{m0}$ occurs as R/V Mirai sailed into and out of the ice-covered water, indicating the representation of an ice edge in the model forcing plays an important role.





Regarding the comparison of three different base models, ERA5 ECWAM consistently has a positive bias compared with other models except for the on-ice wave event on 14 November when they all agree reasonably well. For this event, both WM-2 and Piper#15 have comparable estimates of the measured peak $H_{m0}$ as R/V Mirai was exiting from ice cover to the open ocean, which was a bit over 2.00 m. The ERA5 ECWAM positive bias compared with other models is exacerbated when the strongest off-ice winds were recorded by R/V Mirai between 19 and 22 November. ARCMFC $H_{m0}$ agrees slightly

better with $max(\text{TodaiWW3-ArCS}H_{m0})$. These are evidenced in bias and Root Mean Square Deviation (RMSD) values calculated with respect to the ARCMFC $H_{m0}$: the ERA5 ECWAM $H_{m0}$ has bias = 0.19 m and RMSD = 0.25 m, and the $max(\text{TodaiWW3-ArCS}H_{m0})$ has bias = −0.11 m and RMSD = 0.21 m.

    It is interesting to point out that using different sea ice forcing alone causes wave estimates to deviate in open water. This is also the case between the ERA5 ECWAM and ARCMFC models, which both use ECMWF wind forcing. The deviation of

wave estimates in open ocean is more apparent during the off-ice wind conditions at the end of the transect observation period. A conjecture is that a different treatment of sea ice in each model modifies available fetch for which waves can be generated; for example, having open water to 0.30 SIC in the ERA5 ECWAM model may simply increase the fetch distance at the R/V Mirai locations, which could result in a consistent positive bias compared with other models. Confirming this conjecture is ongoing work using 2019 R/V Mirai observations to better understand wave predictability in the open ocean adjacent to MIZs.

**Piper#13 drifting buoy observation**

    Figure 7 is a Piper#13 equivalent of Figure 6 with the respective observational data. Satellite retrieved SIC for all products at the Piper#13 positions is provided in Figure A7 of Appendix C. The buoy data are discussed first. It measured $H_{m0} > 1.00$ m for two days after being deployed at the finals hours on 6 November 2018. $H_{m0}$ tapered off to around 0.20 m, but briefly rose to 0.60 m when the wind speed increased at around 12 November 2018 00:00. Then, it drifted in ice cover with less wave

penetration; no measurable wave signals propagated to Piper#13 as derived spectra indicate instrument noise except during the on-ice wave event between the late hours on 14 November and the early hours of 15 November . Peak wave periods, $T_p$, which is a frequency inverse corresponding to the maximum variance density, were consistently around 9 s; this is likely a true wave signal even though $H_{m0}$ was only 0.05 m.

    Regarding the wave hindcast, TodaiWW3-ArCS $\Delta H_{m0}$ was the largest in open water and decreased with $mean(H_{m0})$.

In general, Piper#13 $H_{m0}$ during 7 and 8 November are underestimated by all numerical models. When the wave energy tapered off between 8 and 11 November, both ERA5 ECWAM and ARCMFC model $H_{m0}$ are overestimated. However, the episodic increase of wave energy on 11/12 November is reproduced in these models, albeit with a positive bias, whereas the TodaiWW3-ArCS simulations did not show any increase in the $H_{m0}$ at the Piper#13 location. There are no data available for the ERA5 ECWAM and ARCMFC models after 12 November at Piper#13. There are three occasions when all TodaiWW3-

ArCS simulations slightly overestimates $H_{m0}$ compared with the buoy data, which indicate the model attenuation rates may be too weak or SIC forcing is inaccurate.

    A comparison of modelled waves in the MIZs with the indicative in situ waves suggests there is no one forcing or a model that can reproduce the observations. Model tuning to the Piper#13 observation is likely plausible, but without the knowledge of




true SIC, tuned parameters have limited physical relevance. More importantly, the satellite retrieved SIC uncertainty evidently

translates to wave prediction uncertainty when it is used as model forcing to simulate wave evolution in ice-covered ocean.

## 5   Discussion

### 5.1   SIC uncertainty and how it affects modelling wave evolution in MIZs

The preceding Section 4 demonstrated simply changing SIC forcing alone produces considerable $\Delta H_{m0}$ in numerical wave simulations. A more in-depth analysis is conducted here to understand how SIC forcing differences affect the wave-ice inter-

actions as implemented in WW3 from a physical view point. At the most fundamental level, sea ice fields modify the fetch of the ocean: two cases are selected to analyse the effect of varying fetches on the attenuation and growth during on- and off-ice wave conditions on 15 November 2018 00:00 and 21 November 2018 18:00, respectively. The wind magnitudes and vectors for these cases are shown in Figure 8. For the on-ice wave case, relatively strong small-scale south westerly winds as depicted in Figure 8a generated waves with an $H_{m0}$ of about 2.00 m propagating towards the ice edge. When on-ice waves encounter

ice cover, rapid attenuation is expected within the order of tens of kilometres (Ardhuin et al., 2018; Squire, 2018), so the ice edge locations and the SIC variability near it affect the model simulation of wave decay. For the selected off-ice wave case, a low pressure system over Alaska and a high pressure system north west of the Chukchi Sea generated sustained north easterly winds over much of the Chukchi Sea as depicted in Figure 8b, which generated open water $H_{m0} > 3.00$ m. In this case, the ice edge and SIC field determine the fetch on which waves are generated, and as such, the SIC forcing introduces $H_{m0}$ uncertainty.

Owing to the heterogeneity of the wind and SIC fields and how they interact with each other, there is no statistical association for the bivariate uncertainty data even when the $H_{m0}$ is normalised to wind forcing. Examining the model uncertainties in the context of physical processes, a scatter plot is produced for both cases with the following visualisation technique: marker sizes are scaled to $mean(H_{m0})$ like a bubble plot and each marker is colour coded according to the $mean$(SIC forcing) like a colour-coded scatter plot. The former aims to emphasise the model data near the ice edge where the effects of wave attenuation

and growth associated with $\Delta$SIC is most prominent, and the colour coded markers indicate $mean$(SIC) among all forcing. For simplicity, we refer to these figures herein as an enhanced scatter plot.

Figure 9a depicts the spatial distribution of the model $\Delta H_{m0}$ for the on-ice wave case with 0.01, 0.50, and 0.85 $mean$(SIC forcing) contours. Not all MIZs are aligned to the on-ice wind orientation because of the ice edge shape. On-ice wave analysis is, therefore, conducted for a strip of the model grid points roughly 100 km in width along the south westerly on-ice wind

orientation as depicted in a grey dashed quadrilateral. An enhanced scatter plot shown in Figure 9b depicts bivariate uncertainty data likely corresponding to the model grid points along the wind forcing direction. There is a strong indication of a correlation between the $\Delta$SIC forcing and model $\Delta H_{m0}$ for this on-ice wave case. The correlated uncertainties imply that as on-ice waves approach and decay due to wave-ice interactions, larger discrepancies in the representation of SIC as forcing causes greater model $\Delta H_{m0}$.

Figure 9b also shows inverse proportion of marker sizes and uncertainties. Smaller size markers have low $mean(H_{m0})$, so larger $\Delta H_{m0}$ occur when only one or two of $\{H_{m0\,sic1}, ... H_{m0\,sic6}\}$ have $H_{m0} > 0$ while remaining $H_{m0} = 0$ due to waves





being fully attenuated. The figure only comprising blue and light-blue markers indicate the highly forced waves decayed with limited wave penetration no farther than $mean(\text{SIC}) = 0.40$. Furthermore, in theory, the cluster of data must approach the origin of the figure coordinate in the upwind open waters in the central Chukchi Sea. In other words, on-ice waves that are

being generated numerically in the open water must satisfy $\Delta H_{m0} = 0$ & $\Delta \text{SIC} = 0$. The reason they do not converge to the figure coordinate origin is that the SIC uncertainty along the Siberian coast (not shown here) affects the waves in the upwind waters as indicated by very faint yellow shades in Figure 9a.

Analysis of the off-ice case is carried out in a similar manner. The data bound by Quadrilateral 2 as shown in Figure 10a reflect the model grid points with the north easterly off-ice wind conditions. Although the correlation is not as high as the on-ice

wave case, the enhanced scatter plot in Figure 10c shows that the bivariate uncertainty data are correlated for the off-ice wave case as well. Scatter is apparent for $\Delta \text{SIC} > 0.10$, and although the enhanced scatter plot disparages data with low $mean(\text{SIC})$, high $\Delta H_{m0}$ with low $mean(\text{SIC})$ do exist. Analogous to the on-ice wave case, this can occur near the ice edge when only one or two simulations have the SIC forcing representing open water conditions, while the wave growth is suppressed for the remaining simulations due to the sea ice forcing. This effect is depicted in Figure A8c of Appendix D for a transect of $\Delta H_{m0}$

and $\Delta \text{SIC}$ taken along the long axis of Quadrilateral 2. Here, ASI-6km and BST-AMSR2 have the most north east ice edge, and highly forced waves rapidly grow under the north easterly wind forcing whereas the higher SIC forcing of the other simulations suppress the wave growth. Accordingly, $\Delta H_{m0}$ are less correlated with $\Delta \text{SIC}$ for the off-ice wave case. A distinct difference between the off- and on-ice wave cases regarding the $\Delta \text{SIC}$ effect on wave models is that $\Delta H_{m0}$ remains in the downwind open ocean whereas $\Delta \text{SIC} \to 0$. This is clearly shown in Figure 10c where $\Delta H_{m0} = [0.10, 0.60]$ when $\Delta \text{SIC} = 0$ indicating

the effect of $\Delta \text{SIC}$ forcing on model $\Delta H_{m0}$ can extend to the adjacent open water.

The off-ice wave fetch is not only controlled by the ice edge as the cumulative effect of the $\Delta \text{SIC}$ along the fetch distance would be affected by the wave-ice interactions as implemented in WW3. The current numerical approach to simulate wind pumping energy into waves in ice cover is dictated by SIC because the wave evolution is scaled like $(1 - c_i)(s_{\text{wind}} + s_{\text{dissipation}})$ (Equation 2). The parameterised attenuation rates are also important for wave evolution over ice as waves grow when $(1 -$

$c_i)(s_{\text{wind}} + s_{\text{dissipation}}) > c_i s_{\text{ice}}$. The difficulty of modelling off-ice waves using the Thomson et al. (2018) data have been described by Gemmrich et al. (2018) as they proposed a novel cumulative fetch idea to improve wave predictions. Evidently, off-ice wave evolution is a complex process.

Lastly, for both on- and off-ice wave cases, significant $\Delta H_{m0}$ extends to the waters where the wind forcing is orientated along the ice edge; so the model data are briefly examined in the region of MIZs north east of Wrangel Island shown as

Quadrilateral 1 in Figure 10a along the sea ice edge and north easterly wind forcing orientation. This region has considerable $\Delta \text{SIC}$ (not shown here) like Figure 5, and the model $\Delta H_{m0}$ is just as sizeable under the influence of high wind forcing. There is evidence of correlated bivariate uncertainty data in Figure 10b, and a combination of on- and off-ice wave features for the respective enhanced plots discussed in the previous paragraphs are depicted; however, deciphering the physical processes is complicated. Moreover, the modelled data in this region are strongly affected by whether wave evolution in ice cover follows

$(1 - c_i)(s_{\text{wind}} + s_{\text{dissipation}})$ (Equation 2), a question that has been discussed in Rogers et al. (2016); Thomson et al. (2018) with the latter citing Li et al. (2017) who confirmed wind input to high frequencies in the Antarctic Ocean. Future improvements of





wave-ice interaction parameterisations are envisaged to have implications for how wave evolution is simulated when much of the fetch is an MIZ. To illustrate how the bivariate uncertainty data appear along a transect as currently implemented in WW3, two cross sections along the ice edge and off-ice waves with high $\Delta H_{m0}$ on 21 November 2018 18:00 are plotted in Figure A8 390 of Appendix D.

### 5.2 Comparing SIC forcing and wave-ice interaction source term uncertainties

The hindcast experiment conducted in this study intentionally adopted the default IC3 source term parameters. As mentioned in Section 1, considerable progress in the WW3 wave-ice parameterisations has been made owing to the Thomson et al. (2018) measurements; for example, the WW3 manual states that combining scattering and nonlinear inelastic dissipation of Boutin 395 et al. (2018) can provide good results for dominant waves in both hemispheres (Ardhuin et al., 2018). On the other hand, Ardhuin et al. (2018) also explain that observation of wave attenuation could also be reproduced with many model forcing and parameter combinations, which as stated in the article is not unexpected because different wave-ice interaction processes are taking place along the wave propagation path. The knowledge of true SIC, sea ice thickness, ice rheology, and floe size distributions are surely necessary for the physical representation of wave evolution in ice-covered ocean: because only limited 400 sea ice information is available, tuning a wave-ice interaction source term to reproduce observations is not an easy task. While there is no doubt that model tuning to in situ observations can be achieved and is paramount for maritime applications, it can perhaps also be argued the model tuning has limited physical relevance to the knowledge of wave-ice interactions. As discussed in Section 2.3, WW3 offers three physics-based parameterisation source terms: IC2, IC3, and IC5. They are based on different attenuation mechanisms and dispersion relation, but their default ice rheological parameters (as given in the manual or the 405 WW3 regression test cases) also vary. Three principal parameters that form the sea ice forcing are: eddy viscosity (m²s⁻¹), $\nu$, ice density (kg²m⁻³), $\rho$, and effective elastic shear modulus (Pa), $G$, although $G$ is not used in IC2. The default $\rho$ values are consistent for all source terms. The $\nu$ values are 153.6e−3, 1.0e+0, and 5.0e+7 for IC2, IC3, and IC5, respectively, and the default $G$ values are 1.0e+3 and 4.9e+12 for IC3 and IC5, respectively. Rogers et al. (2016) adopted $G = 0$ assuming it is negligible in nilas and pancake ice fields. Based on the enormous range of tunable parameters and other factors described 410 in this paragraph, wave-ice interaction source term parameterisation is an error source in predicting waves in ice cover. To evaluate the relative significance of SIC forcing $\Delta H_{m0}$, we compared it with the wave-ice interaction source term $\Delta H_{m0}$ for the modelling period between 5 and 25 November 2018. The wave hindcast experiment was conducted using the IC2, IC3, and IC5 wave-ice interaction source terms based on the same model setup described in Section 2.3 except all three simulations were forced with the BST-AMSR2 sea ice forcing. Note that a SIC forcing sensitivity experiment was conducted using ASI-6km 415 with the same outcome as described below.

A pair of $\Delta H_{m0}$ for SIC forcing and wave-ice interaction source terms were calculated for the respective wave hindcast dataset. They were collated for the entire simulation period within the model domain shown in Figures 9a and 10a. Uncertainty distributions are visualised in a Q-Q plot by simply sorting each dataset, and this is shown in Figure 11. The figure depicts that both uncertainties are considerable with $max(\Delta H_{m0})$ values of 1.95 m and 1.44 m for the SIC forcing and wave-ice 420 interaction source term uncertainties, respectively. The values here do not provide indication against observations, however,





both uncertainty sources are sizeable based on the adopted default source term parameters. Our analysis demonstrates the accuracy of satellite retrieved SIC estimates used as model forcing is the more dominant error source of wave predictions in MIZs.

This outcome has significance to wave-ice interaction parameterisations of spectral wave models. Table 2 is a list of SIC
forcing used in the recent WW3 wave-ice interaction source term developments as well as the forcing of the wave models analysed in Section 4. It is interesting to learn that each wave-ice modelling study used different SIC data. Some studies employed numerical sea ice models or considered various sources and assessed their accuracy/suitability. For example, Rogers et al. (2016) used sea ice forecast (and analysis) as described in Hebert et al. (2015), which was produced specifically for the R/V Sikuliaq expedition of Thomson et al. (2018). Among various data assimilation sources, the model assimilates SIC
estimates based on the NASA-Team2 algorithm applied to SSMIS and the Bootstrap algorithm applied to AMSR2 (Hebert et al., 2015). Cheng et al. (2017)'s study, which is closely related to Rogers et al. (2016), considered various data sources including visual observation and the sea ice model of Hebert et al. (2015); they determined the SIC data retrieved from AMSR2 based on the NASA-Team2 algorithm was in better agreement with the in situ observations, and as such, their main study outcomes were derived from these data. Our firsthand experience suggests validation of satellite retrieved SIC in heterogeneous
MIZs is challenging as it would require in situ observations with comparable footprint, and the previous intercomparison studies conclude there is no one product that is superior than others (Ivanova et al., 2015; Comiso et al., 2017). There is not an immediate solution, but the SIC uncertainty revealed here surely warrants some attention when wave-ice interactions are studied/parameterised in future. Furthermore, SIC uncertainty likely also affects the lower atmospheric conditions as shown in Inoue et al. (2011); SIC uncertainty feedback to the lower atmosphere in a coupled air-sea-ice-wave system is perhaps of
interest to the atmospheric science community. Finally, when a mosaic of SAR NRCS data have sufficient coverage, it appears to be a powerful tool to depict irregular sea ice edges and heterogeneous sea ice fields of ice-covered ocean.

## 6   Conclusions

Reliable modelling of waves in ice cover is necessary as the global warming continues to melt the Arctic Ocean sea ice. Two primary uncertainty sources of wave predictions in MIZs are the accuracy of SIC forcing and wave-ice interaction parameteri-
sations. This study investigated the SIC uncertainty and its effect on modelling wave evolution in MIZs.

Satellite microwave radiometry is the most effective tool for retrieving comprehensive coverage of polar oceans' sea ice fields. The estimates, however, have considerable uncertainty, which has been a long-known issue within the remote sensing community. A 12 day fixed MIZ transect observation in the refreezing Chukchi Sea during the R/V Mirai November 2018 expedition provided a series of sea-truth images. These data were used to connect the satellite retrieved SIC estimates and the
refreezing ocean surface. Eight SIC data sets based on the four leading retrieval algorithms, NASA-Team, Bootstrap, OSISAF, and ARTIST-sea-ice applied to the SSMIS and AMSR2 sensors were analysed for comparison by interpolating the gridded SIC data in time and space to R/V Mirai's track. The sea-truth observation revealed that the refreezing MIZ sea ice forms in strips and patches of new and young ice within and in between the expanse of open water, and as such, the spatial distribution





is highly variable. Uncertainty in the interpolated SIC estimates along the R/V Mirai track is persistent throughout the 12
day transect observation period. A direct-qualitative gridded data comparison among three selected products, OSISAF-SSMIS,
BST-AMSR2, and ASI-3km for the SIC estimates on 15 November 2018 shows the SIC uncertainty in the region north east
of Wrangel Island is striking. SIC contours plotted over the mosaic of SAR NRCS images from Sentinel-1 demonstrate the
disparate 0.85 SIC contours for OSISAF-SSMIS and BST-AMSR2 can be some 200 km apart. Although only qualitative,
ASI-3km appears to be best equipped to reproduce the spatial variability and irregular ice edges that are depicted in the NRCS
images.

A wave hindcast experiment was conducted using an in-house wave model, TodaiWW3-ArCS based on WW3, and six
SIC products (discarding two products that were considered unsuitable) as model forcing to examine how the SIC uncertainty
translates to wave predictions in ice cover. The model estimates were compared with available in situ wave data, which revealed
that there is no one forcing or a model that can reproduce the observations. The model $H_{m0}$ uncertainty was examined in the
context of wave decay and growth for on- and off-ice wave cases, respectively. There is evidence that bivariate model $H_{m0}$
and SIC forcing uncertainty data are correlated, although off-ice wave evolution is more complicated as it is affected by the
cumulative effect of SIC uncertainty and the WW3 implementation of wave-ice interactions along an MIZ fetch.

Lastly, we compared the $\Delta H_{m0}$ distribution of the SIC forcing against the wave-ice interaction source term $\Delta H_{m0}$ distribu-
tion. Both uncertainties are found to be considerable during the simulation period with maximum $\Delta H_{m0}$ values of 1.95 m and
1.44 m, respectively. Although both uncertainty sources are significant, a Q-Q plot of the uncertainty distributions depicts the
accuracy of satellite retrieved SIC used as model forcing is the more dominant uncertainty source. The most important message
of this study is that without the knowledge of true SIC fields, accurately modelling wave evolution in and near MIZs remains
a difficult challenge.

## Appendix A:  R/V Mirai measurement system

R/V Mirai is equipped with two anemometers that were located on the foremast at 25 m elevation, and indicative wind condi-
tions at the ship positions were derived from 10 minute vector moving averages of 6 s interval instantaneous true wind speed
and direction. SST was measured −1 m below the sea surface with further 5 m inlet to the gauge while air temperatures were
measured on the foremast at 23 m elevation. Shipboard waves were obtained based on two methods: microwave radar system
(WM-2) (TSK Tsurumi Seiki Co., 2019) at the bow and stern of the ship, and nine-axis Inertial Moment Unit (Piper#15), which
is a variant but similar device of Kohout et al. (2015). WM-2 has a sampling frequency of 2 Hz and collects raw sea surface
elevation for 1,152 seconds at 35 minutes past each hour, and its integrated analogue system removes hull agitation and carries
out Doppler correction. Bulk parameters like the significant wave height and period are produced based on the zero-crossing
method. Wave observations during the campaign from the WM-2 integrated analog system (TSK Tsurumi Seiki Co., 2019)
were significantly affected by Doppler correction errors. Collins III et al. (2015) have shown shipboard measurements are less
affected by this effect when ship speed is $< 3\ \mathrm{ms^{-1}}$. Applying a $2\ \mathrm{ms^{-1}}$ ship speed threshold greatly reduced conspicuously spu-
rious data, and these are used in this study (e.g., Figure 6). Validation of these wave data remains as ongoing work. Piper#15





on board the vessel relies on an IMU, so it has limitations on measuring shorter waves. The processing method is consistent
with Kohout et al. (2015) except 15 minute intervals were used instead of 1 hour. Response Amplitude Operator of R/V Mi-
rai and the WM-2 data can be combined in theory to transfer IMU's high frequency signals to true surface elevation signals,
but post-processing is also still ongoing work. Reliable shipboard wave measurements require serious analysis to validate us-
ing the knowledge of ship response to true surface motions, and a more robust validation exercise is planned using the 2019
R/V Mirai October expedition that also employed a stereo-imaging system. Nevertheless, WM-2 produced seemingly sensible
wave estimates when ship speed was slow. Indicative wave heights combined with model estimates are still useful for analysis,
especially when two sensors have comparable estimates.

## Appendix B:  OSISAF-AMSR2 noise in open ocean

The OSISAF-AMSR2 SIC estimates during the November 2018 study period consist of prevalent erroneous estimates in the
open ocean. At the R/V Mirai positions, inaccurate SIC is estimated on 14 and 20 November when R/V Mirai was not in ice
cover. These estimates are noise likely arising from atmospheric effects because SSTs were too warm for new and young ice
to form, so only perennial ice could survive. R/V Mirai has strenuous restrictions on sailing near first-year and perennial sea
ice and sightings of them are logged by the experienced ice navigator. This is supported by the wave model as the TodaiWW3-
ArCS simulation using OSISAF-AMSR2 as forcing calculates $H_{m0}$ interpolated at R/V Mirai as 0 m during the Phase 2 on-ice
event as shown in Figure A1. The open ocean SIC estimates for 14 and 20 November are shown in Figure A2.

## Appendix C:  Satellite retrieved SIC and shipboard data at in situ observation locations

A supplementary material for Section 3 is provided in this Appendix. Figure A3 presents the shipboard measured wind and
SST data as well as bilinearly interpolated ERA5 10 m wind speeds. Figures A4 to A6 present full time series of satellite
retrieved SIC interpolated in time to hourly intervals and bilinearly interpolated in space at the R/V Mirai positions for the
fixed transect observation period between 9 and 20 November 2018 as described in Section 3.1. The figure schematics follow
Figure 2. Lastly, satellite retrieved SIC data at the Piper#13 drifting wave buoy locations also interpolated in time and space
are shown in Figure A7.

## Appendix D:  Cross sections of bivariate uncertainty data

Two cross sections of the bivariate model $H_{m0}$ and SIC forcing uncertainty data are presented for two transects shown in
Figure A8 on 21 November 2018 18:00. The green line is a transect for which the wind forcing is orientated along the ice
edge, and the magenta line shows off-ice wave evolution that is affected by the SIC uncertainty. Note that the green line is not
a transect of wave evolution.





*Author contributions.*   TN conducted the analysis. TN prepared and all contributed to the manuscript. JI designed the R/V Mirai sea-truth observational plan.

*Competing interests.*   Authors declare we have no competing interests.

*Acknowledgements.*   This work was sponsored by the Japanese Ministry of Education, Culture, Sports, Science, and Technology through the ArCS project. A part of this study was also conducted under JSPS KAKENHI Grant Number JP 16H02429. JI acknowledges support
from the KAKENHI grant numbers 18H03745 and 18KK0292. The authors greatly appreciate the comprehensive data available freely on the web. Satellite retrieved SIC were obtained from these sources (Cavalieri et al., 1996; Meier et al., 2018; Comiso, 2017; Hori et al., 2012; Spreen et al., 2008). The OSI-401-b and OSI-408 were used and are the product of the EUMETSAT Ocean and Sea Ice Satellite Application Facility. Sentinel-1 SAR data were downloaded from NOAA (2019). ERA5 data were downloaded from Copernicus (2019), and ARCMFC wave model data were obtained from ARCMFC (2019). We are grateful to the R/V Mirai crew on board MR18-05C who made the sea-truth
observation possible.



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





FIGURES.

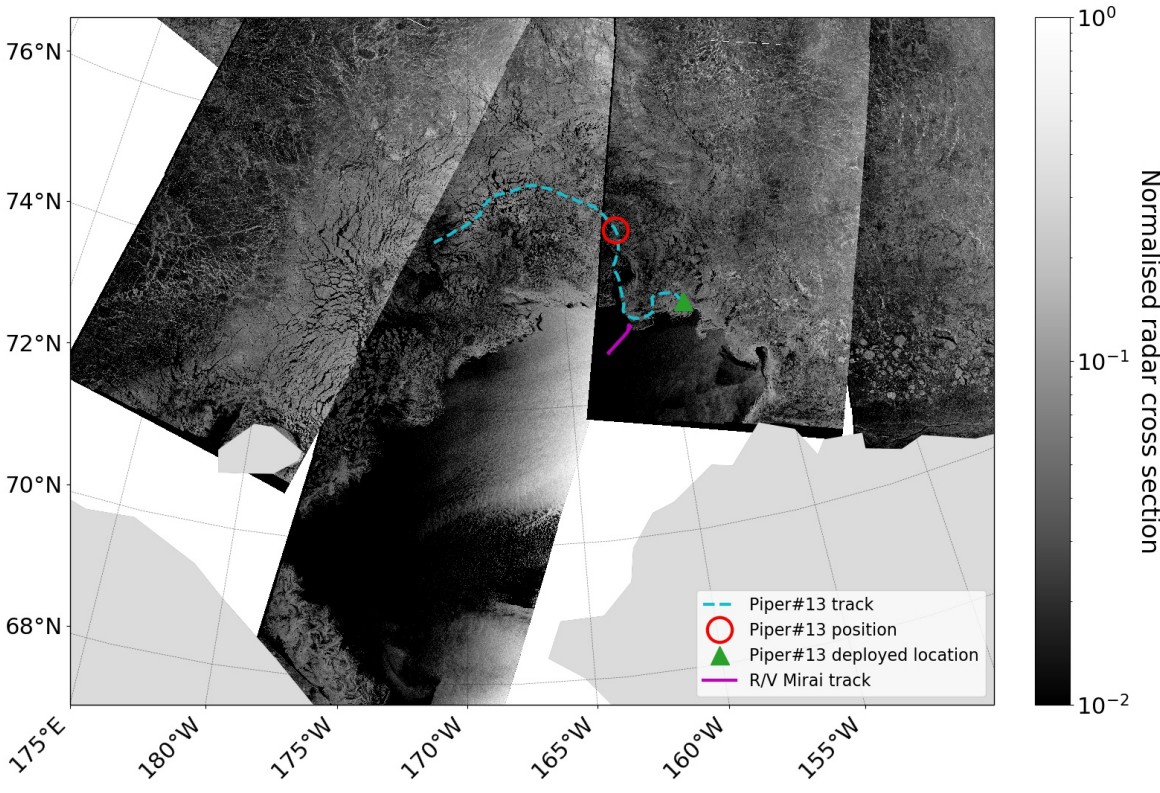

**Figure 1.** Observation locations are overlaid on the mosaic of Sentinel-1 NRCS images (NOAA, 2019) acquired on 15 November 2018. R/V Mirai track on this date is shown as the solid magenta line, and the Piper#13 drifting wave buoy track between 6 and 28 November is shown in the dashed cyan line. The green triangle shows the deployment location, and the red circle represent the buoy position on 15 November 12:00.

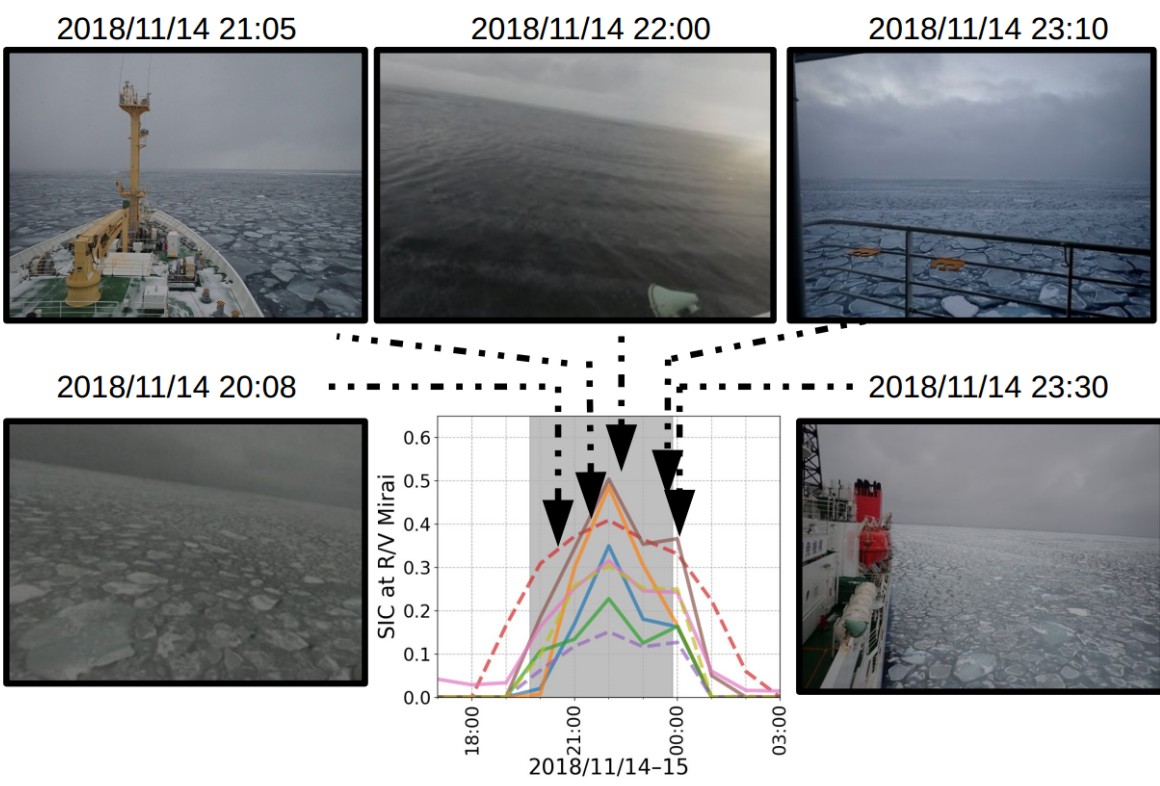

**Figure 2.** Sea-truth images compared with satellite retrieved SIC estimates from eight products on 14 November 2018 during the on-ice wave event. The figure schematics of SIC estimates are as follows: ASI-3km (blue), ASI-6km (orange), BST-AMSR2 (green), BST-SSMIS (red), NT2-AMSR2 (purple), NT-SSMIS (brown), OSISAF-AMSR2 (pink), and OSISAF-SSMIS (olive), and SSMIS and AMSR2 are distinguished by dashed and solid lines, respectively. Grey highlighted times indicate when the vessel was in ice cover based on the ice navigator's logs: from the first (known) encounter of sea ice to the ship proceeding to the ice-free water.


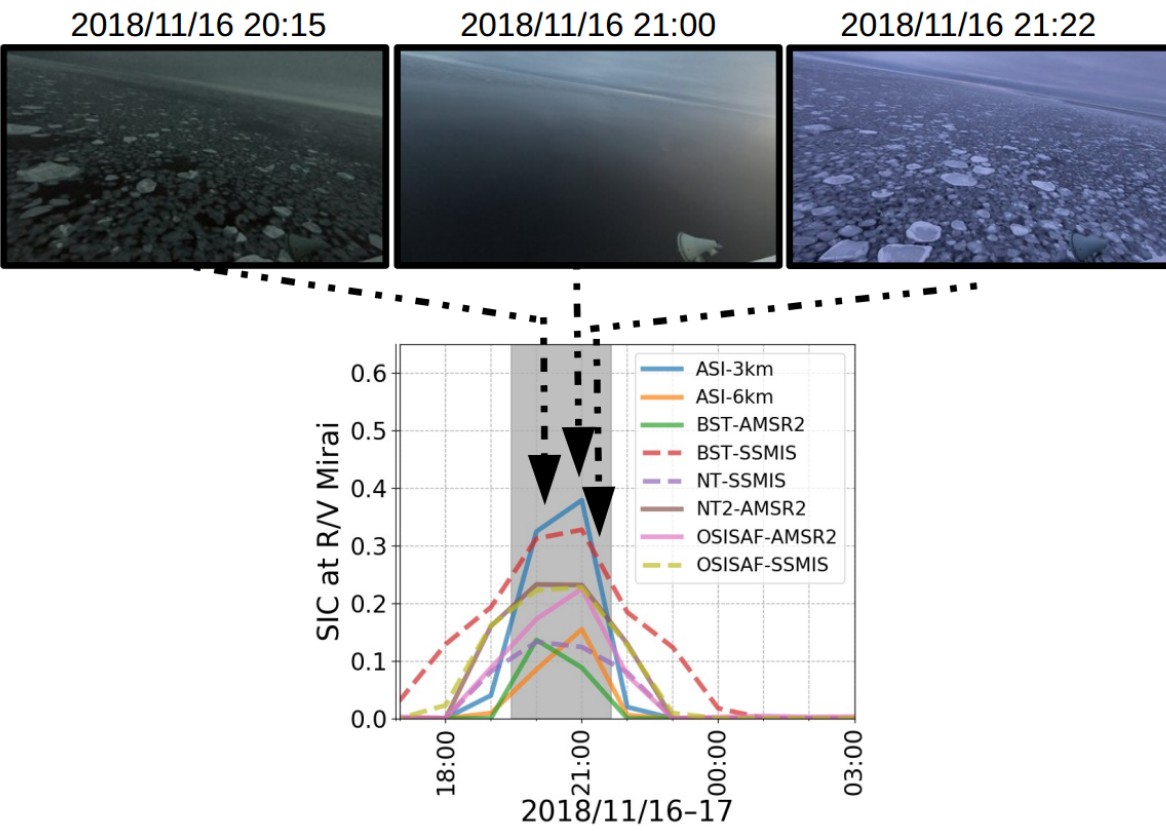

**Figure 3.** Sea-truth images compared with satellite retrieved SIC estimates from eight products on 16 November 2018. The figure schematics follow Figure 2.



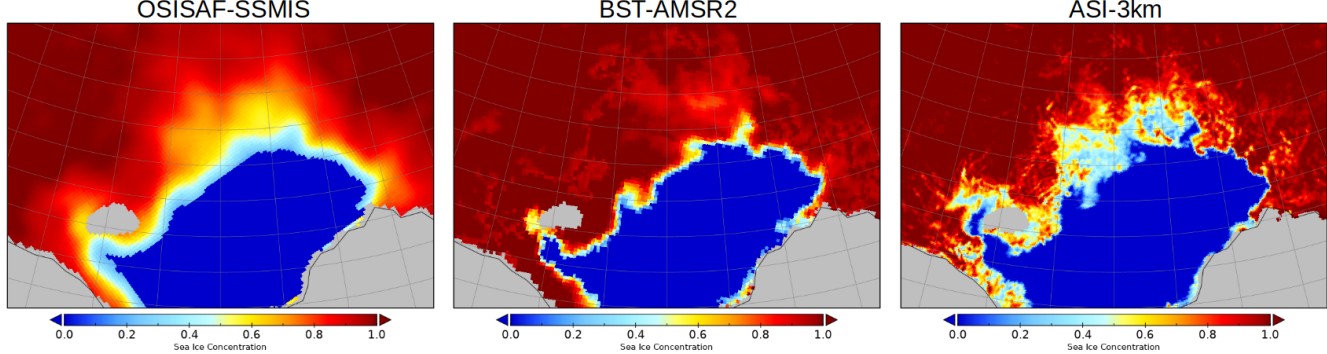

**Figure 4.** SIC estimates of OSISAF-SSMIS, BST-AMSR2, and ASI-3km for 15 November 2018 showing considerable uncertainty.



**Figure 5.** 0.15 (solid) and 0.85 (dashed) SIC contours of OSISAF-SSMIS, BST-AMSR2, and ASI-3km for 15 November 2018 shown respectively as cyan, lime green, and magenta lines are overlaid on the mosaic of Sentinel-1 NRCS images (NOAA, 2019) acquired on the same day.



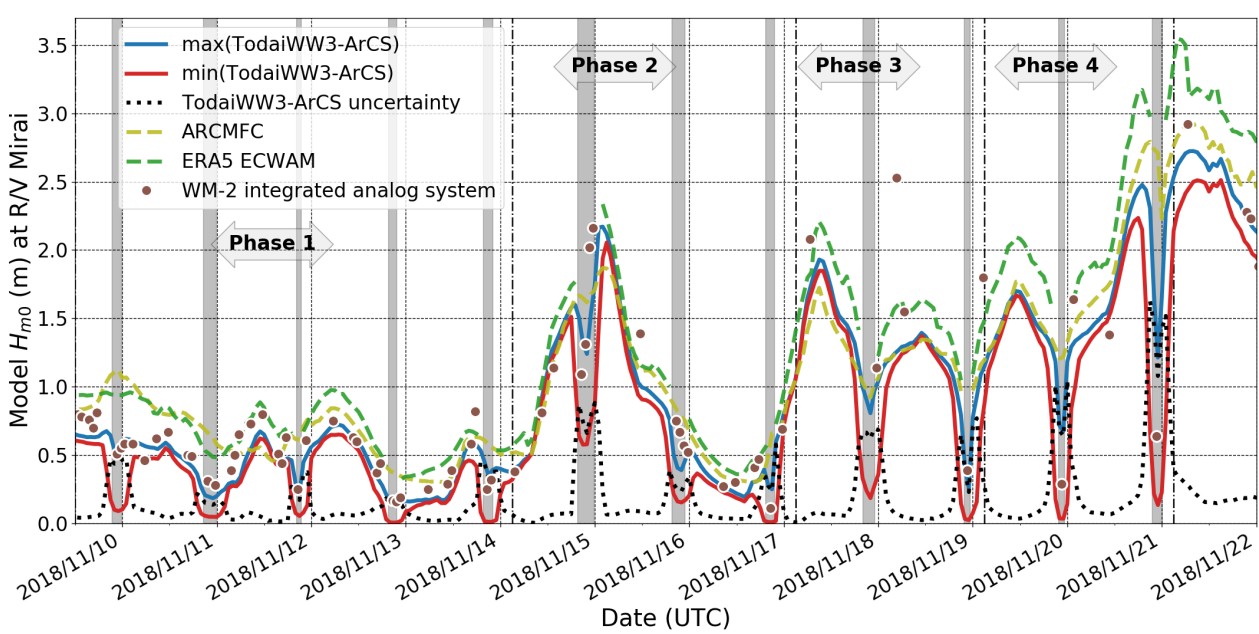

**Figure 6.** $\Delta H_{m0}$ of TodaiWW3-ArCS estimates using various SIC products as sea ice forcing interpolated at R/V Mirai positions are shown during the transect observation . The figure also shows the WM-2 data when R/V Mirai ship speed was $< 2\,\mathrm{ms}^{-1}$. Two independent predictions from ERA5 ECWAM and the ARCMFC wave model are presented to indicate the quality of TodaiWW3-ArCS. Grey highlighted times indicate when the vessel was in ice-covered sea based on the ice navigator's logs. Note that ERA5 ECWAM uses ice mask for SIC $> 0.30$ based on the sea ice forcing used in their model.



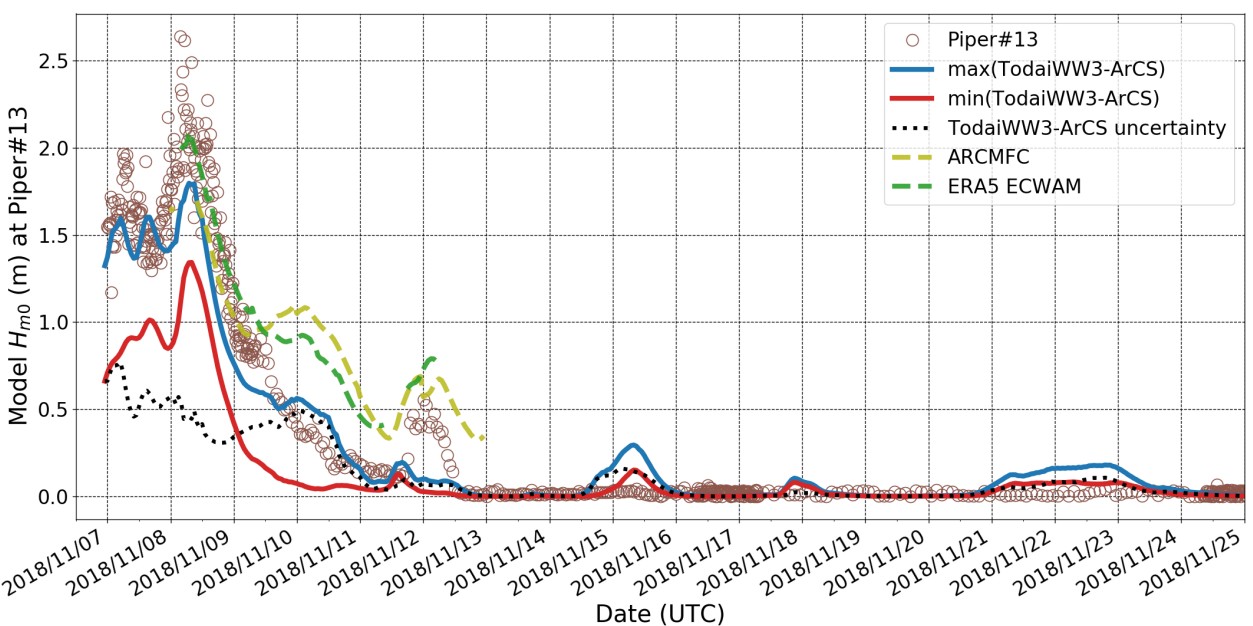

**Figure 7.** Piper#13 wave data are presented with the $\Delta H_{m0}$ of TodaiWW3-ArCS using various SIC products as sea ice forcing interpolated at the Piper#13 positions. Two independent predictions from ERA5 ECWAM and the ARCMFC wave model are also presented to indicate the quality of TodaiWW3-ArCS.





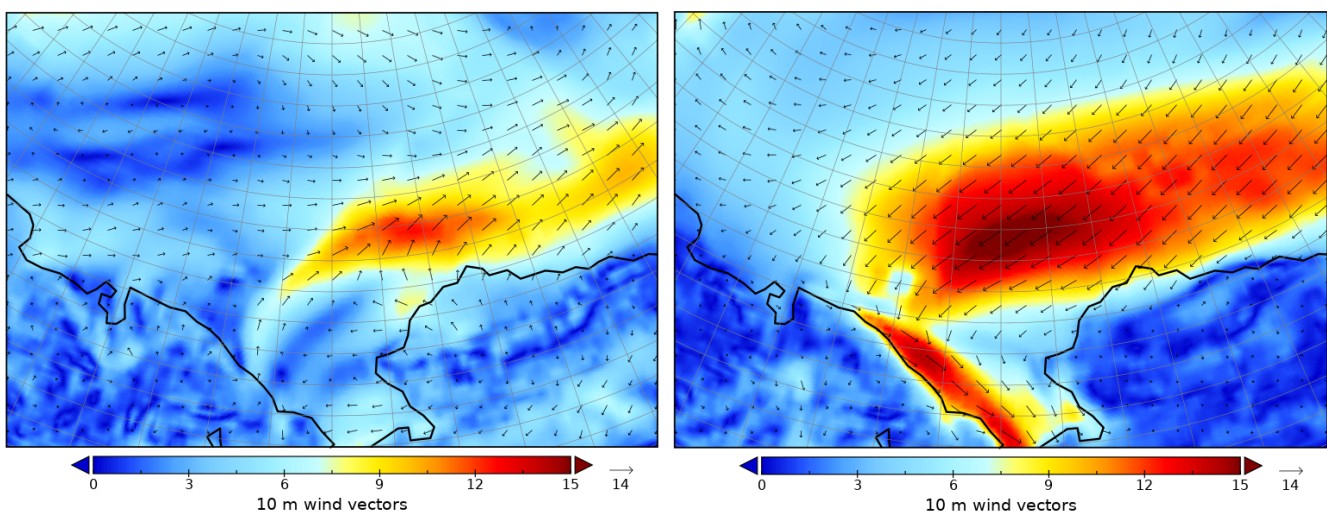

(a) South westerly on-ice winds on 15 November 2018 00:00.   (b) North easterly off-ice winds on 21 November 2018 21:00.

**Figure 8.** ERA5 10 m wind speed (ms$^{-1}$) and vectors to depict the forcing for the selected on- and off-ice wave cases.





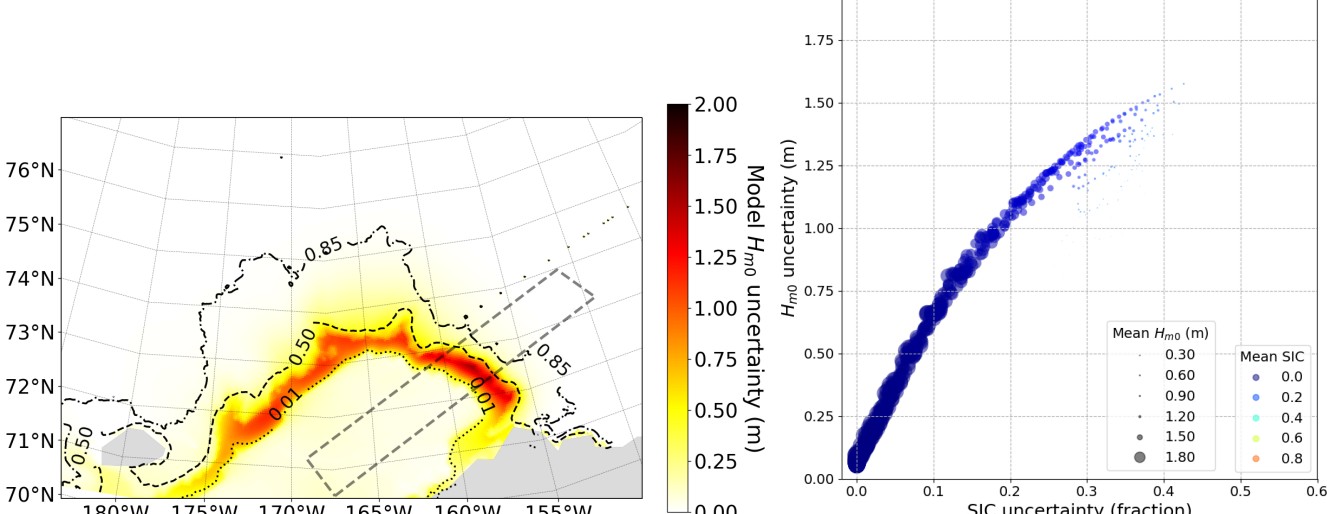

(a) TodaiWW3-ArCS $\Delta H_{m0}$ map with 0.01 (dotted), 0.50 (dashed), and 0.85 (dash-dotted) $mean$(SIC forcing) contours shown in black. Data enclosed in the grey dotted quadrilateral, which is orientated along the south westerly wind forcing direction, are plotted in Figure 8b.

(b) Bivariate model $\Delta H_{m0}$ and $\Delta$SIC forcing uncertainty data for the on-ice wave case are shown in an enhanced scatter plot for the quadrilateral area in Figure 9a. The marker sizes are scaled by $mean(H_{m0})$, and the marker colours indicate $mean$(SIC forcing).

**Figure 9.** TodaiWW3-ArCS simulation on 15 November 2018 00:00 during on-ice south westerly wind conditions as shown in Figure 8a.





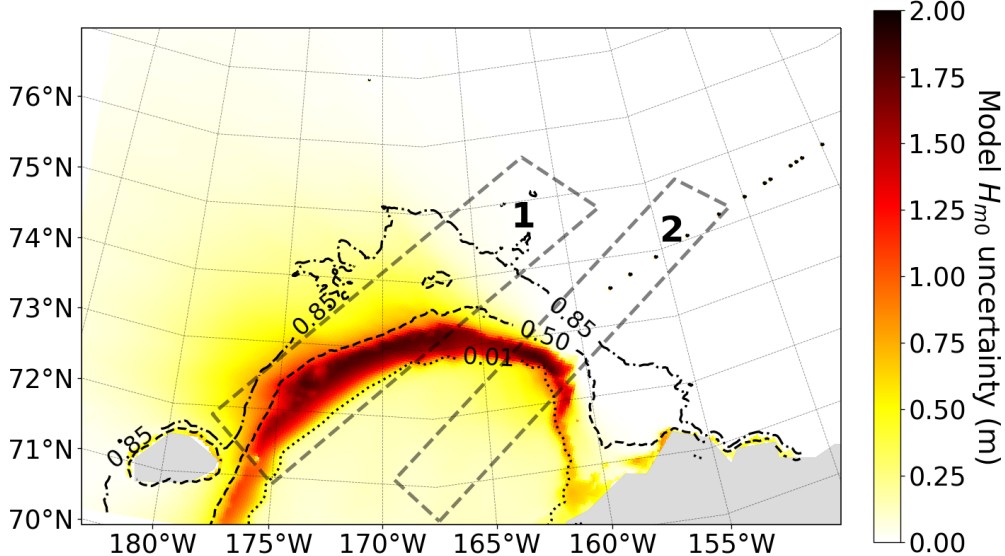

(a) TodaiWW3-ArCS $\Delta H_{m0}$ map with 0.01 (dotted), 0.50 (dashed), and 0.85 (dash-dotted) $mean$(SIC forcing) contours shown in black. Data enclosed in two grey dotted quadrilaterals are plotted in Figures 10b and 10c. Quadrilateral 1 depicts region where the north easterly wind forcing is orientated along the ice edge, and Quadrilateral 2 reflects the north easterly off-ice wave case.

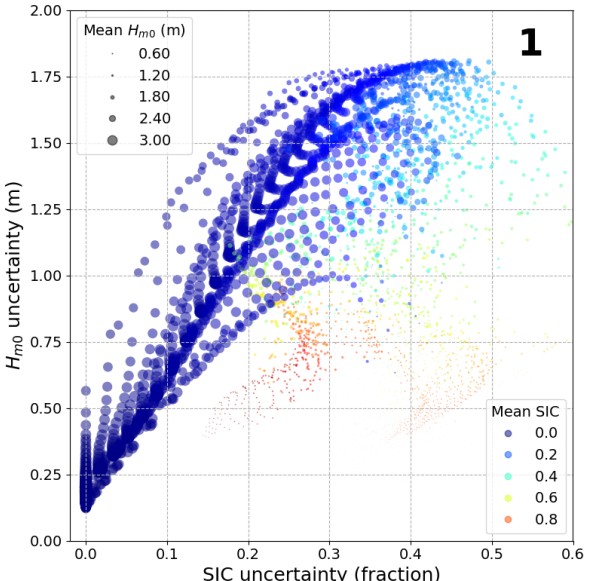

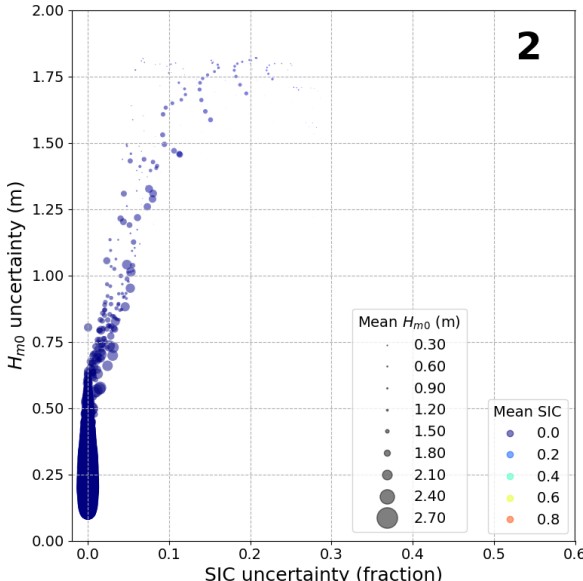

(b) Bivariate model $\Delta H_{m0}$ and $\Delta$SIC forcing uncertainty data shown in an enhanced scatter plot for the Quadrilateral 1 area in Figure 10a where the wind forcing is orientated along the ice edge. The marker sizes are scaled by $mean(H_{m0})$, and the marker colours indicate $mean$(SIC forcing).

(c) Bivariate model $\Delta H_{m0}$ and $\Delta$SIC forcing uncertainty data for the off-ice wave case are shown in an enhanced scatter plot for the Quadrilateral 2 area in Figure 10a. The marker sizes are scaled by $mean(H_{m0})$, and the marker colours indicate $mean$(SIC forcing).

**Figure 10.** TodaiWW3-ArCS simulation on 21 November 2018 21:00 during north easterly off-ice wind conditions as shown in Figure 8b.



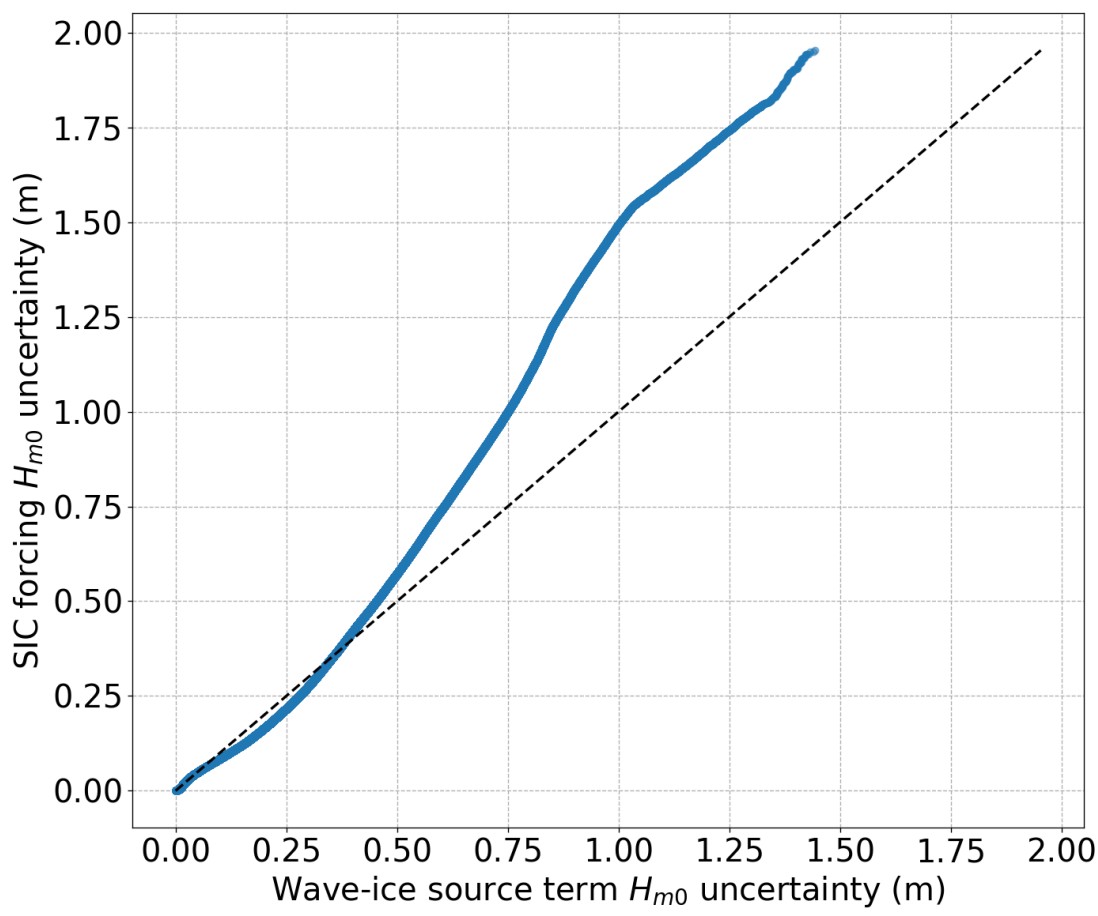

**Figure 11.** A Q-Q plot depicting the $\Delta H_{m0}$ distributions for SIC forcing and wave-ice source term datasets for which the former is shown to be the more dominant uncertainty source.

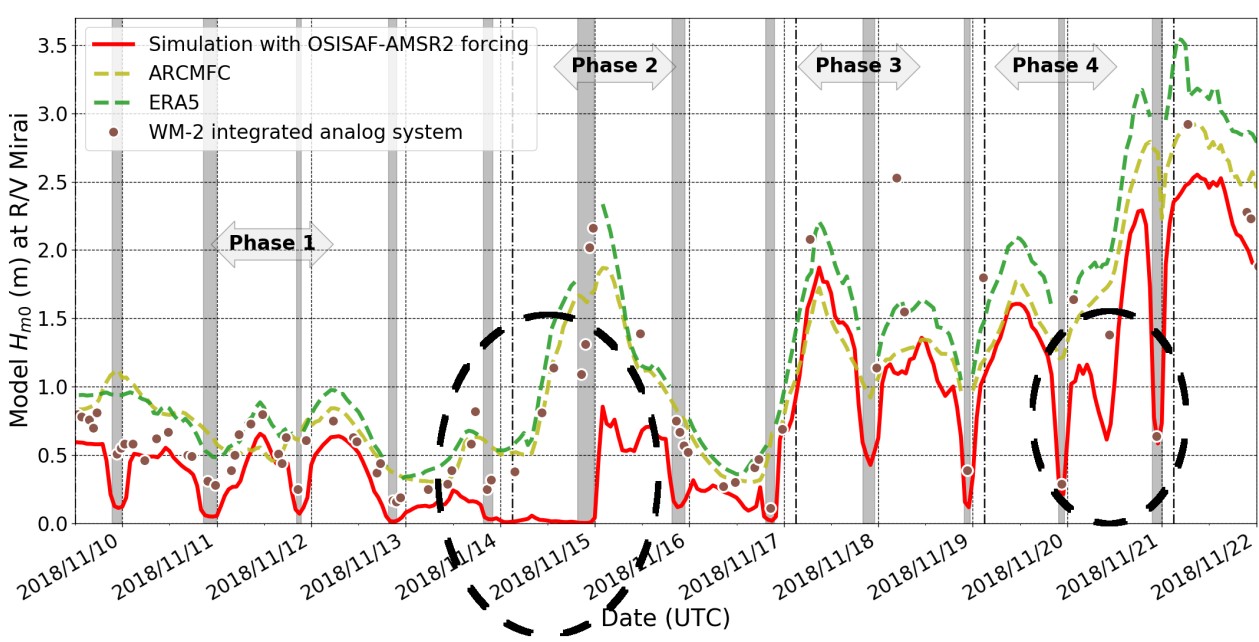

**Figure A1.** The TodaiWW3-ArCS $H_{m0}$ estimates using the OSISAF-AMSR2 SIC as forcing interpolated at R/V Mirai positions are shown during the transect observation. The figure also shows the WM-2 data when R/V Mirai ship speed was $< 2\,\mathrm{ms}^{-1}$. Two independent predictions from ERA5 ECWAM and the ARCMFC wave model are also shown. Blacked dotted circles indicate times when the erroneous SIC forcing caused inaccurate estimates of $H_{m0}$ at the R/V Mirai position. Grey highlighted times indicate when the vessel was in ice covered based on the ice navigator's logs.





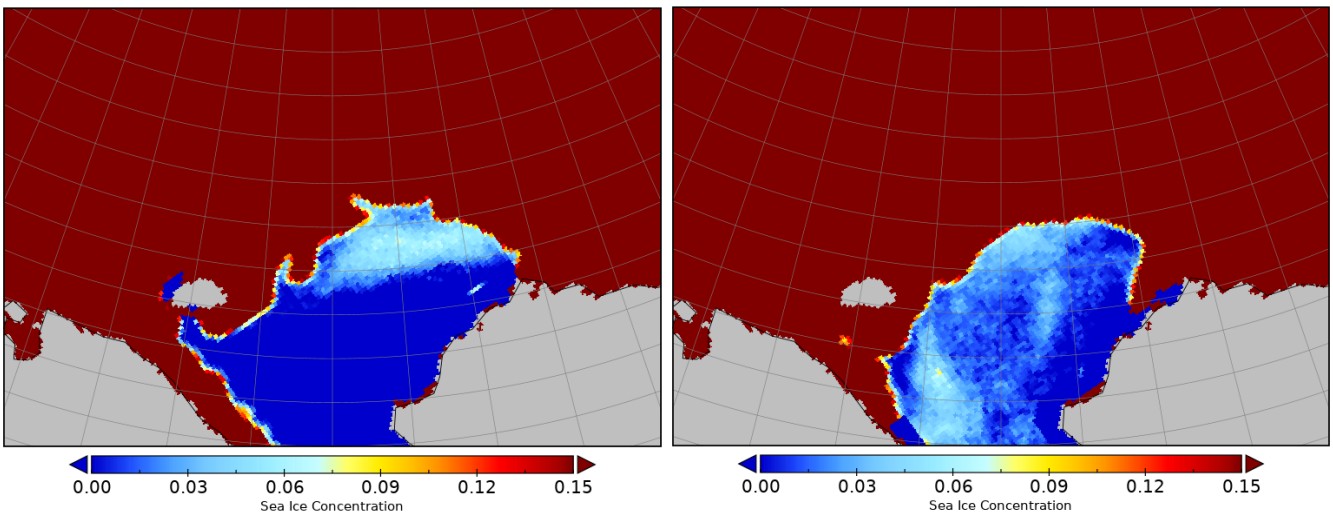

(a) SIC estimates on 15 November 2018.

(b) SIC estimates on 20 November 2018.

**Figure A2.** Apparent OSISAF-AMSR2 SIC noise in the open water during the R/V Mirai transect observation.



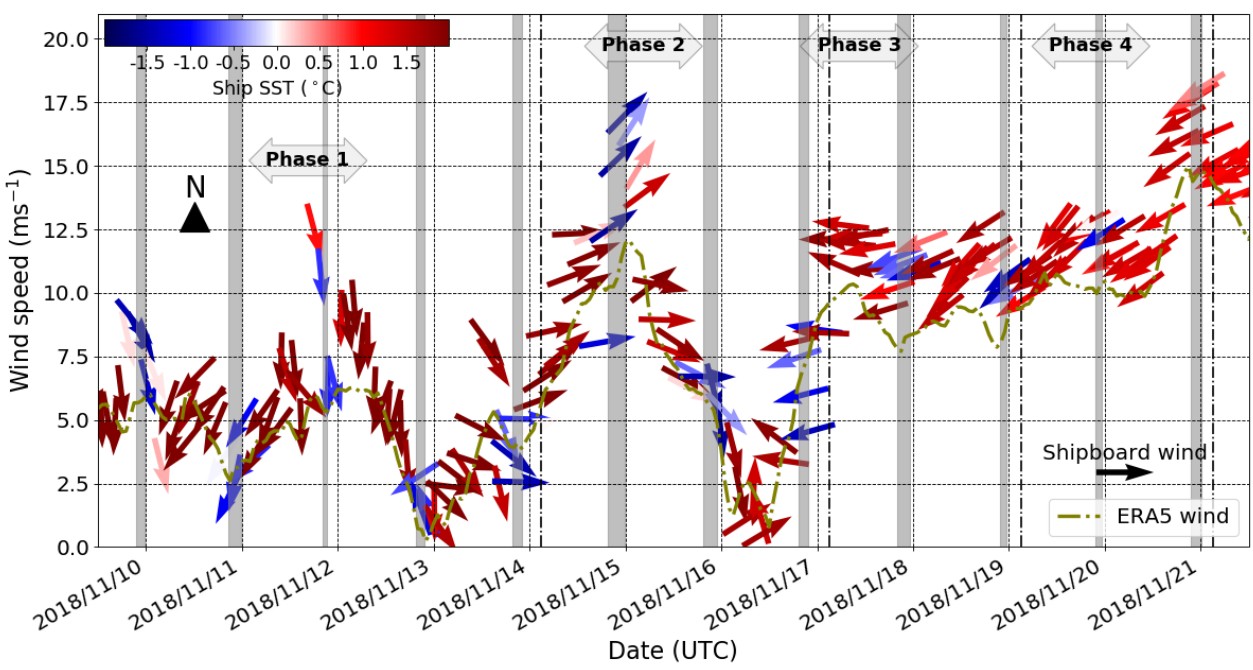

**Figure A3.** Shipboard wind and SST and bilinearly interpolated ERA5 10 m wind at the R/V Mirai position. Grey highlighted times indicate when the vessel was in ice cover based on the ice navigator's logs: from the first (known) encounter of sea ice to the ship proceeding to the ice-free water.





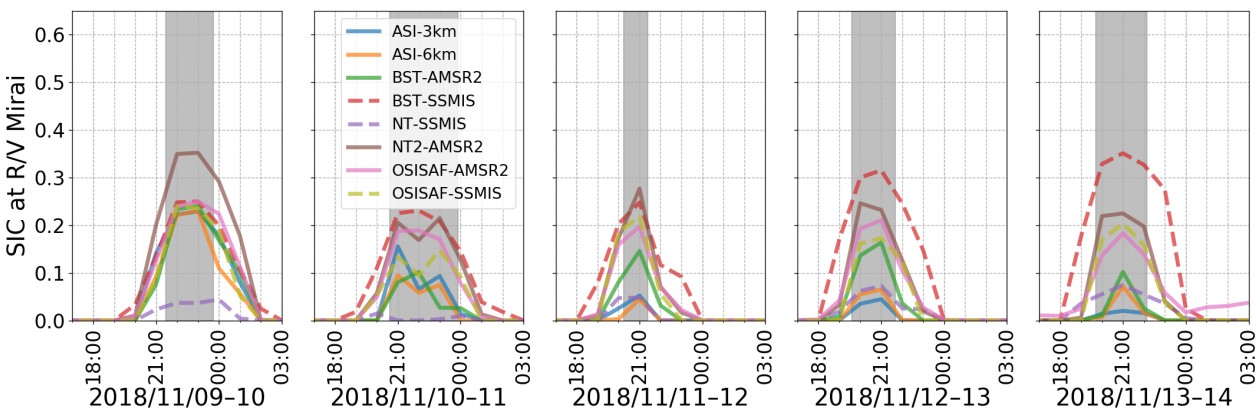

**Figure A4.** Satellite retrieved SIC for all products along the R/V Mirai track during Phase 1 between 9 and 13 November 2018. The figure schematics follow Figure 2. Grey highlighted times indicate when the vessel was in ice cover based on the ice navigator's logs.



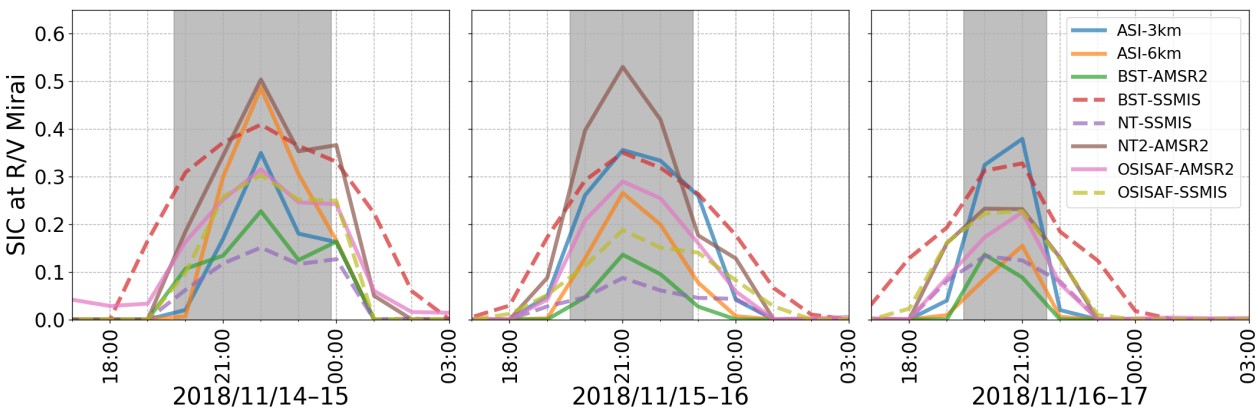

**Figure A5.** Satellite retrieved SIC for all products along the R/V Mirai track during Phase 2 between 14 and 16 November 2018. The figure schematics follow Figure 2. Grey highlighted times indicate when the vessel was in ice cover based on the ice navigator's logs.





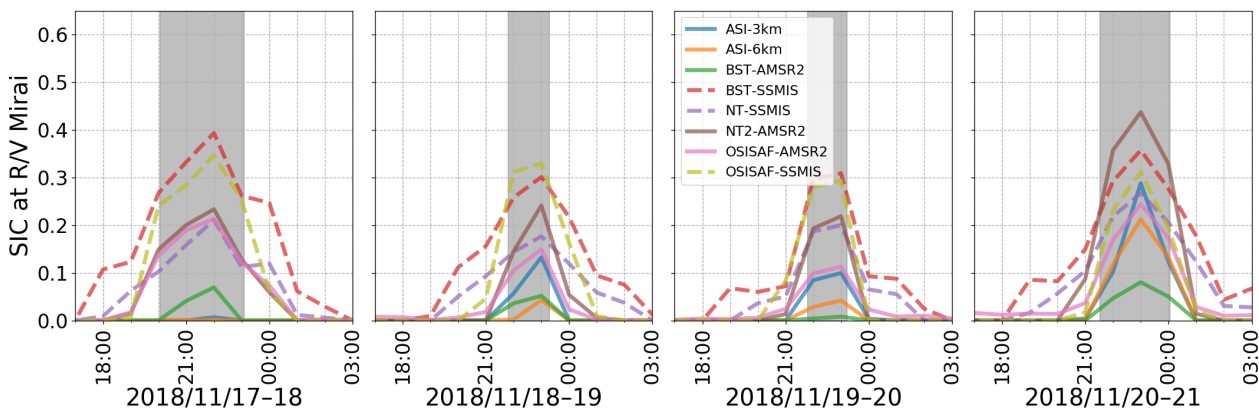

**Figure A6.** Satellite retrieved SIC for all products along the R/V Mirai track during Phases 3 and 4. between 17 and 20 November 2018. The figure schematics follow Figure 2. Grey highlighted times indicate when the vessel was in ice cover based on the ice navigator's logs.

The Cryosphere



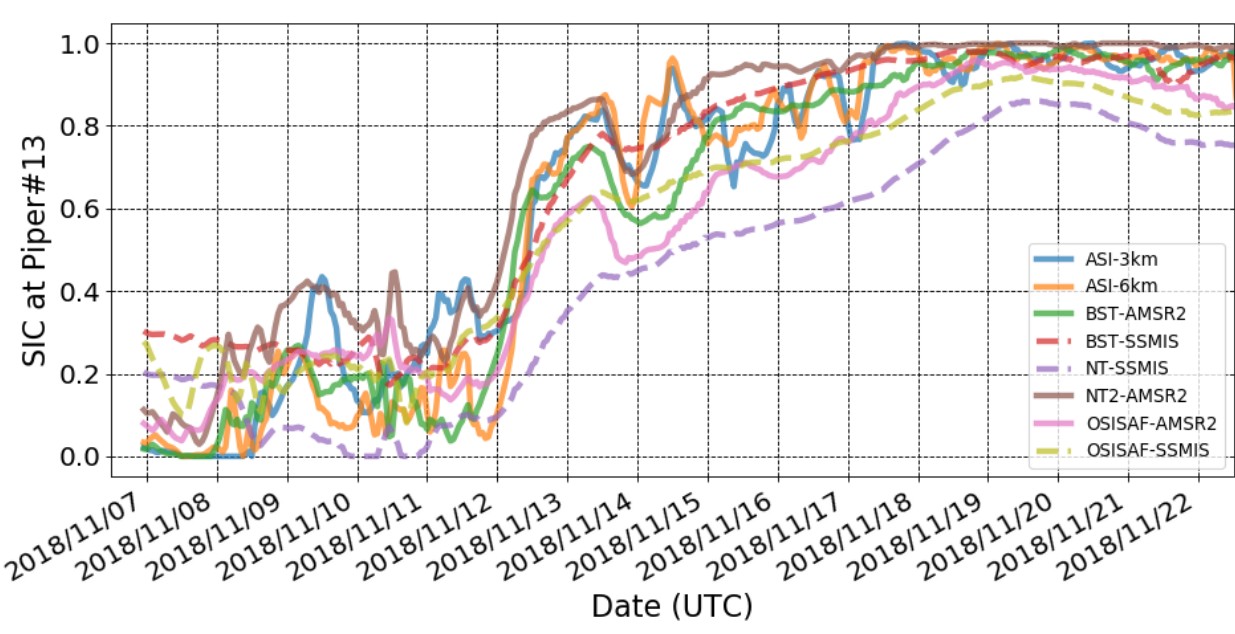

**Figure A7.** Satellite retrieved SIC for all products along the Piper#13 track between 7 and 25 November 2018. The figure schematics follow Figure 2.


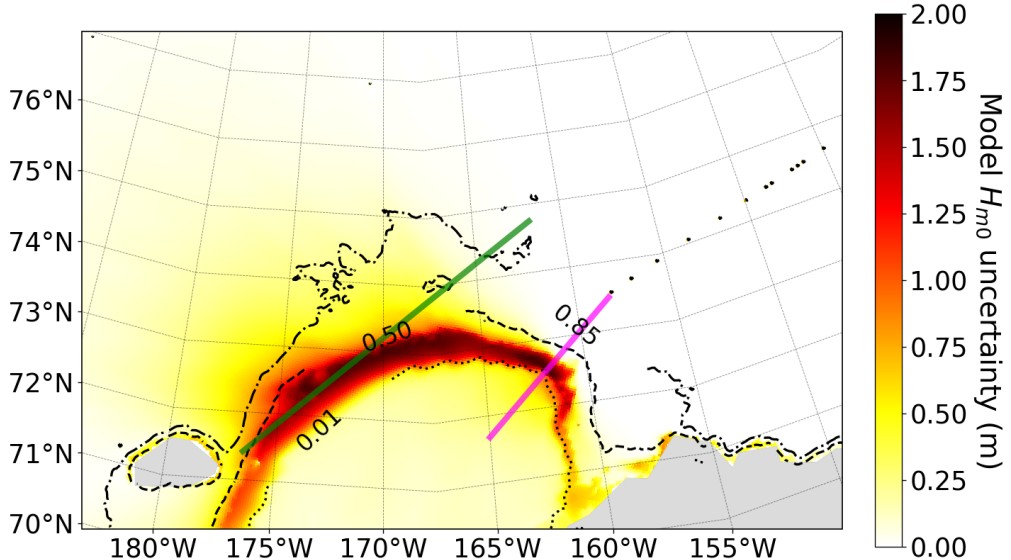

(a) TodaiWW3-ArCS $H_{m0}$ uncertainty map with 0.01 (dotted), 0.50 (dashed), and 0.85 (dash-dotted) $mean$(SIC forcing) contours shown in black. Cross sections of $H_{m0}$ and SIC forcing along the green and magenta transects are shown in Figures A8b and A8c, respectively.

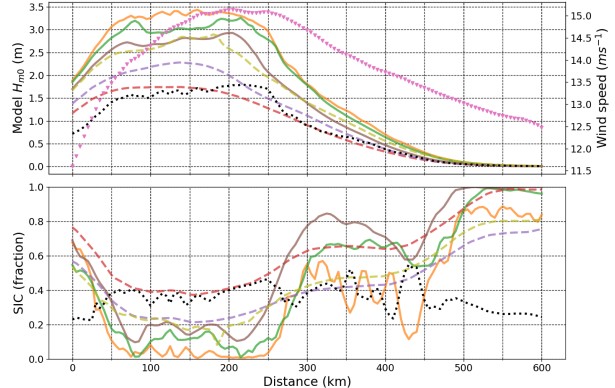

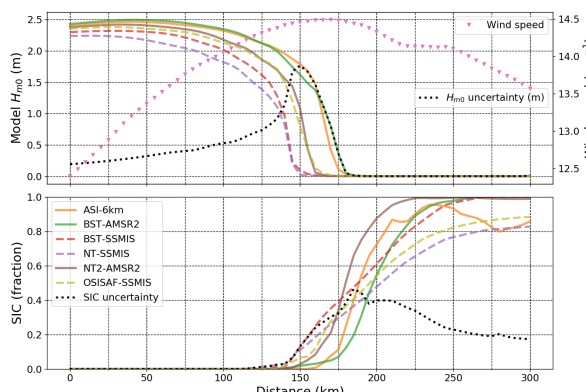

(b) Model $H_{m0}$ (top) and SIC (bottom) forcing along the green line in Figure A8a are shown for each SIC forcing simulation. The figure schematics follow Figure 2. Magenta markers indicate wind forcing magnitude, and dotted black lines represent respective uncertainties.

(c) Model $H_{m0}$ (top) and SIC (bottom) forcing for an off-ice wave transect shown as a magenta line in Figure A8a are shown for each SIC forcing simulation. The figure schematics follow Figure 2. Magenta markers indicate wind forcing magnitude, and dotted black lines represent respective uncertainties.

**Figure A8.** TodaiWW3-ArCS simulation on 21 November 2018 21:00 for the north easterly off-ice wave case.





TABLES

**Table 1.** Details of satellite retrieved SIC products used in this study.

| Product name | Instruments | Abbreviation | Data reference (specified grid resolution) |
|---|---|---|---|
| NASA-Team (NT) | SSMIS | NT-SSMIS | Cavalieri et al. (1996) (25 km) |
| NASA-Team 2 (NT2) | AMSR2 | NT2-AMSR2 | Meier et al. (2018) (12.5 km) |
| Comiso-Bootstrap (BST) | SSMIS | BST-SSMIS | Comiso (2017) (25 km) |
| | AMSR2 | BST-AMSR2 | Hori et al. (2012) (10 km) |
| OSISAF | SSMIS | OSISAF-SSMIS | OSI-401-b: SIC product of the EUMETSAT Ocean and Sea Ice Satellite Application Facility (10 km) |
| | AMSR2 | OSISAF-AMSR2 | OSI-408: AMSR-2 SIC product of the EUMETSAT Ocean and Sea Ice Satellite Application Facility (10 km) |
| ARTIST-Sea-Ice (ASI) | AMSR2 | ASI-6km | Spreen et al. (2008) (6.25 km Arctic grid) |
| | AMSR2 | ASI-3km | Spreen et al. (2008) (3.125 km Chukchi-Beaufort grid) |





**Table 2.** A list of SIC data source used for various wave-ice interaction modelling studies.

| Reference | Model (wave-ice interaction) | SIC data source |
|---|---|---|
| Rogers et al. (2016) | WW3 (IC3) | NASA-Team2 applied to SSMIS and Bootstrap applied to AMSR2 (assimilated in sea ice model) |
| Cheng et al. (2017) | WW3 (IC3) | NASA-Team2 applied to AMSR2 |
| Ardhuin et al. (2018) | WW3 (IC2 including IS2 scattering) | ARTIST-Sea-Ice applied AMSR2 |
| Copernicus (2019) | ERA5 ECWAM (ice mask) | OSISAF applied to SSMIS (indirectly from OSTIA (Donlon et al., 2012)) |
| ARCMFC (2019) | ARCMFC wave model (Sutherland et al., 2019) | OSISAF applied to SSMIS (assimilated in sea ice model) |