# Peer review of "Satellite retrieved sea ice concentration uncertainty and its effect on modelling wave evolution in marginal ice zones"

_The Cryosphere, 2019_

## Referee Comment (RC1) · Anonymous Referee #1 · 30 Dec 2019

**General comments:**

This paper aims to study the uncertainty of sea ice concentration (SIC) retrieval from satellites with a particular focus on the effect of this uncertainty on the wave height estimate by spectral wave models in the MIZ. The authors compare 8 different SIC products inferred from 2 different passive microwave sensors with observations made during a cruise in the Chukchi Sea in 2018 and SAR images that coincide with this cruise. Their conclusion is that none of these products are able to provide a consistent estimate of the SIC in the MIZ, mostly because sea ice in this region is very dynamic and heterogeneous. Following this analysis, they run a set of hindcasts using the spectral wave model WW3 forced by the different SIC products and discuss the sensitivity of the resulting wave height in the MIZ to the estimated SIC field. They show that this sensitivity is substantial, mainly due to the fact that the wave height decay in ice is very quick, and the SIC products strongly disagree on the position of the sea ice edge. They end their discussion by showing that wave model results are actually more sensitive to the SIC forcing than to the choice of the wave-in-ice parameterisations that they tested.

This paper presents interesting results, confirming some statements about the sensitivity of wave-in-ice modelling to sea ice concentration made in previous studies. The fact that the uncertainty in the estimated sea ice concentration has a larger effect on the uncertainty of the wave height estimated in-ice than the change in the wave-in-ice attenuation parameterization is a nice finding that illustrates very well the difficulties faced by wave-in-ice modellers. However, the paper suffers from a writing style that is confusing in a number of places. This is particularly the case for sections 3, 4 and 5. It is a bit paradoxal, as sometimes the important information is hidden in a succession of very wordy sentences, making it hard for the reader to get the message, and sometimes it seems that the authors wanted to avoid repeating themselves whereas the reader would happily appreciate some help. In the following comments, I will try to point out some of these unclear paragraphs, but overall the whole paper should undergo a careful rewriting aiming to make it clearer. I am also not convinced by the usefulness and the novelty of section 3, at least in its current form. To me, the most interesting part of the study lies in section 5, but it is overshadowed by the lack of clarity of the previous sections. It would be worth shifting the emphasis of the paper to this section more quickly. I will only recommend this paper for publication after these problems have been properly addressed. I also have other specific comments concerning the content, which require at least an answer from the authors. I also suggest a non-exhaustive list of typos and sentences that, in my opinion, need to be rephrased.

**Specific comments:**

P1L22: "model uncertainties...", which models are the authors talking about here?

P2L25: The authors should consider giving a definition of the MIZ.

P2L36: "wave-ice interaction source term": the authors haven't introduced the principle of spectral wave models yet, so it is not clear for everyone what the wave-ice interaction source term is.

The introduction is overall pretty clear and the gap in knowledge clearly identified.

P3L64: Here the authors introduce the methods used for the measurements on board on the R/V Mirai. But the way the so called "sea truth" images were taken is only described in section 3.1 (P7L212), lost in comments on the results. Similarly, before the end of section 3.1, a definition of how the uncertainty of a given quantity is computed is suddenly introduced, without even a proper

transition (P8L225). This mix between comments on the results and details of the methods makes section 3.1 very confusing and much longer than it should be.

P4L88 I think the word "translated" is not appropriate here (and in some other places). The authors could consider using "interpreted", "inferred".

P4L118 The sentences about the different grids the authors could have used and the one they are actually using are really confusing. Maybe they should try to cut them into more but shorter sentences, each dealing with one region and one resolution.

P5L143 "A curvilinear grid [...] sea point cells." I found this whole paragraph very confusing. As a reader I found very difficult to understand the links between each sentence, and the expression "The grid" seems to be applied to different things. As an example, they first refer to the model "geographical" grid, then to the spectral grid, then they use again the expression "the grid" to give details about the bathymetry. A quick reminder of the region they are focusing on would also be welcome, especially as they refer to the "other seas" at the border of "the domain".

P6L152 "During the version upgrade of TodaiWW3-ArCS...": which version upgrade are the authors referring to? Are they sure it is relevant for the paper? I think the authors could just state that they are using the ST6 parameterization for the non-ice source terms as it was previously shown to give the best results for the case discussed in Nose et al. (2018) with the model being forced by ERA5 winds.

P6L157 "The s_ice term is composed": I think the use of composed is misleading here. The attenuation terms the authors mention are included in different parameterizations (ISX, ICX), and they are not all compatible with each other. I would suggest an expression like "The s_ice term represents wave-in-ice attenuation processes such as..." for instance.

P6L160 "The dominant floe size [...] IS0 switch.": Here I have the feeling that the authors want to justify why they did not include scattering terms in their wave-in-ice source terms. I think this justification is very long and with unnecessary information (the way scattering terms work in WW3 for instance). I think it would be much clearer simply stating that during the cruise, sea ice in the MIZ was mainly made of grease, nilas and pancake ice, for which scattering is not expected to be the dominant process (Montiel et al., 2018), and therefore scattering was not considered here. Specifying the WW3 switch IS0 is also unnecessary.

P6L170 "The underlying principle of sea ice models is that sea ice is treated a continuum." Firstly, there is a small typo, it should be "treated as...". Secondly, this statement might be true for the sea ice models used in climate models, but it ignores discrete elements sea ice models, often used for sea ice-structure interactions. They can also be used to study wave-ice interactions (Herman et al., 2015). Actually I think the sentences between P6L170 and P6L173 could be shortened. The reason the authors choose this parameterization is because it has been developed to represent similar ice conditions to the ones encountered by the R/V Mirai, which is not the case for the other parameterizations.

P6L175 "... the treatment of independent SIC and sea ice thickness data sets is not a trivial matter." I am not sure I understand this statement. Would it be possible to develop this idea a bit more?

P6L179 "the former [...] experiment." The first part of the sentence is unnecessary in my opinion. I would also recommend avoiding using the word "domain" for the the SIC product, as it usually refers to the study region.

P6L184 "By doing so [...] in atmospheric models." This passage is very confusing, I do not understand the point the authors are trying to make. They should either rewrite it, if they think it is important, or remove it.

P7L189 "WW3 is a standalone wave model...": In this case, it is indeed used in standalone mode, but WW3 can be coupled.

P7L190 "numerical stability is unaffected": By what?

P7L195 Section 3
I am not particularly convinced by the major interest of this section, and particularly by the interest of comparing pictures taken from the boat to the sea ice concentration products in section 3.1. Which angle does the pictures cover? Which surface area are they representative of? As the authors say, sea ice tends to cluster in the MIZ, and the fact that the sea ice concentration is not uniformly distributed spatially is well known by anyone who has had the opportunity to go in sea ice covered places. As I understand it, these observations motivated the study, but to me the interest of this paper does not lie here, and I actually think the removal of section 3.1 could potentially improve the clarity of the paper. If the authors want to keep it, they must make clearer the novelty of these observations and their interest. Moreover, I find the writing style very confusing in section 3.1. Section 3.2 is more convincing and clearer, but it is hard to see any novelty in it. It could maybe bring more to the study by linking it more closely to the results of section 4 and 5.

P7L197 I don't understand the use of "respectively" here.

P9L270 : "... the sea ice cut-off criterion is not clear in the documentation." This is not a very satisfying statement. Have the authors considered contacting the people in charge of ArcMFC to get more information about this criterion?

P9L275 "... but the Piper [...] did not reflect the sea state.": It is quite confusing, please rephrase.

P9L283 "Furthermore [...] an important role." This is very noticeable indeed. It would be very interesting to give an estimate of the spatial attenuation coefficient at the ice edge assuming an exponential wave attenuation, in order to show how it compares with the models and other reported observations (for instance Kohout et al., 2015)

P11L343 I don't think that one can write that the MIZ is aligned with the wind. The MIZ is an area. Maybe the authors could substitute MIZ by "the ice edge".

P12L352 "The figure only comprising...": I don't understand this sentence. What does "highly forced waves" mean?

P12L362 "this can occur": What is "this"?

P12L365 "Here, ..." I don't understand this sentence either, please consider rephrasing it.

P12L371 "The off-ice [...] of Appendix D." These two paragraphs are very confusing in my opinion, mostly because they are not well structured. It makes it very hard for the reader to understand the problem the authors are trying to address. They should be entirely rewritten.

P13L405 "Three principal parameters that form the sea ice forcing are": This formulation is misleading. I would instead say: "The three main parameters used to tune the wave-in-ice attenuation in the IC2,IC3 and IC5 parameterizations are"

P14L421 "The values here [...] the adopted default source term parameters." I am not sure I understand this sentence, it should be rewritten.

P14L422 "Our analysis demonstrates..." This statement should be at least discussed a bit more. For instance, the authors have used a limited number of wave-in-ice attenuation parameterizations, and none of them represent the wave scattering. Also, could these results change in a MIZ made of large floes and thicker ice for example? In addition, the authors have assumed a constant sea ice thickess of 10cm, and it is known that the behaviour of attenuation processes can change significantly depending on the sea ice thickness (see for instance Boutin et al., 2018). The sensitivity of these results to the sea ice thickness should be explored and discussed, for example by setting it to 20/30cm instead of 10cm.

P14L442 The conclusion is, in my opinion, much longer than it should be. I think it would have more impact if the results were synthesized in a few sentences only, and if it was ending with a discussion on the perspestives and the consequences of the findings presented here.

P14L443 "Reliable modelling [...] melt the Arctic Ocean sea ice." I don't really see the cause/consequence link in this sentence.

P16L490 "Reliable shipboard [...] was slow." I find the formulations used in these sentences a bit ambiguous. For instance, what do the authors mean by "seemingly sensible"? I think I get the idea, but it is not very clearly expressed. It is also not clear to me what is validated in the first sentence.

P25 Figure 4 "showing considerable uncertainty": This comment should be in the main text, not in the caption. The font size of the legend is also too small.

**Technical corrections :**

General : I would recommend using a roman text font for the subscritpts that are made of more than one character in the equations (low in f_low) for instance (If you are using latex, it means that you should add "\rm" or "\mathrm{my_subscript}" in your equations). It would improve the readability of the paper.

P1L21 "encountering high winds...": encounters is already the verb of the sentence, so no need to repeat.

P6L174 "Sea thickness" ->"Sea ice thickness"

P14L440 "have"->"has"

P15L478: ":microwave..." -> ":a microwave..."

P15L480: "a variant but similar device of Kohout et al..." -> "a device similar to the one used by Kohout et al. (2015)"

**References:**

Herman, A., 2015. Discrete-Element bonded particle Sea Ice model DESIgn, version 1.3 – model description and implementation. Geoscientific Model Development Discussions 8, 5481–5533. https://doi.org/10.5194/gmdd-8-5481-2015

Kohout, A.L., Williams, M.J.M., Toyota, T., Lieser, J., Hutchings, J., 2015. In situ observations of wave-induced sea ice breakup. Deep Sea Research Part II: Topical Studies in Oceanography. https://doi.org/10.1016/j.dsr2.2015.06.010

Montiel, F., Squire, V. A., Doble, M., Thomson, J., & Wadhams, P. ( 2018). Attenuation and directional spreading of ocean waves during a storm event in the autumn Beaufort Sea marginal ice zone. *Journal of Geophysical Research: Oceans*, 123, 5912– 5932. https://doi.org/10.1029/2018JC013763

---

## Referee Comment (RC2) · Anonymous Referee #2 · 16 Jan 2020

The manuscript examines the impact of observational uncertainty in sea ice concentration on numerical modelling of ocean surface waves in sea ice, using Wavewatch III. The most interesting finding, in my opinion, is that the effect of this uncertainty on wave modelling is large, and larger than the differences between various wave-ice interaction source terms currently included in WW3.

In general, I find the results to be well supported by thorough analysis. The manuscript is a little hard to follow in places (some instances of this noted below in 'presentational comments'). As currently presented, my sense is that this work is of interest to a narrow section of the community (i.e. those working on spectral models of wave-ice in-

teractions) and may be better suited to a more targeted journal. However, this could be addressed by revising the text to stress the aspects that could be interesting to a more general audience e.g. the differences between SIC products and their accuracy near the ice edge; that the sea ice coverage may be more important in determining ocean surface waves in ice than a detailed understanding of wave-ice interaction physics; the level to which waves in the open ocean are influenced by sea ice.

Major comments

'Analysis of significant wave height uncertainty distributions for SIC forcing and wave-ice interaction source terms reveals that they are both sizeable; however, the study concludes the more dominant uncertainty source of modelling wave-ice interactions is the accuracy of satellite retrieved SIC estimates that are used as model forcing.' The authors need to clarify that uncertainty in wave-ice interaction terms is an estimate based only on the various wave-ice interaction terms included in WW3, which as the authors note later in the manuscript are based on a limited number of field observations. It does not sample all possible wave-ice interaction terms, nor does it sample different parameter values.

It is well-known that SIC satellite observations underestimate concentrations for low thicknesses, less than 35 cm (Ivanova et al. 2015). This should be stated in the Introduction. I think the newer result here is how varied the satellite products are for these low thicknesses. This should be made clearer in the text.

Sec 2.3 This section needs more explanation to make it suitable for a more general audience - i.e. those not familiar with WW3 wave-ice interaction terms. What is the IS0 switch?

What is the estimated of spatial domain of the sea-truth concentration observations? Are these comparable to the resolution of the satellite products?

Sec 3.2 - why are only three products considered here? Conclusions on which products

perform best (or worst) for this region would be useful and of broader interest to the cryospheric community.

Minor comments

L22 'which are greater in the Arctic regions.' As compared to what?

L23 'sustainable developments of the Arctic Ocean' - this statement sounds strange

L28 'dispersion relation' -> 'the dispersion relation'

L32 Other recent work on wave-ice interactions could be cited here: Zhang et al. 2019 (https://doi.org/10.1016/j.ocemod.2019.101532), Roach et al. 2019 (https://doi.org/10.1029/2019MS001836)

L37 'Besides the model interior' What does this mean?

L37 'like SIC' -> 'in products such as SIC'?

L69 'generally' -> 'generally between'

L70 'Albeit the MIZ coverage..' - this sentence is unclear

L82 'mostly same' -> 'mostly the same'

L96 remove 'exhaustive' (this is an opinion)

L102 'most frequent' -> 'most frequently'

L102 'leading algorithms' leading in what? Perhaps you mean the most commonly used?

L152 'sanity checked' inappropriate language for a publication

L153 'leading packages' again, leading in what way? Most commonly used? Most skill?

L211 change to 'SSTs exceeded 0oC'

[Figure]

L211 At what depth were the shipboard SSTs measured? Are they affected by the ship itself?

L212 No s after MIZ

L218 'forseeable' -> 'visible'

L223 'The sea ice tends to cluster, so it appears dense where the ice exists.' - reword

L230 'depict the' -> 'show that the' ?

L264. What does 'best practice wave models' mean? Perhaps replace with 'commonly-used'?

L266 'a bit over' - informal language

L277 'as uncertainty generally increased in the MIZs.' - unclear. No s after MIZ.

L299 remove the rest of the sentence after 'ongoing work' - not necessary.

L323 add 'that' - 'demonstrated that'

L338 'like a' -> 'as a'

L339 'effects. . .are' (not is)

L443 Reword

---

## Author Comment (AC1) · 25 Feb 2020

**Author response to tc-2019-285-RC1**

Takehiko Nose on behalf of all authors

We sincerely appreciate the critique and insightful comments that have improved this manuscript. We are humbled by how succinct the revised manuscript is compared with the submitted version. We are grateful for the reviewer's attention to detail. Below outlines our response to each of the critical point raised by the reviewer. The document structure corresponds to that of tc-2019-285-RC1-supplement except for the first section.

**Overview of major changes**

We begin with an overview of major changes in the revised manuscript to address both reviewers' comment regarding readability of the paper. The change of section titles is shown in bold text (if changed).

**Abstract** The abstract has been rewritten to more clearly emphasise the focus of the study and its findings.

**Section 2.3** The description of WAVEWATCH III $^{®}$ (WW3) has been modified in a way that it is focused on the core idea of the paper: wave-ice interactions. The readability is improved by using equations and compact notations, thereby demonstrating clearly the link between sea ice concentration (SIC) and wave-ice models.

**Section 3 Sea ice concentration: definition, characteristics, and the use in wave-ice models**
The presentation of this section is one of the major changes in the revised version. We have rewritten this section to discuss the concept of SIC from a wave-ice modelling perspective. Despite different scales and objectives, the SIC is being used to moderate the attenuation rate of waves in the MIZ without the model having any consideration to the sea ice field heterogeneity, i.e., subgrid scale physics. We discuss the implication of the length scale of satellite retrieved SIC to the SIC and wave-ice model formulation.
Section 3.2 is shortened and moved to the beginning of Section 5 to give an overview of how satellite retrieved SIC varies on a regional scale so that the transition to the wave hindcast analysis of the region is more fluent.
Original Figures 3 and 4 are removed as they were deemed redundant.

**Section 4 $\Delta c_i$ effects on wave modelling at the observation sites**
Note $\Delta c_i$ is denoted as the SIC uncertainty in the text.

**Section 5 $\Delta c_i$ and wave modelling in the refreezing Chukchi Sea**

**Section 5.1 On- and off-ice wave evolution in the refreezing Chukchi Sea MIZs**

**Section 5.2 Relative significance of $\Delta c_i$ compared with wave-ice interaction parameterisation uncertainty**
We tested the robustness of the results based on sensitivity analyses that tested sea ice thickness (SIT) and the inclusion of scattering. The study findings remain unchanged.

**Section 6 Conclusions and discussions**
This section is shortened considerably and includes discussions on the study outcome.

**General comments**

**Reviewer comment R#1-1** This paper presents interesting results, confirming some statements about the sensitivity of wave-inice modelling to sea ice concentration made in previous studies. The fact that the uncertainty in the estimated sea ice concentration has a larger effect on the uncertainty of the wave height estimated in-ice than the change in the wave-in-ice attenuation parameterization is a nice finding that illustrates very well the difficulties faced by wave-in-ice modellers. However, the paper suffers from a writing style that is confusing in a number of places. This is particularly the case for sections 3, 4 and 5. It is a bit paradoxal, as sometimes the important information is hidden in a succession of very wordy sentences, making it hard for the reader to get the message, and sometimes it seems that the authors wanted to avoid repeating themselves whereas the reader would happily appreciate some help.

**Author response** We are pleased the essence of this manuscript was conveyed to the reviewer. To enhance the clarity of our message, efforts were made to improve the readability of the paper by following reviewer's suggestions. Please see the overview of major changes.

**Reviewer comment R#1-2** I am also not convinced by the usefulness and the novelty of section 3, at least in its current form. To me, the most interesting part of the study lies in section 5, but it is overshadowed by the lack of clarity of the previous sections. It would be worth shifting the emphasis of the paper to this section more quickly.

**Author response** We appreciate the critical comment regarding the Section 3 content. Section 3 now reads as below.

**Author's changes in manuscript** WMO [2014] defines SIC as "the ratio expressed in tenths describing the amount of the sea surface covered by ice as a fraction of the whole area being considered". The so-called "area considered" presumably varies for different objectives. Length scale of $O(10)$ km may be adequate for sea ice extent climatology, but for wave-ice interactions, the wave provides a scale in a phase-resolved sense. Satellite derived SIC represents the fraction of ice-covered water over a large area, sufficiently large enough that the SIC represents a property of a continuum. In reality, the sea ice in the marginal ice zone (MIZ) is granular, and ice floes jam due to horizontal convergence by Langmuir circulation, internal waves, and wind variability, resulting in a formation of features such as ice bands and wind streaks—with which waves likely interact distinctively.

On 14 November 2018 during the MIZ transect observation, R/V Mirai encountered moderate on-ice waves with an $H_{m0}$ up to around 2.00 m propagating towards the ice edge (this $H_{m0}$ estimate is consistent from both the shipboard wave data described in Appendix A and hindcast models as discussed later). Figure 2 presents a series of snapshot images of the sea ice field during the encounter. R/V Mirai traversed over 10 km in the MIZ from the ice edge, and each image area extends at least over 1 km conservatively (using the crude distance to horizon calculation). These images depict the heterogeneous sea ice field, both in distribution and ice types, that waves propagate when they enter an MIZ. Because WW3 wave-ice interaction models are scaled according to $\frac{dN}{dt} = c_i s_{\text{ice}}$ (Equation 4), the subgrid scale physics is completely missing. It is plausible the subgrid scale distribution of SIC and ice types can be treated in a stochastic manner to provide meaningful mean values to the grid-scale model. On the other hand, $c_i$ also affects the WW3 wave-ice model by means of scaling (Equation 4). Figure 2 shows SIC data from eight satellite retrieved products described in Section 2.2 during the event. The SIC estimates interpolated at the R/V Mirai positions largely deviate among the products, characterising the uncertainty of the satellite retrieved SIC. Moreover, the entire time series of the MIZ transect observation depicts $\Delta c_i$ is persistent (Figures A4 to A6 of Appendix D). Hereafter, we show how large the effect of $\Delta c_i$ on modelling MIZ waves can be, so much so that it overwhelms the choice of $s_{ice}$, e.g., ICX.

**Specific comments**

**Reviewer comment R#1-3 :** **P1L22.** "model uncertainties...", which models are the authors talking about here?

**Author response** We made a change to this passage (bold text) as below.

**Author's changes in manuscript** . . . From a practical view point, this downward trend of sea ice decline opens trans-Arctic shipping routes connecting Europe and Asia for longer times of the year; potential global economic benefits of non-ice breakers accessing routes like Northern Sea Route and North West Passage are substantial [Stephenson et al., 2013, Bekkers et al., 2018]. **The increasing vessel traffic implies that adequate prediction capabilities will become crucial to assist ships in polar waters to circumnavigate hazards such as high winds and waves, collision with perennial sea ice, and sea-spray icing; however, Jung et al. [2016] describe that the existing polar prediction systems need to be urgently enhanced to effectively manage the risks and opportunities associated with growing human activities, and the Polar Prediction Project (PPP) has contributed to advancing the predictive capabilities. While wave forecasting in polar oceans is still in its early years, the need for advancing wave forecast capacity will only grow in the emerging Arctic Ocean.** This paper focuses on the effect of SIC uncertainty on third-generation spectral wave model simulations in and near a MIZ.

**Reviewer comment R#1-4 :** **P2L25.** The authors should consider giving a definition of the MIZ.

**Author response** The following text is added.

**Author's changes in manuscript** WMO [2014] defines the MIZ as "the region of an ice cover which is affected by waves and swell penetrating into the ice from the open ocean". This study is primarily focused on the MIZ region near the open ocean.

**Reviewer comment R#1-5 :** **P2L36.** "wave-ice interaction source term": the authors haven't introduced the principle of spectral wave models yet, so it is not clear for everyone what the wave-ice interaction source term is.

**Author response** Changed the object from "the wave-ice interaction source term" to "the wave-ice interaction parameterisation".

**Reviewer comment R#1-6 :** **P3L64.** Here the authors introduce the methods used for the measurements on board on the R/V Mirai. But the way the so called "sea truth" images were taken is only described in section 3.1 (P7L212), lost in comments on the results. Similarly, before the end of section 3.1, a definition of how the uncertainty of a given quantity is computed is suddenly introduced, without even a proper transition (P8L225). This mix between comments on the results and details of the methods makes section 3.1 very confusing and much longer than it should be.

**Author response** Please see author's response to R#1-2. Regarding the uncertainty definition, it is moved to Section 2.2 and introduced where the eight SIC products are described.

**Reviewer comment R#1-7 :** **P4L88.** I think the word "translated" is not appropriate here (and in some other places). The authors could consider using "interpreted", "inferred".

**Author response** SIC is a variable that is calculated from brightness temperatures, so we would prefer to have a quantitative phrase rather than being qualitative. In that sense, "calculate" is more appropriate here. Regarding the use of "translate" in other places, we have replaced them with more straightforward expressions.

**Author's changes in manuscript** SIC estimates from Earth-orbiting satellites are an indirect measurement calculated from microwave brightness temperatures.

**Reviewer comment R#1-8 :** **P4L118.** The sentences about the different grids the authors could

have used and the one they are actually using are really confusing. Maybe they should try to cut them into more but shorter sentences, each dealing with one region and one resolution.

**Author response** Changed to a shorter sentence as suggested.

**Author's changes in manuscript** ARTIST-Sea-Ice—this algorithm uses 89 GHz frequency signal to produce high resolution SIC estimates. This algorithm was selected as accurate higher resolution forcing is generally desirable for numerical models. For this product, we only use the AMSR2 data but analyse two different grids: the pan-Arctic data with 6.250 km resolution and the regional Chukchi-Beaufort data with 3.125 km grid resolution.

**Reviewer comment R#1-9 : P5L143.** "A curvilinear grid [...] sea point cells." I found this whole paragraph very confusing. As a reader I found very difficult to understand the links between each sentence, and the expression "The grid" seems to be applied to different things. As an example, they first refer to the model "geographical" grid, then to the spectral grid, then they use again the expression "the grid" to give details about the bathymetry. A quick reminder of the region they are focusing on would also be welcome, especially as they refer to the "other seas" at the border of "the domain".

**Author response** Adding appropriate modifiers to "grid" was necessary, and the spectral grid description was mixed up in the regional grid description, which was confusing. The paragraph is clarified and shortened as follows.

**Author's changes in manuscript** The TodaiWW3-ArCS used in this study has a horizontal resolution of 4 km, and its domain covers most of the Pacific side of the Arctic Ocean including the East Siberian, Chukchi, and Beaufort Seas. The model boundaries connected to the seas of the Arctic Ocean was enclosed by ice cover during the November 2018 modelling period (corresponding to the R/V Mirai observation), so nesting was unnecessary. Similar to Rogers et al. [2016], we neglected swell penetration through the Bering Strait. The technical details of TodaiWW3-ArCS 's geographical and spectral grids are provided in Appendix B.

**Reviewer comment R#1-10 : P6L152.** "During the version upgrade of TodaiWW3-ArCS...": which version upgrade are the authors referring to? Are they sure it is relevant for the paper? I think the authors could just state that they are using the ST6 parameterization for the non-ice source terms as it was previously shown to give the best results for the case discussed in Nose et al. (2018) with the model being forced by ERA5 winds.

**Author response** Corrected as suggested. We have rephrased the sentence to make this paragraph clearer.

**Author's changes in manuscript** $s_{\text{wind}}$ and $s_{\text{dissipation}}$ parameterisations and wind forcing for The Arctic Ocean wave model developed at the University of Tokyo (TodaiWW3-ArCS) were tested. We compared the most commonly used physics packages, ST4 [Ardhuin et al., 2010, Rascle and Ardhuin, 2013] and ST6 [Rogers et al., 2012, Zieger et al., 2015, Liu et al., 2019], using European Centre for Medium-Range Weather Forecasts (ECMWF) global reanalysis (ERA5) 10 m wind ($U_{10}$) against the 2016 September storm [Nose et al., 2018] when TodaiWW3-ArCS and observations agreed well. The ST6 parameterisation showed marginally improved agreement using the default parameters; so all simulations used the ST6 parameterisation and were forced with ERA5 wind fields. The default $s_{\text{non-linear interactions}}$, which is not affected numerically by sea ice, was used for all simulations.

**Reviewer comment R#1-11 : P6L157.** "The s_ice term is composed": I think the use of composed is misleading here. The attenuation terms the authors mention are included in different parameterizations (ISX, ICX), and they are not all compatible with each other. I would suggest an expression like "The s_ice term represents wave-in-ice attenuation processes such as..." for instance.

**Author response** The passage relating to the WW3 wave-ice interaction term is one of the major changes. This now passage reads as follows.

**Author's changes in manuscript** . . . The sum of these source terms $s$ is expressed based on the

following default scaling in ice-covered waters:

$$s = (1 - c_i)(s_{\text{wind}} + s_{\text{dissipation}}) + c_i s_{\text{ice}} + s_{\text{non-linear interactions}}.$$

Specifically to this study, $c_i$ relates to the satellite retrieved SIC and $s_{\text{ice}}$ to the ice type, i.e., how the model treats sea ice. The effect of sea ice on waves are represented via the modified dispersion relation $\sigma = \sigma(\overline{k})$ where $|\overline{k}| = k = k_r + i k_i$. The real part $k_r$ is the physical wavenumber and alters the propagation speed of waves in a sea ice field (analogous to effects of shoaling and refraction by bathymetry), and the imaginary part $k_i$ is the exponential decay coefficient. $k_i$ is introduced in the model as $s_{\text{ice}} = -2 c_g k_i N$ for fully ice-covered sea, i.e., $c_i{=}1$, and the solution to $\frac{dN}{dt} = s_{\text{ice}}$ is $N_0 e^{-2c_g k_i t}$. There are five options for treating sea ice in WW3 denoted as IC1–5; $c_i$ provides the scaling in the linkage between $s_{\text{ice}}$ and ICX as

$$\frac{dN}{dt} = c_i s_{\text{ice}} = -2 c_i c_g k_i(f, p_1, ..., p_n) N$$

where $p_1, ..., p_n$ are the sea ice properties, e.g., effective shear modulus and effective viscosity. Therefore, the rate of attenuation depends on the wave period and sea ice properties, which is moderated by $c_i$, i.e., $N_0 e^{-2 c_i c_g k_i t}$.

The wave-ice models implemented in WW3 that calculate $k_r$ to model $k_i$ are as follows: IC2 calculates dissipation due to basal friction in the boundary layer below an ice sheet, which is modelled as a continuous thin elastic plate based on the work of Liu and Mollo-Christensen [1988]; IC3 treats sea ice as a visco-elastic layer based on Wang and Shen [2010], which calculates the internal stress of the ice cover based on storage and dissipation; and IC5 is a visco-elastic beam model based on Mosig et al. [2015]. The dispersion relation of these models are provided in Appendix B. Ardhuin et al. [2018], Boutin et al. [2018] (IC2) and Rogers et al. [2016], Cheng et al. [2017] (IC3) describe the progress of these $s_{ice}$ parameterisations using the refreezing Beaufort Sea data of Thomson et al. [2018]. These wave-ice models can be combined with an energy-conservative scattering attenuation model denoted as IS1 and IS2 [Meylan and Masson, 2006, Dumont et al., 2011, Williams et al., 2013, Ardhuin et al., 2018, Boutin et al., 2018].

**Reviewer comment R#1-12: P6L160.** "The dominant floe size [...] IS0 switch.": Here I have the feeling that the authors want to justify why they did not include scattering terms in their wave-in-ice source terms. I think this justification is very long and with unnecessary information (the way scattering terms work in WW3 for instance). I think it would be much clearer simply stating that during the cruise, sea ice in the MIZ was mainly made of grease, nilas and pancake ice, for which scattering is not expected to be the dominant process (Montiel et al., 2018), and therefore scattering was not considered here. Specifying the WW3 switch IS0 is also unnecessary.

**Author response** Thank you for this suggestion. The suggested text is succinct and used in the manuscript.

**Author's changes in manuscript** During the cruise, sea ice in the MIZ was mainly grease, nilas, and pancake ice, so the hindcast experiment was conducted using the IC3 package with the default parameters as it has been designed for these ice types [Rogers et al., 2016, Cheng et al., 2017]. Scattering is not expected to be the dominant process in this type of ice fields [Montiel et al., 2018], so it was not considered in the experiment.

**Reviewer comment R#1-13: P6L170.** "The underlying principle of sea ice models is that sea ice is treated a continuum." Firstly, there is a small typo, it should be "treated as...". Secondly, this statement might be true for the sea ice models used in climate models, but it ignores discrete elements sea ice models, often used for sea ice-structure interactions. They can also be used to study wave-ice interactions (Herman et al., 2015). Actually I think the sentences between P6L170 and P6L173 could be shortened. The reason the authors choose this parameterization is because it has been developed to represent similar ice conditions to the ones encountered by the R/V Mirai, which is not the case for the other parameterizations.

**Author response** We have incorporated the reviewer's comment. Please see the first sentence of author's response to R#1-12.

**Reviewer comment R#1-14: P6L175.** "... the treatment of independent SIC and sea ice thickness data sets is not a trivial matter." I am not sure I understand this statement. Would it be possible to develop this idea a bit more?

**Author response** It is much clearer just to say the constant thickness was applied, so we can evaluate solely the SIC uncertainty on wave modelling. The text is modified as below.

**Author's changes in manuscript** Regarding sea ice thickness forcing, a homogeneous input option with a value of 10 cm was applied; the constant forcing was applied so we can evaluate solely the $\Delta$SIC effect on wave-ice interaction parameterisations. 10 cm was initially chosen because the MIZ transect observation was mostly characterised by new and young ice whose upper bound of sea ice thickness is of a similar order [Canadian Ice Service-Environment Canada, 2005].

**Reviewer comment R#1-15: P6L179.** "the former [...] experiment." The first part of the sentence is unnecessary in my opinion. I would also recommend avoiding using the word "domain" for the the SIC product, as it usually refers to the study region.

**Author response** Corrected as suggested.

**Author's changes in manuscript** ASI-3km and OSISAF-AMSR2 data were excluded for the wave hindcast experiment. The former has a regional coverage that is too small for the TodaiWW3-ArCS domain, and the OSISAF-AMSR2 data have noise in the open ocean, which yield erroneous wave simulation results when they are used as model forcing (as described in Appendix C).

**Reviewer comment R#1-16: P6L184 .** "By doing so [...] in atmospheric models." This passage is very confusing, I do not understand the point the authors are trying to make. They should either rewrite it, if they think it is important, or remove it.

**Author response** The passage is rewritten as below.

**Author's changes in manuscript** It should be noted that when satellite derived SIC data are used as forcing, the heat and momentum fluxes are distorted in the marine atmospheric boundary layer because the lower atmosphere and the ocean surface are no longer coupled. Inoue et al. [2011] evaluated surface heat transfer from three reanalysis products by focusing on how the models treat sea ice; they found the accuracy of SIC is a key variable for estimating surface turbulent heat fluxes. Guest et al. [2018] have elucidated the ice-edge jet generation mechanism based on the in situ data obtained in the refreezing Beaufort Sea. Undoubtedly, altering the sea ice field would feedback to the wind, but this is not captured in the wave hindcast experiment.

**Reviewer comment R#1-17: P7L189.** "WW3 is a standalone wave model...": In this case, it is indeed used in standalone mode, but WW3 can be coupled.

**Author response** This is true the modifier "standalone" was unnecessary. Further, the entire sentence turned out to be superfluous when we addressed R#1-16, and as such, the sentence is removed.

**Reviewer comment R#1-18: P7L190.** "numerical stability is unaffected": By what?

**Author response** This sentence is removed. Please see author's responses to R#-16 and R#-17.

**Reviewer comment R#1-19: P7L195, Section 3.** I am not particularly convinced by the major interest of this section, and particularly by the interest of comparing pictures taken from the boat to the sea ice concentration products in section 3.1. Which angle does the pictures cover? Which surface area are they representative of? As the authors say, sea ice tends to cluster in the MIZ, and the fact that the sea ice concentration is not uniformly distributed spatially is well known by anyone who has had the opportunity to go in sea ice covered places. As I understand it, these observations motivated the study, but to me the interest of this paper does not lie here, and I actually think the removal of

section 3.1 could potentially improve the clarity of the paper. If the authors want to keep it, they must make clearer the novelty of these observations and their interest. Moreover, I find the writing style very confusing in section 3.1. Section 3.2 is more convincing and clearer, but it is hard to see any novelty in it. It could maybe bring more to the study by linking it more closely to the results of section 4 and 5.

**Author response** Please see the overview of major changes and author's response to R#1-2.

**Reviewer comment R#1-20: P7L197.** I don't understand the use of "respectively" here.

**Author response** This passage is moved to an appendix, and the "respectively" is removed there.

**Reviewer comment R#1-21: P9L270.** "... the sea ice cut-off criterion is not clear in the documentation." This is not a very satisfying statement. Have the authors considered contacting the people in charge of ArcMFC to get more information about this criterion?

**Author response** Thank you for this comment: the statement was unsatisfactory. We found out that the wave-ice interaction of Sutherland et al. [2019] was implemented in December, 2019: the month after our Arctic Ocean observation. We still find value in reporting the ARCMFC model data in our results, but we have amended the ARCMFC model description.

**Reviewer comment R#1-22: P9L275.** "... but the Piper [...] did not reflect the sea state.": It is quite confusing, please rephrase.

**Author response** Waves in the Chukchi Sea during the study period was dominated by wind seas, which have shorter wavelength relative to the ship dimensions. These short waves are impeded by R/V Mirai's hull, so the shipboard Piper#15 has limitations on measuring wind seas. As such, most of the Piper#15 data did not reflect the true wave field. We have rephrased the paragraph, and the explanation of this point is moved to an appendix. So it is removed from the main text. Please refer to author's changes in manuscript to R#1-34 for how we addressed this comment.

**Reviewer comment R#1-23: P9L283.** "Furthermore [...] an important role." This is very noticeable indeed. It would be very interesting to give an estimate of the spatial attenuation coefficient at the ice edge assuming an exponential wave attenuation, in order to show how it compares with the models and other reported observations (for instance Kohout et al., 2015).

**Author response** Thank you for the comment. We realised we used incorrect expressions to describe the wave height uncertainty earlier in this paragraph, which likely led the reviewer to inquire about the attenuation rates. In P9L278 of the original manuscript, we stated "waves decay with varying attenuation rates"; however, this should have been "waves decay at different timing depending on the sea ice edge location of the respective SIC forcing used". The whole point of the experiment is testing different SIC forcing using the consistent wave-ice interaction source term setting, i.e., the same attenuation coefficient. We apologise for the misleading sentence.

We are also mindful that a successful study of wave attenuation rates depends strictly on the knowledge of the ice edge or availability of two and more buoys along the fetch. As such, we are unable to repeat the novel analysis of Kohout et al. [2016] because we did not have enough measurements along the fetch.

**Reviewer comment R#1-24: P11L343.** I don't think that one can write that the MIZ is aligned with the wind. The MIZ is an area. Maybe the authors could substitute MIZ by "the ice edge".

**Author response** The sentence is corrected by replacing "MIZs" with "ice edges".

**Reviewer comment R#1-25: P12L352.** "The figure only comprising...": I don't understand this sentence. What does "highly forced waves" mean?

**Author response** Highly forced waves imply a wave field, specifically wind seas, that is rapidly

growing under the wind forcing. The sentence is rewritten to clarify the point.

**Author's changes in manuscript** The figure shows only blue and light-blue markers, which indicate the waves generated by the strong localised wind decayed with limited wave penetration no farther than $mean(c_i) = 0.40$.

**Reviewer comment R#1-26: P12L362.** "this can occur": What is "this"?

**Author response** "this" is specified.

**Author's changes in manuscript** . . . and although the enhanced scatter plot disparages data with low $mean(c_i)$, high $\Delta H_{m0}$ with low $mean(c_i)$ does exist. Analogous to the on-ice wave case, high $\Delta H_{m0}$ can occur near . . .

**Reviewer comment R#1-27: P12L365.** "Here, ..." I don't understand this sentence either, please consider rephrasing it.

**Author response** In this sentence, "highly forced" was unnecessary and is removed. The subject of the sentence is also clarified instead of using "Here".

**Author's changes in manuscript** Along this transect, ASI-6km and BST-AMSR2 have the most north east ice edge, and the waves rapidly grow under the strong north easterly wind forcing whereas . . .

**Reviewer comment R#1-28: P12L371.** "The off-ice [...] of Appendix D." These two paragraphs are very confusing in my opinion, mostly because they are not well structured. It makes it very hard for the reader to understand the problem the authors are trying to address. They should be entirely rewritten.

**Author response** The repeat introduction of the equation in both paragraphs made them seem unstructured. We also made the second paragraph more succinct. The two paragraphs are modified as below.

**Author's changes in manuscript** Off-ice wave evolution is a complex process because the fetch is not only controlled by the location of the ice edge, but also wave-ice interactions as implemented in WW3. The current numerical approach to simulate wind pumping energy into waves in ice cover is dictated by $c_i$ because waves grow when $(1 - c_i)(s_{\text{wind}} + s_{\text{dissipation}}) > c_i s_{\text{ice}}$. Whether wave evolution in ice cover follows the Equation 3 scaling has been discussed in Rogers et al. [2016], Thomson et al. [2018]; the latter cites Li et al. [2017] who confirmed wind input to high frequency wave energy in the Antarctic Ocean. The off-ice $\Delta H_{m0}$ is apparently also influenced by the cumulative effect of the $\Delta c_i$ along the fetch distance affected by the wave-ice interactions as implemented in WW3.
Lastly, for both on- and off-ice wave cases, significant $\Delta H_{m0}$ extends to the waters where the wind forcing is orientated along the ice edge; so the model data are briefly examined in the region of MIZs north east of Wrangel Island, which is shown as Quadrilateral 1 in Figure 8a along the sea ice edge and north easterly wind forcing orientation. This region has considerable $\Delta c_i$ (not shown here) in a similar manner to Figure 5, and the model $\Delta H_{m0}$ is just as sizeable under the influence of high wind forcing. There is evidence of correlated bivariate uncertainty data in Figure 8b, and a combination of on- and off-ice wave features for the respective enhanced plots discussed in the previous paragraphs are depicted. Deciphering the physical processes is complicated; however, the bivariate uncertainty data along a transect illustrates how $\Delta c_i$ and $\Delta H_{m0}$ are related; Figure 9b shows these results for a cross section oriented along the ice edge (the long axis of Quadrilateral 2) on 21 November 2018 18:00.

**Reviewer comment R#1-29: P13L405.** "Three principal parameters that form the sea ice forcing are": This formulation is misleading. I would instead say: "The three main parameters used to tune the wave-in-ice attenuation in the IC2,IC3 and IC5 parameterizations are"

**Author response** The sentence is changed as suggested.

**Reviewer comment R#1-30: P14L421.** "The values here [...] the adopted default source term

parameters." I am not sure I understand this sentence, it should be rewritten.

**Author response** This sentence was unnecessary and is removed.

**Reviewer comment R#1-31: P14L422 .** "Our analysis demonstrates..." This statement should be at least discussed a bit more. For instance, the authors have used a limited number of wave-in-ice attenuation parameterizations, and none of them represent the wave scattering. Also, could these results change in a MIZ made of large floes and thicker ice for example? In addition, the authors have assumed a constant sea ice thickness of 10cm, and it is known that the behaviour of attenuation processes can change significantly depending on the sea ice thickness (see for instance Boutin et al., 2018). The sensitivity of these results to the sea ice thickness should be explored and discussed, for example by setting it to 20/30cm instead of 10cm.

**Author response** Agreed that further discussion is warranted. We tested robustness of the study finding by checking sensitivity to SIT and scattering. The modified text is as follows.

**Author's changes in manuscript** . . . Uncertainty distributions are visualised in a Q-Q plot by simply sorting each dataset, and this is shown in Figure 10. The figure depicts that both uncertainties are considerable with $max(\Delta H_{m0})$ values of 1.95 m and 1.44 m for the $\Delta c_i$ and $s_{ice}$ uncertainty experiments, respectively. The robustness of this result was examined via SIT forcing sensitivity analysis. From a physical view point, the choice of 10 cm was made to match the observed sea ice types during the R/V Mirai MIZ transect observation. For observational evidence of SIT in the refreezing Arctic Ocean, we defer to Ardhuin et al. [2018] to determine the test case and selected 50 cm. From a wave-ice modelling perspective, SIT effectively serves as a tuning parameter when forced as a homogeneous field. For example, the attenuation rate $k_i$ of IC2 as shown in Appendix B has SIT in the form of $(1 + k_r M)$ in the denominator: $M = \frac{\rho h_i}{\rho_w}$ [Liu and Mollo-Christensen, 1988] where $h_i$ is the SIT, and $\rho$ and $\rho_w$ are the ice and sea water density. If we take a deep water wavelength corresponding to 7 s wave period, changing the SIT from 10 cm to 50 cm increase the $k_i$ by at most 3 %. $k_i$ sensitivity on SIT examined in Wang and Shen [2010], Mosig et al. [2015] (IC3 and IC5) appears more sensitive; as such, sensitivity analysis was conducted for our model. Repeating the $\Delta c_i$ and $s_{ice}$ uncertainty experiments using 50 cm SIT, $max(\Delta H_{m0})$ values increased respectively to 2.34 m and 1.95 m, but the $\Delta c_i$ remains as the dominant error source. Further, IC3 was most affected by the SIT change for the equivalent transects of Figure 9 (not shown). Even though there was no event during the study period when scattering were expected to be the dominant process (the implication of this is given in Section 6), sensitivity of the finding to scattering was also examined by combining IS2 scattering with IC2 and IC3 with the default parameters. The results remained robust. Lastly, sensitivity to the choice of SIC data used in the $s_{ice}$ uncertainty experiment was examined by using ASI-6km instead of BST-AMSR2: the experiment also resulted in the same outcome.

**Reviewer comment R#1-32: P14L422 .** The conclusion is, in my opinion, much longer than it should be. I think it would have more impact if the results were synthesized in a few sentences only, and if it was ending with a discussion on the perspestives and the consequences of the findings presented here.

**Author response** The conclusion section has been made more succinct and reads as follows.

**Author's changes in manuscript** The WW3 wave-ice models represent the exponential decay of waves in the presence of sea ice as $\frac{dN}{dt} = c_i s_{\text{ice}} = -2c_i c_g k_i(f, p_1, ..., p_n)N$ (Equation 4). We investigated the effect of the satellite derived SIC uncertainty $\Delta c_i$ on modelling waves in the refreezing Chukchi Sea MIZ using six SIC data sets based on the four commonly used retrieval algorithms: NASA-Team, Bootstrap, OSISAF, and ARTIST-sea-ice. The wave hindcast experiment reveals $\Delta c_i$ causes model wave height uncertainty $\Delta H_{m0}$, and there is evidence that bivariate uncertainty data ($\Delta H_{m0}$ and $\Delta c_i$) are correlated, although off-ice wave growth is more complicated due to the cumulative effect of $\Delta c_i$ along an MIZ fetch

We compared the $\Delta H_{m0}$ distribution of the $\Delta c_i$ experiment with that of the $s_{ice}$ uncertainty experiment. Both uncertainties are found to be considerable during the simulation period with maximum $\Delta H_{m0}$ values of 1.95 m and 1.44 m, respectively. This result is found to be robust based on the

sensitivity analyses that tested the SIT forcing and the inclusion of scattering. Despite the $s_{ice}$ parameterisations being derived from different concepts and the WW3 wave-ice models completely missing the subgrid scale physics relating to sea ice field heterogeneity, the accuracy of satellite retrieved SIC used as model forcing is the primary error source of modelling MIZ waves in the refreezing ocean. The study outcome suggests wave-ice model tuning may not be as effective at this time when the knowledge of the true SIC field is too uncertain. It is worthy to note that swell waves that propagate $O(100)$ km into the ice-covered water where the scattering would likely be the dominant process were not observed during the study period. As such, the effect of $\Delta c_i$ for such waves remains to be resolved. Future improvements on the wave-ice models should come from two ends; continual developments of parameterised physics on the regional and pan-Arctic scale and working on a subgrid scale physical model on the other end. Solid and robust observational evidence through remote sensing and shipboard measurements is likely the key to connecting these two ends.

**Reviewer comment R#1-33: P14L443.** "Reliable modelling [...] melt the Arctic Ocean sea ice." I don't really see the cause/consequence link in this sentence.

**Author response** The sentence is removed as it was unnecessary. See author's changes in manuscript to R#1-32.

**Reviewer comment R#1-34: P16L490.** "Reliable shipboard [...] was slow." I find the formulations used in these sentences a bit ambiguous. For instance, what do the authors mean by "seemingly sensible"? I think I get the idea, but it is not very clearly expressed. It is also not clear to me what is validated in the first sentence.

**Author response** How these data were used in the study is described with more clarity. We have rewritten this passage in Appendix A explaining how the shipboard data were used as below.

**Author's changes in manuscript** Wave observations during the campaign from the WM-2 integrated analog system [TSK Tsurumi Seiki Co., 2019] were significantly affected by Doppler correction errors. Collins III et al. [2015] have shown shipboard measurements are less affected by this effect when ship speed is $< 3$ ms$^{-1}$. Applying a 2 ms$^{-1}$ ship speed threshold greatly reduced conspicuously spurious data, and these data are used as indicative wave heights in this study (e.g., Figure 6). Piper#15 on board the vessel relies on an IMU. The processing method is consistent with Kohout et al. [2015] except 15 minute intervals were used instead of 1 hour. Waves in the Chukchi Sea during the study period was dominated by wind seas, which have shorter wavelength relative to the ship dimensions. These waves are impeded by R/V Mirai's hull, so the shipboard Piper#15 has limitations on measuring wind seas. Response Amplitude Operator of R/V Mirai and the WM-2 data can be combined in theory to transfer IMU's high frequency signals to true surface elevation signals, but post-processing remains ongoing work. Although most of the Piper#15 data did not reflect the true wave field, the peak Piper#15 $H_{m0}$ of 2.00 m during the on-ice wave event on 14 November 2018 agreed with the peak WM-2 $H_{m0}$; this value is also comparable with the ERA5 $H_{m0}$ as well. This provides confidence that the waves observed during this event was at least around 2.00 m.

**Reviewer comment R#1-35: P25 Figure 4.** "showing considerable uncertainty": This comment should be in the main text, not in the caption. The font size of the legend is also too small.

**Author response** The phrase pointed out by the reviewer is removed from the caption, and bigger legend text is inserted.

**Technical corrections**

**Reviewer comment R#1-36: General.** I would recommend using a roman text font for the subscripts that are made of more than one character in the equations (low in f_low) for instance (If you are using latex, it means that you should add "\rm" or "\mathrmmy_subscript" in your equations). It would improve the readability of the paper.

**Author response** Subscripts are changed to roman text as recommended.

**Reviewer comment R#1-37: P1L21.** "encountering high winds...": encounters is already the verb of the sentence, so no need to repeat.

**Author response** Removed the verb phrase. Please see author's changes to manuscript in R#1-3.

**Reviewer comment R#1-38: P6L174.** "Sea thickness"→"Sea ice thickness"

**Author response** Corrected.

**Reviewer comment R#1-39: P14L440.** "have"→"has".

**Author response** Corrected.

**Reviewer comment R#1-40: P15L478.** ":microwave..."→":a microwave...".

**Author response** Corrected.

**Reviewer comment R#1-41: P15L480.** "a variant but similar device of Kohout et al..."→"a device similar to the one used by Kohout et al. (2015)".

**Author response** Changed as suggested.

Thank you kindly for your review and consideration.

**References**

Fabrice Ardhuin, Erick Rogers, Alexander V. Babanin, Jean-François Filipot, Rudy Magne, Aaron Roland, Andre van der Westhuysen, Pierre Queffeulou, Jean-Michel Lefevre, Lotfi Aouf, and Fabrice Collard. Semiempirical dissipation source functions for ocean waves. Part I: definition, calibration, and validation. *Journal of Physical Oceanography*, 40(9):1917–1941, 2010. `https://doi.org/10.1175/2010JP04324.1`.

Fabrice Ardhuin, Guillaume Boutin, Justin Stopa, Fanny Girard-Ardhuin, Christian Melsheimer, Jim Thomson, Alison Kohout, Martin Doble, and Peter Wadhams. Wave attenuation through an Arctic marginal ice zone on October 12, 2015: Part 2. Numerical modeling of waves and associated ice break-up. *Journal of Geophysical Research: Oceans*, 2018. `https://doi.org/10.1002/2018JC013784`.

Eddy Bekkers, Joseph F. Francois, and Hugo Rojas-Romagosa. Melting ice caps and the economic impact of opening the northern sea route. *The Economic Journal*, 128(610):1095–1127, 2018. `https://doi.org/10.1111/ecoj.12460`.

Guillaume Boutin, Fabrice Ardhuin, Dany Dumont, Caroline Sevigny, Fanny Girard-Ardhuin, and Mickael Accensi. Floe size effect on wave-ice interactions: Possible effects, implementation in wave model, and evaluation. *Journal of Geophysical Research: Oceans*, 123(7):4779–4805, 2018. `https://doi.org/10.1029/2017JC013622`.

Canadian Ice Service-Environment Canada. Manual of ice (manice). manual of standard procedures for observing and reporting ice conditions. Technical report, Canadian Ice Service—Environment Canada, 2005. CATALOGUE NO. En56-175/2005.

Sukun Cheng, W. Erick Rogers, Jim Thomson, Madison Smith, Martin J. Doble, Peter Wadhams, Alison L. Kohout, Björn Lund, Ola P.G. Persson, Clarence O. Collins III, Stephen F. Ackley, Fabien

Montiel, and Hayley H. Shen. Calibrating a viscoelastic sea ice model for wave propagation in the Arctic fall marginal ice zone. *Journal of Geophysical Research: Oceans*, 122(11):8770–8793, 2017. `https://doi.org/doi/abs/10.1002/2017JC013275`.

Clarence O. Collins III, W. Erick Rogers, Aleksey Marchenko, and Alexander V. Babanin. In situ measurements of an energetic wave event in the Arctic marginal ice zone. *Geophysical Research Letters*, 42(6):1863–1870, mar 2015. `https://doi.org/10.1002/2015gl063063`.

D. Dumont, A. Kohout, and L. Bertino. A wave-based model for the marginal ice zone including a floe breaking parameterization. *Journal of Geophysical Research: Oceans*, 116(C4), 2011. `https://doi.org/10.1029/2010JC006682`.

Peter Guest, P. Ola G. Persson, Shouping Wang, Mary Jordan, Yi Jin, Byron Blomquist, and Christopher Fairall. Low-level baroclinic jets over the new Arctic Ocean. *Journal of Geophysical Research: Oceans*, 123(6):4074–4091, 2018. `https://doi.org/10.1002/2018JC013778`.

Jun Inoue, Masatake E. Hori, Takeshi Enomoto, and Takashi Kikuchi. Intercomparison of surface heat transfer near the arctic marginal ice zone for multiple reanalyses: A case study of september 2009. *SOLA*, 7:57–60, 2011. `https://doi.org/10.2151/sola.2011-015`.

Thomas Jung, Neil D. Gordon, Peter Bauer, David H. Bromwich, Matthieu Chevallier, Jonathan J. Day, Jackie Dawson, Francisco Doblas-Reyes, Christopher Fairall, Helge F. Goessling, Marika Holland, Jun Inoue, Trond Iversen, Stefanie Klebe, Peter Lemke, Martin Losch, Alexander Makshtas, Brian Mills, Pertti Nurmi, Donald Perovich, Philip Reid, Ian A. Renfrew, Gregory Smith, Gunilla Svensson, Mikhail Tolstykh, and Qinghua Yang. Advancing polar prediction capabilities on daily to seasonal time scales. *Bulletin of the American Meteorological Society*, 97(9):1631–1647, 2016. `https://doi.org/10.1175/BAMS-D-14-00246.1`.

A.L. Kohout, M.J.M. Williams, T. Toyota, J. Lieser, and J. Hutchings. In situ observations of wave-induced sea ice breakup. *Deep Sea Research Part II: Topical Studies in Oceanography*, 131:22 – 27, 2016. ISSN 0967-0645. East Antarctic sea-ice physics and ecosystem processes.

Alison L. Kohout, Bill Penrose, Scott Penrose, and Michael J.M. Williams. A device for measuring wave-induced motion of ice floes in the Antarctic Marginal Ice Zone. *Annals of Glaciology*, 56(69):415–424, 2015. `https://doi.org/10.3189/2015aog69a600`.

Jingkai Li, Alison L. Kohout, Martin J. Doble, Peter Wadhams, Changlong Guan, and Hayley H. Shen. Rollover of apparent wave attenuation in ice covered seas. *Journal of Geophysical Research: Oceans*, 122(11):8557–8566, 2017. `https://doi.org/10.1002/2017JC012978`.

Antony K. Liu and Erik Mollo-Christensen. Wave propagation in a solid ice pack. *Journal of Physical Oceanography*, 18(11):1702–1712, 1988. `https://doi.org/10.1175/1520-0485(1988)018<1702:WPIASI>2.0.CO;2`.

Qingxiang Liu, W. Erick Rogers, Alexander V. Babanin, Ian R. Young, Leonel Romero, Stefan Zieger, Fangli Qiao, and Changlong Guan. Observation-based source terms in the third-generation wave model WAVEWATCH III: Updates and verification. *Journal of Physical Oceanography*, 49(2):489–517, 2019. `https://doi.org/10.1175/JPO-D-18-0137.1`.

Michael H. Meylan and Diane Masson. A linear boltzmann equation to model wave scattering in the marginal ice zone. *Ocean Modelling*, 11(3):417 – 427, 2006. ISSN 1463-5003. `https://doi.org/10.1016/j.ocemod.2004.12.008`.

F. Montiel, V. A. Squire, M. Doble, J. Thomson, and P. Wadhams. Attenuation and directional spreading of ocean waves during a storm event in the autumn beaufort sea marginal ice zone. *Journal of Geophysical Research: Oceans*, 123(8):5912–5932, 2018. URL `https://agupubs.onlinelibrary.wiley.com/doi/abs/10.1029/2018JC013763`. `https://doi.org/10.1029/2018JC013763`.

Johannes E. M. Mosig, Fabien Montiel, and Vernon A. Squire. Comparison of viscoelastic-type models for ocean wave attenuation in ice-covered seas. *Journal of Geophysical Research: Oceans*, 120(9): 6072–6090, 2015. `https://doi.org/10.1002/2015JC010881`.

Takehiko Nose, Adrean Webb, Takuji Waseda, Jun Inoue, and Kazutoshi Sato. Predictability of storm wave heights in the ice-free Beaufort Sea. *Ocean Dynamics*, 68(10):1383–1402, Oct 2018. `https://doi.org/10.1007/s10236-018-1194-0`.

Nicolas Rascle and Fabrice Ardhuin. A global wave parameter database for geophysical applications. Part 2: Model validation with improved source term parameterization. *Ocean Modelling*, 70:174 – 188, 2013. ISSN 1463-5003. `https://doi.org/10.1016/j.ocemod.2012.12.001`.

W. Erick Rogers, Alexander V. Babanin, and David W. Wang. Observation-consistent input and whitecapping dissipation in a model for wind-generated surface waves: Description and simple calculations. *Journal of Atmospheric and Oceanic Technology*, 29(9):1329–1346, 2012. `https://doi.org/10.1175/JTECH-D-11-00092.1`.

W. Erick Rogers, Jim Thomson, Hayley H. Shen, Martin J. Doble, Peter Wadhams, and Sukun Cheng. Dissipation of wind waves by pancake and frazil ice in the autumn Beaufort Sea. *Journal of Geophysical Research: Oceans*, 121(11):7991–8007, nov 2016. `https://doi.org/10.1002/2016jc012251`.

Scott R. Stephenson, Laurence C. Smith, Lawson W. Brigham, and John A. Agnew. Projected 21st-century changes to Arctic marine access. *Climatic Change*, 118(3-4):885–899, jan 2013. `https://doi.org/10.1007/s10584-012-0685-0`.

Graig Sutherland, Jean Rabault, Kai H. Christensen, and Atle Jensen. A two layer model for wave dissipation in sea ice. *Applied Ocean Research*, 88:111 – 118, 2019. `https://doi.org/10.1016/j.apor.2019.03.023`.

Jim Thomson, Stephen Ackley, Fanny Girard-Ardhuin, Fabrice Ardhuin, Alex Babanin, Guillaume Boutin, John Brozena, Sukun Cheng, Clarence Collins, Martin Doble, Chris Fairall, Peter Guest, Claus Gebhardt, Johannes Gemmrich, Hans C. Graber, Benjamin Holt, Susanne Lehner, Björn Lund, Michael H. Meylan, Ted Maksym, Fabien Montiel, Will Perrie, Ola Persson, Luc Rainville, W. Erick Rogers, Hui Shen, Hayley Shen, Vernon Squire, Sharon Stammerjohn, Justin Stopa, Madison M. Smith, Peter Sutherland, and Peter Wadhams. Overview of the Arctic Sea State and Boundary Layer Physics Program. *Journal of Geophysical Research: Oceans*, 123(12):8674–8687, 2018. `https://doi.org/10.1002/2018JC013766`.

Ltd. TSK Tsurumi Seiki Co. Oceanographic Equipment Water Quality Monitoring Equipment | TSK Tsurumi Seiki Co., Ltd. - Microwave Type Wave Height Meter WM-2, Jul 2019. URL `http://www.tsk-jp.com/index.php?page=/product/detail/21/2`. [Online; accessed 23. Jul. 2019].

Ruixue Wang and Hayley H. Shen. Gravity waves propagating into an ice-covered ocean: A viscoelastic model. *Journal of Geophysical Research: Oceans*, 115(C6), 2010. `10.1029/2009JC005591`.

Timothy D. Williams, Luke G. Bennetts, Vernon A. Squire, Dany Dumont, and Laurent Bertino. Wave–ice interactions in the marginal ice zone. Part 1: Theoretical foundations. *Ocean Modelling*, 71:81 – 91, 2013. `https://doi.org/10.1016/j.ocemod.2013.05.010`.

WMO. Wmo sea-ice nomenclature. Technical Report 259, The Joint Technical Commission for Oceanography and Marine Meteorology (JCOMM), Mar 2014. URL `https://www.jcomm.info/index.php?option=com_oe&task=viewDocumentRecord&docID=14598`. [Online; accessed 22. Jan. 2020].

Stefan Zieger, Alexander V. Babanin, W. Erick Rogers, and Ian R. Young. Observation-based source terms in the third-generation wave model WAVEWATCH. *Ocean Modelling*, 96:2 – 25, 2015. `https://doi.org/10.1016/j.ocemod.2015.07.014`.

---

## Author Comment (AC2) · 25 Feb 2020

**Author response to tc-2019-285-RC2**

Takehiko Nose on behalf of all authors

We sincerely appreciate the critique and insightful comments that will improve this manuscript. Below outlines our response to each of the critical point raised by the reviewer. The document structure corresponds to that of tc-2019-285-RC2 except for the first section. Please also refer to our response to Reviewer#1 as some major changes were made to address Reviewer#1's comments.

**Overview of major changes**

We begin with an overview of major changes in the revised manuscript to address both reviewers' comment regarding readability of the paper. The change of section titles is shown in bold text (if changed).

**Abstract** The abstract has been rewritten to more clearly emphasise the focus of the study and its findings.

**Section 2.3** The description of WAVEWATCH III $^{®}$ (WW3) has been modified in a way that it is focused on the core idea of the paper: wave-ice interactions. The readability is improved by using equations and compact notations, thereby demonstrating clearly the link between sea ice concentration (SIC) and wave-ice models.

**Section 3 Sea ice concentration: definition, characteristics, and the use in wave-ice models**
The presentation of this section is one of the major changes in the revised version. We have rewritten this section to discuss the concept of SIC from a wave-ice modelling perspective. Despite different scales and objectives, the SIC is being used to moderate the attenuation rate of waves in the MIZ without the model having any consideration to the sea ice field heterogeneity, i.e., subgrid scale physics. We discuss the implication of the length scale of satellite retrieved SIC to the SIC and wave-ice model formulation.
Section 3.2 is shortened and moved to the beginning of Section 5 to give an overview of how satellite retrieved SIC varies on a regional scale so that the transition to the wave hindcast analysis of the region is more fluent.
Original Figures 3 and 4 are removed as they were deemed redundant.

**Section 4 $\Delta c_i$ effects on wave modelling at the observation sites**
Note $\Delta c_i$ is denoted as the SIC uncertainty in the text.

**Section 5 $\Delta c_i$ and wave modelling in the refreezing Chukchi Sea**

**Section 5.1 On- and off-ice wave evolution in the refreezing Chukchi Sea MIZs**

**Section 5.2 Relative significance of $\Delta c_i$ compared with wave-ice interaction parameterisation uncertainty**
We tested the robustness of the results based on sensitivity analyses that tested sea ice thickness (SIT) and the inclusion of scattering. The study findings remain unchanged.

**Section 6 Conclusions and discussions**
This section is shortened considerably and includes discussions on the study outcome.

**General comments**

**Reviewer comment R#2-1** The manuscript examines the impact of observational uncertainty in sea ice concentration on numerical modelling of ocean surface waves in sea ice, using Wavewatch III. The most interesting finding, in my opinion, is that the effect of this uncertainty on wave modelling is large, and larger than the differences between various wave-ice interaction source terms currently included in WW3.

In general, I find the results to be well supported by thorough analysis. The manuscript is a little hard to follow in places (some instances of this noted below in 'presentational comments'). As currently presented, my sense is that this work is of interest to a narrow section of the community (i.e. those working on spectral models of wave-ice interactions) and may be better suited to a more targeted journal. However, this could be addressed by revising the text to stress the aspects that could be interesting to a more general audience e.g. the differences between SIC products and their accuracy near the ice edge; that the sea ice coverage may be more important in determining ocean surface waves in ice than a detailed understanding of wave-ice interaction physics; the level to which waves in the open ocean are influenced by sea ice.

**Author response** We are pleased the significance of the SIC uncertainty on wave-ice models has been conveyed to the reviewer. Regarding the suggested approach to broaden the audience (*"e.g. the differences between SIC products and their accuracy near the ice edge; that the sea ice coverage may be more important in determining ocean surface waves in ice than a detailed understanding of wave-ice interaction physics; the level to which waves in the open ocean are influenced by sea ice"*), our analysis aims to convey that wave-ice model tuning may not be as effective at this time when the knowledge of the true SIC field is too uncertain; however, our work was not intended to suggest that SIC forcing accuracy takes precedence over the detailed understanding of wave-ice interaction physics. Our view is that advancing the knowledge of wave-ice interactions and its implementation in WW3 (or other wave model platforms) constitute the foundation to improving the predictability of ocean waves in ice-covered waters.

The research of ocean waves are becoming an important polar region research topic; waves feedback to the dynamics of atmosphere, ocean, and sea ice, and as such, we believe the manuscript contents are of interest to a general audience of The Cryosphere (TC). The recent acceptance of a coupled ice-ocean-wave model paper [Boutin et al., 2019] in TC reflects the increasing interest of the audience to the topic of ocean waves. We also cited the coupled wave-ice model references in the text to address R#2-10, which should help attract a wider audience.

**Major comments**

**Reviewer#2 comment R#2-2** 'Analysis of significant wave height uncertainty distributions for SIC forcing and wave-ice interaction source terms reveals that they are both sizeable; however, the study concludes the more dominant uncertainty source of modelling wave-ice interactions is the accuracy of satellite retrieved SIC estimates that are used as model forcing.' The authors need to clarify that uncertainty in wave-ice interaction terms is an estimate based only on the various wave-ice interaction terms included in WW3, which as the authors note later in the manuscript are based on a limited number of field observations. It does not sample all possible wave-ice interaction terms, nor does it sample different parameter values.

**Author response** Abstract has been rewritten as mentioned in the overview of major changes. The new abstract reads as below.

**Author's changes in manuscript** Ocean waves are known to decay exponentially when they interact with sea ice. Wave-ice models implemented in a spectral wave model, e.g., WAVEWATCH III ® (WW3), derive the attenuation coefficient based on several different model ice types, i.e., how the model treats sea ice. In the marginal ice zone (MIZ) with SIC < 1, the wave attenuation is moderated by SIC: this implies that the subgrid scale physics is missing, and the accuracy of SIC plays an important role in the predictability. Satellite retrieved SIC data (or a sea ice model that assimilates

them) are often used to force wave-ice models, but these data are known to have uncertainty. Six satellite retrieved SIC products, based on four algorithms applied to SSMIS and AMSR2 data, were used in the WW3 hindcast experiment to study the effect of SIC uncertainty $\Delta$SIC on modelling MIZ waves during the 2018 R/V Mirai observational campaign in the refreezing Chukchi Sea. The results show that $\Delta$SIC can cause wave prediction discrepancies in ice cover. There is evidence that bivariate uncertainty data (model significant wave heights and SIC forcing) are correlated, although off-ice wave growth is more complicated due to the cumulative effect of $\Delta$SIC along an MIZ fetch. Further, we found that the effect of $\Delta$SIC can be large enough, such that it overwhelms the choice of model ice types, i.e., wave-ice interaction parameterisations. Despite these parameterisations being derived from different concepts and missing the subgrid scale physics relating to sea ice field heterogeneity, the accuracy of satellite retrieved SIC used as model forcing is the primary error source of modelling MIZ waves in the refreezing ocean.

**Reviewer comment R#2-3** It is well-known that SIC satellite observations underestimate concentrations for low thicknesses, less than 35 cm (Ivanova et al. 2015). This should be stated in the Introduction. I think the newer result here is how varied the satellite products are for these low thicknesses. This should be made clearer in the text.

**Author response** The long-known deficiency of satellite derived SIC accuracy in thin ice was indeed missing. We have incorporated the comment in Section 1 Introduction.

**Author's changes in manuscript** . . . To date, there is no robust validation of any algorithm, so users are urged to understand strengths and weaknesses of the algorithms when using and interpreting the data [Ivanova et al., 2015, Comiso et al., 2017]. The long-known SIC discrepancies imply there is uncertainty in the knowledge of true sea ice coverage [Notz, 2014]. The uncertainty is potentially greater for MIZs in the refreezing ocean as satellite derived SIC estimates are known to underestimate thin ice less than 35 cm [Heygster et al., 2014, Ivanova et al., 2015].

**Reviewer comment R#2-4** Sec 2.3 This section needs more explanation to make it suitable for a more general audience - i.e. those not familiar with WW3 wave-ice interaction terms. What is the IS0 switch?

**Author response** We have modified Section 2.3 to be more readable to a broader audience. Please see the overview of major changes. The passage relating to the major change of this section is given below.

**Author's changes in manuscript** . . . The sum of these source terms $s$ is expressed based on the following default scaling in ice-covered waters:

$$s = (1 - c_i)(s_{\text{wind}} + s_{\text{dissipation}}) + c_i s_{\text{ice}} + s_{\text{non-linear interactions}}.$$

Specifically to this study, $c_i$ relates to the satellite retrieved SIC and $s_{\text{ice}}$ to the ice type, i.e., how the model treats sea ice. The effect of sea ice on waves are represented via the modified dispersion relation $\sigma = \sigma(\overline{k})$ where $|\overline{k}| = k = k_r + ik_i$. The real part $k_r$ is the physical wavenumber and alters the propagation speed of waves in a sea ice field (analogous to effects of shoaling and refraction by bathymetry), and the imaginary part $k_i$ is the exponential decay coefficient. $k_i$ is introduced in the model as $s_{\text{ice}} = -2c_g k_i N$ for fully ice-covered sea, i.e., $c_i$=1, and the solution to $\frac{dN}{dt} = s_{\text{ice}}$ is $N_0 e^{-2c_g k_i t}$. There are five options for treating sea ice in WW3 denoted as IC1–5; $c_i$ provides the scaling in the linkage between $s_{\text{ice}}$ and ICX as

$$\frac{dN}{dt} = c_i s_{\text{ice}} = -2c_i c_g k_i(f, p_1, ..., p_n)N$$

where $p_1, ..., p_n$ are the sea ice properties, e.g., effective shear modulus and effective viscosity. Therefore, the rate of attenuation depends on the wave period and sea ice properties, which is moderated by $c_i$, i.e., $N_0 e^{-2c_i c_g k_i t}$.

The wave-ice models implemented in WW3 that calculate $k_r$ to model $k_i$ are as follows: IC2 calculates dissipation due to basal friction in the boundary layer below an ice sheet, which is modelled as a

continuous thin elastic plate based on the work of Liu and Mollo-Christensen [1988]; IC3 treats sea ice as a visco-elastic layer based on Wang and Shen [2010], which calculates the internal stress of the ice cover based on storage and dissipation; and IC5 is a visco-elastic beam model based on Mosig et al. [2015]. The dispersion relation of these models are provided in Appendix B. Ardhuin et al. [2018], Boutin et al. [2018] (IC2) and Rogers et al. [2016], Cheng et al. [2017] (IC3) describe the progress of these $s_{ice}$ parameterisations using the refreezing Beaufort Sea data of Thomson et al. [2018]. These wave-ice models can be combined with an energy-conservative scattering attenuation model denoted as IS1 and IS2 [Meylan and Masson, 2006, Dumont et al., 2011, Williams et al., 2013, Ardhuin et al., 2018, Boutin et al., 2018].

**Reviewer comment R#2-5** What is the estimated of spatial domain of the sea-truth concentration observations? Are these comparable to the resolution of the satellite products?

**Author response** Here, we infer the comment pertains to Section 3.1. This section has been rewritten to address Reviewer#1's comments (refer to R#1-1, R#1-2, and R#1-19 in our response to the reviewer#1). Please also refer to the overview of major changes. Section 3 now reads as below.

**Author's changes in manuscript** WMO [2014] defines SIC as "the ratio expressed in tenths describing the amount of the sea surface covered by ice as a fraction of the whole area being considered". The so-called "area considered" presumably varies for different objectives. Length scale of $O(10)$ km may be adequate for sea ice extent climatology, but for wave-ice interactions, the wave provides a scale in a phase-resolved sense. Satellite derived SIC represents the fraction of ice-covered water over a large area, sufficiently large enough that the SIC represents a property of a continuum. In reality, the sea ice in the MIZ is granular, and ice floes jam due to horizontal convergence by Langmuir circulation, internal waves, and wind variability, resulting in a formation of features such as ice bands and wind streaks—with which waves likely interact distinctively.

On 14 November 2018 during the MIZ transect observation, R/V Mirai encountered moderate on-ice waves with an $H_{m0}$ up to around 2.00 m propagating towards the ice edge (this $H_{m0}$ estimate is consistent from both the shipboard wave data described in Appendix A and hindcast models as discussed later). Figure 2 presents a series of snapshot images of the sea ice field during the encounter. R/V Mirai traversed over 10 km in the MIZ from the ice edge, and each image area extends at least over 1 km conservatively (using the crude distance to horizon calculation). These images depict the heterogeneous sea ice field, both in distribution and ice types, that waves propagate when they enter an MIZ. Because WW3 wave-ice interaction models are scaled according to $\frac{dN}{dt} = c_i s_{\text{ice}}$ (Equation 4), the subgrid scale physics is completely missing. It is plausible the subgrid scale distribution of SIC and ice types can be treated in a stochastic manner to provide meaningful mean values to the grid-scale model. On the other hand, $c_i$ also affects the WW3 wave-ice model by means of scaling (Equation 4). Figure 2 shows SIC data from eight satellite retrieved products described in Section 2.2 during the event. The SIC estimates interpolated at the R/V Mirai positions largely deviate among the products, characterising the uncertainty of the satellite retrieved SIC. Moreover, the entire time series of the MIZ transect observation depicts $\Delta c_i$ is persistent (Figures A4 to A6 of Appendix D). Hereafter, we show how large the effect of $\Delta c_i$ on modelling MIZ waves can be, so much so that it overwhelms the choice of $s_{ice}$, e.g., ICX.

**Reviewer comment R#2-6** Sec 3.2 - why are only three products considered here? Conclusions on which products perform best (or worst) for this region would be useful and of broader interest to the cryospheric community.

**Author response** The three products were selected to convey the magnitude of uncertainty. Since the uncertainty was defined as $\Delta c_i = max(c_i) - min(c_i)$, three products are sufficient to convey this point. Showing all eight products clutters the figure and potentially distract readers from the main point.

In addition, the revised manuscript is presented such that the focus is apparent, which is the wave-ice models and SIC forcing and not about the SIC data. We anticipate this will be apparent when the revised manuscript is read. Nevertheless, regarding ranking the product performance, we agree such a

conclusion would be a significant contribution. We have attempted to rank the SIC products, but the result is inconclusive primarily due to the number of days comprehensive mosaic can be produced and inconsistent variability of SIC estimates. We expect such an analysis warrants a separate dedicated study to produce noteworthy outcomes, especially considering the previous intercomparison studies, such as Ivanova et al. [2015] and Comiso et al. [2017], concluded there is no one product that is superior. They state that the choice of SIC products are largely dictated by the SIC data application, and we concur with this statement based on our field observation and analysis conducted.

**Minor comments**

**Reviewer comment R#2-7 :** **L22.** 'which are greater in the Arctic regions.' As compared to what?

**Author response** We made a change to this passage (bold text) as below.

**Author's changes in manuscript**. . . From a practical view point, this downward trend of sea ice decline opens trans-Arctic shipping routes connecting Europe and Asia for longer times of the year; potential global economic benefits of non-ice breakers accessing routes like Northern Sea Route and North West Passage are substantial [Stephenson et al., 2013, Bekkers et al., 2018]. **The increasing vessel traffic implies that adequate prediction capabilities will become crucial to assist ships in polar waters to circumnavigate hazards such as high winds and waves, collision with perennial sea ice, and sea-spray icing; however, Jung et al. [2016] describe that the existing polar prediction systems need to be urgently enhanced to effectively manage the risks and opportunities associated with growing human activities, and the Polar Prediction Project (PPP) has contributed to advancing the predictive capabilities. While wave forecasting in polar oceans is still in its early years, the need for advancing wave forecast capacity will only grow in the emerging Arctic Ocean.** This paper focuses on the effect of SIC uncertainty on third-generation spectral wave model simulations in and near a MIZ.

**Reviewer comment R#2-8 :** **L23.** 'sustainable developments of the Arctic Ocean' - this statement sounds strange.

**Author response** Please see author response to R#2-7.

**Reviewer comment R#2-9 :** **L28.** 'dispersion relation'→'the dispersion relation'.

**Author response** Corrected.

**Reviewer comment R#2-10:** **L32.** Other recent work on wave-ice interactions could be cited here: Zhang et al. 2019 (https://doi.org/10.1016/j.ocemod.2019.101532), Roach et al. 2019 (https://doi.org/10.1029/2019MS001836).

**Author response** This passage pertains specifically to the standalone WW3 wave-ice interaction source term developments, so the suggested references are unrelated. We added new sentences at the end of the paragraph as shown in the bold text below. This way, we can relate the study to a broader audience, which also helps to address R#2-1.

**Author's changes in manuscript** . . . Standalone contemporary spectral wave models simulate wave-ice interactions using sea ice as forcing; in this space, the intensive field measurements of the Arctic Sea State and Boundary Layer Physics Program [Thomson et al., 2018] have made a solid contribution to the recent advance of The WAVEWATCH III ® Development Group (WW3DG) [2019] wave-ice interaction parameterisation. Rogers et al. [2016], Cheng et al. [2017], Ardhuin et al. [2018], Boutin et al. [2018] describe the development and optimisation of the latest WW3 parameterisations for wave evolution in sea ice cover. Despite the progress, Squire [2018], Thomson et al. [2018] qualify accurately quantifying the wave decay and connecting the associated mechanisms over a large domain still remain a challenge because sea ice fields are notoriously heterogeneous; therefore, the wave-ice interaction source term is a source of uncertainty when simulating wave evolution in MIZs. **Recent**

developments of coupled wave-ice-ocean models on a pan-Arctic scale [Boutin et al., 2019, Roach et al., 2019, Zhang et al., 2020] reflect the growing interest in the surface wave's role in the atmosphere, ocean, and sea ice dynamics: perhaps this indicates advancing the wave-ice interaction physics is becoming a more pertinent issue to broader scientific communities.

**Reviewer comment R#2-11 : L37.** 'Besides the model interior' What does this mean?

**Author response** We are referring to the inner part of numerical models. We added "wave-ice interaction parameterisations" as an example.

**Reviewer comment R#2-12: L37.** 'like SIC'→'in products such as SIC'?

**Author response** Changed as suggested.

**Reviewer comment R#2-13: L69.** 'generally'→'generally between'.

**Author response** Corrected.

**Reviewer comment R#2-14: L70.** 'Albeit the MIZ coverage..' - this sentence is unclear.

**Author response** We have attempted to make it clearer.

**Author's changes in manuscript** Although the MIZ coverage was less expansive, daily observation of the sea ice conditions at the same geographical locations for an extended period is rare if not unique because of exhaustive ship time required.

**Reviewer comment R#2-15: L82.** 'mostly same'→'mostly the same'.

**Author response** Corrected.

**Reviewer comment R#2-16 : L96.** remove 'exhaustive' (this is an opinion).

**Author response** "exhaustive" changed to "large".

**Reviewer comment R#2-17 : L102.** 'most frequent'→'most frequently'.

**Author response** Changed as suggested.

**Reviewer comment R#2-18 :  L102.** 'leading algorithms' leading in what? Perhaps you mean the most commonly used?.

**Author response** We realised the phrase is unnecessary, so it is removed.

**Reviewer comment R#2-19 : L152.** 'sanity checked' inappropriate language for a publication.

**Author response** Changed to "tested".

**Reviewer comment R#2-20 : L153.** 'leading packages' again, leading in what way? Most commonly used? Most skill?.

**Author response** Changed to "commonly used".

**Reviewer comment R#2-21 : L211.** change to 'SSTs exceeded 0oC'.

**Author response** Corrected.

**R#2-22 : L211.** At what depth were the shipboard SSTs measured? Are they affected by the ship itself?.

**Author response** The technical details of the instruments were given in Appendix A, which was

introduced in L76 of the original version: "−1 m below the sea surface with further 5 m inlet to the gauge" (L458). We are not aware of any ship effects on the SST measurement. During the cruise, we note that sea ice, even grease ice, was observed only when shipboard SSTs were less than around −1.5 ℃. Accordingly, the gauge was sufficiently accurate for this discussion.

**Reviewer comment R#2-23 : L212.** No s after MIZ.

**Author response** "s" removed.

**Reviewer comment R#2-24 : L218.** 'forseeable'→'visible'.

**Author response** Corrected the elementary error: thank you.

**Reviewer comment R#2-25 : L223.** 'The sea ice tends to cluster, so it appears dense where the ice exists.' - reword.

**Author response** Please see author's response to R#2-5.

**Reviewer comment R#2-26 : L230.** 'depict the'→'show that the' ?

**Author response** Changed as suggested.

**Reviewer comment R#2-27 : L264.** What does 'best practice wave models' mean? Perhaps replace with 'commonly used'?

**Author response** Changed to "high quality".

**Reviewer comment R#2-28 : L266.** 'a bit over' - informal language.

**Author response** Changed to "slightly".

**Reviewer comment R#2-29 : L277.** 'as uncertainty generally increased in the MIZs.' - unclear. No s after MIZ.

**Author response** "in the MIZs" removed as it was unnecessary.

**Reviewer comment R#2-30 : L299.** remove the rest of the sentence after 'ongoing work' - not necessary.

**Author response** Removed the sentence as it was unnecessary.

**Reviewer comment R#2-31 : L323.** add 'that' - 'demonstrated that'.

**Author response** Added "that".

**Reviewer comment R#2-32 : L338.** 'like a'→'as a'.

**Author response** Changed as suggested.

**Reviewer comment R#2-33 : L339.** 'effects. . .are' (not is).

**Author response** Corrected.

**Reviewer comment R#2-34 : L443.** Reword.

**Author response** The sentence is removed as it was unnecessary.

Thank you kindly for your review and consideration.

**References**

Fabrice Ardhuin, Guillaume Boutin, Justin Stopa, Fanny Girard-Ardhuin, Christian Melsheimer, Jim Thomson, Alison Kohout, Martin Doble, and Peter Wadhams. Wave attenuation through an Arctic marginal ice zone on October 12, 2015: Part 2. Numerical modeling of waves and associated ice break-up. *Journal of Geophysical Research: Oceans*, 2018. `https://doi.org/10.1002/2018JC013784`.

Eddy Bekkers, Joseph F. Francois, and Hugo Rojas-Romagosa. Melting ice caps and the economic impact of opening the northern sea route. *The Economic Journal*, 128(610):1095–1127, 2018. `https://doi.org/10.1111/ecoj.12460`.

G. Boutin, C. Lique, F. Ardhuin, C. Rousset, C. Talandier, M. Accensi, and F. Girard-Ardhuin. Toward a coupled model to investigate wave-sea ice interactions in the arctic marginal ice zone. *The Cryosphere Discussions*, 2019:1–39, 2019. `https://doi.org/10.5194/tc-2019-92`.

Guillaume Boutin, Fabrice Ardhuin, Dany Dumont, Caroline Sevigny, Fanny Girard-Ardhuin, and Mickael Accensi. Floe size effect on wave-ice interactions: Possible effects, implementation in wave model, and evaluation. *Journal of Geophysical Research: Oceans*, 123(7):4779–4805, 2018. `https://doi.org/10.1029/2017JC013622`.

Sukun Cheng, W. Erick Rogers, Jim Thomson, Madison Smith, Martin J. Doble, Peter Wadhams, Alison L. Kohout, Björn Lund, Ola P.G. Persson, Clarence O. Collins III, Stephen F. Ackley, Fabien Montiel, and Hayley H. Shen. Calibrating a viscoelastic sea ice model for wave propagation in the Arctic fall marginal ice zone. *Journal of Geophysical Research: Oceans*, 122(11):8770–8793, 2017. `https://doi.org/doi/abs/10.1002/2017JC013275`.

Josefino C. Comiso, Walter N. Meier, and Robert Gersten. Variability and trends in the Arctic sea ice cover: Results from different techniques. *Journal of Geophysical Research: Oceans*, 122(8):6883–6900, 2017. `https://doi.org/10.1002/2017JC012768`.

D. Dumont, A. Kohout, and L. Bertino. A wave-based model for the marginal ice zone including a floe breaking parameterization. *Journal of Geophysical Research: Oceans*, 116(C4), 2011. `https://doi.org/10.1029/2010JC006682`.

G. Heygster, M. Huntemann, N. Ivanova, R. Saldo, and L. T. Pedersen. Response of passive microwave sea ice concentration algorithms to thin ice. In *2014 IEEE Geoscience and Remote Sensing Symposium*, pages 3618–3621, July 2014. `https://doi.org/10.1109/IGARSS.2014.6947266`.

N. Ivanova, L. T. Pedersen, R. T. Tonboe, S. Kern, G. Heygster, T. Lavergne, A. Sørensen, R. Saldo, G. Dybkjær, L. Brucker, and M. Shokr. Inter-comparison and evaluation of sea ice algorithms: towards further identification of challenges and optimal approach using passive microwave observations. *The Cryosphere*, 9(5):1797–1817, 2015. `https://doi.org/10.5194/tc-9-1797-2015`.

Thomas Jung, Neil D. Gordon, Peter Bauer, David H. Bromwich, Matthieu Chevallier, Jonathan J. Day, Jackie Dawson, Francisco Doblas-Reyes, Christopher Fairall, Helge F. Goessling, Marika Holland, Jun Inoue, Trond Iversen, Stefanie Klebe, Peter Lemke, Martin Losch, Alexander Makshtas, Brian Mills, Pertti Nurmi, Donald Perovich, Philip Reid, Ian A. Renfrew, Gregory Smith, Gunilla Svensson, Mikhail Tolstykh, and Qinghua Yang. Advancing polar prediction capabilities on daily to seasonal time scales. *Bulletin of the American Meteorological Society*, 97(9):1631–1647, 2016. `https://doi.org/10.1175/BAMS-D-14-00246.1`.

Antony K. Liu and Erik Mollo-Christensen. Wave propagation in a solid ice pack. *Journal of Physical Oceanography*, 18(11):1702–1712, 1988. `https://doi.org/10.1175/1520-0485(1988)018<1702:WPIASI>2.0.CO;2`.

Michael H. Meylan and Diane Masson. A linear boltzmann equation to model wave scattering in the marginal ice zone. *Ocean Modelling*, 11(3):417 – 427, 2006. ISSN 1463-5003. `https://doi.org/10.1016/j.ocemod.2004.12.008`.

Johannes E. M. Mosig, Fabien Montiel, and Vernon A. Squire. Comparison of viscoelastic-type models for ocean wave attenuation in ice-covered seas. *Journal of Geophysical Research: Oceans*, 120(9): 6072–6090, 2015. `https://doi.org/10.1002/2015JC010881`.

D. Notz. Sea-ice extent and its trend provide limited metrics of model performance. *The Cryosphere*, 8(1):229–243, 2014. `https://doi.org/10.5194/tc-8-229-2014`.

Lettie A. Roach, Cecilia M. Bitz, Christopher Horvat, and Samuel M. Dean. Advances in modeling interactions between sea ice and ocean surface waves. *Journal of Advances in Modeling Earth Systems*, 11(12):4167–4181, 2019. `https://doi.org/10.1029/2019MS001836`.

W. Erick Rogers, Jim Thomson, Hayley H. Shen, Martin J. Doble, Peter Wadhams, and Sukun Cheng. Dissipation of wind waves by pancake and frazil ice in the autumn Beaufort Sea. *Journal of Geophysical Research: Oceans*, 121(11):7991–8007, nov 2016. `https://doi.org/10.1002/2016jc012251`.

Vernon A. Squire. A fresh look at how ocean waves and sea ice interact. *Philosophical Transactions of the Royal Society A: Mathematical, Physical and Engineering Sciences*, 376(2129):20170342, 2018. `https://doi.org/10.1098/rsta.2017.0342`.

Scott R. Stephenson, Laurence C. Smith, Lawson W. Brigham, and John A. Agnew. Projected 21st-century changes to Arctic marine access. *Climatic Change*, 118(3-4):885–899, jan 2013. `https://doi.org/10.1007/s10584-012-0685-0`.

The WAVEWATCH III [R] Development Group (WW3DG). User manual and system documentation of WAVEWATCH III [R] version 6.07. Tech. Note 333, NOAA/NWS/NCEP/MMAB, College Park, MD, USA, 2019. 465 pp. + Appendices.

Jim Thomson, Stephen Ackley, Fanny Girard-Ardhuin, Fabrice Ardhuin, Alex Babanin, Guillaume Boutin, John Brozena, Sukun Cheng, Clarence Collins, Martin Doble, Chris Fairall, Peter Guest, Claus Gebhardt, Johannes Gemmrich, Hans C. Graber, Benjamin Holt, Susanne Lehner, Björn Lund, Michael H. Meylan, Ted Maksym, Fabien Montiel, Will Perrie, Ola Persson, Luc Rainville, W. Erick Rogers, Hui Shen, Hayley Shen, Vernon Squire, Sharon Stammerjohn, Justin Stopa, Madison M. Smith, Peter Sutherland, and Peter Wadhams. Overview of the Arctic Sea State and Boundary Layer Physics Program. *Journal of Geophysical Research: Oceans*, 123(12):8674–8687, 2018. `https://doi.org/10.1002/2018JC013766`.

Ruixue Wang and Hayley H. Shen. Gravity waves propagating into an ice-covered ocean: A viscoelastic model. *Journal of Geophysical Research: Oceans*, 115(C6), 2010. `10.1029/2009JC005591`.

Timothy D. Williams, Luke G. Bennetts, Vernon A. Squire, Dany Dumont, and Laurent Bertino. Wave–ice interactions in the marginal ice zone. Part 1: Theoretical foundations. *Ocean Modelling*, 71:81 – 91, 2013. `https://doi.org/10.1016/j.ocemod.2013.05.010`.

WMO. Wmo sea-ice nomenclature. Technical Report 259, The Joint Technical Commission for Oceanography and Marine Meteorology (JCOMM), Mar 2014. URL `https://www.jcomm.info/index.php?option=com_oe&task=viewDocumentRecord&docID=14598`. [Online; accessed 22. Jan. 2020].

Yang Zhang, Changsheng Chen, Robert C. Beardsley, William Perrie, Guoping Gao, Yu Zhang, Jianhua Qi, and Huichan Lin. Applications of an unstructured grid surface wave model (fvcom-swave) to the arctic ocean: The interaction between ocean waves and sea ice. *Ocean Modelling*, 145:101532, 2020. `https://doi.org/10.1016/j.ocemod.2019.101532`.

---

## Author Response (AR1)

**Rebuttal for tc-2019-285**

Takehiko Nose on behalf of all authors

**Overview of major changes**

We begin with an overview of major changes in the revised manuscript to address both reviewers' comment regarding readability of the paper. The change of section titles is shown in bold text (if changed).

**Abstract** The abstract has been rewritten to more clearly emphasise the focus of the study and its findings.

**Section 2.3** The description of WAVEWATCH III $^®$ (WW3) has been modified in a way that it is focused on the core idea of the paper: wave-ice interactions. The readability is improved by using equations and compact notations, thereby demonstrating clearly the link between sea ice concentration (SIC) and wave-ice models. Please see the modified Section 2.3 starting at P5L142 (i.e., page 5 line 142) of the revised manuscript.

**Section 3 Sea ice concentration: definition, characteristics, and the use in wave-ice models**
The presentation of this section is one of the major changes in the revised version. We have rewritten this section to discuss the concept of SIC from a wave-ice modelling perspective. Despite different scales and objectives, the SIC is being used to moderate the attenuation rate of waves in the MIZ without the model having any consideration to the sea ice field heterogeneity, i.e., subgrid scale physics. We discuss the implication of the length scale of satellite retrieved SIC to the SIC and wave-ice model formulation. Please see the modified Section 3 starting at P7L205 of the revised manuscript.
Section 3.2 is shortened and moved to the beginning of Section 5 (P10L284 of the revised manuscript) to give an overview of how satellite retrieved SIC varies on a regional scale so that the transition to the wave hindcast analysis of the region is more fluent.
Original Figures 3 and 4 are removed as they were deemed unnecessary.

**Section 4 $\Delta c_i$ effects on wave modelling at the observation sites**
Note $\Delta c_i$ is denoted as the SIC uncertainty in the text.

**Section 5 $\Delta c_i$ and wave modelling in the refreezing Chukchi Sea**

**Section 5.1 On- and off-ice wave evolution in the refreezing Chukchi Sea MIZs**

**Section 5.2 Relative significance of $\Delta c_i$ compared with wave-ice interaction parameterisation uncertainty**
We tested the robustness of the results based on sensitivity analyses that tested sea ice thickness (SIT) and the inclusion of scattering. The study findings remain unchanged. Please see the revised manuscript P12L383–P13L398.

**Section 6 Conclusions and discussions**
This section is shortened considerably and includes discussions on the study outcome. Please see the revised manuscript P13L403–P14L420.

**Reviewer #1 General comments**

**Reviewer comment R#1-1** This paper presents interesting results, confirming some statements about the sensitivity of wave-inice modelling to sea ice concentration made in previous studies. The fact that the uncertainty in the estimated sea ice concentration has a larger effect on the uncertainty of the wave height estimated in-ice than the change in the wave-in-ice attenuation parameterization is a nice finding that illustrates very well the difficulties faced by wave-in-ice modellers. However, the paper suffers from a writing style that is confusing in a number of places. This is particularly the case for sections 3, 4 and 5. It is a bit paradoxal, as sometimes the important information is hidden in a succession of very wordy sentences, making it hard for the reader to get the message, and sometimes it seems that the authors wanted to avoid repeating themselves whereas the reader would happily appreciate some help.

**Author response** We are pleased the essence of this manuscript was conveyed to the reviewer. To enhance the clarity of our message, efforts were made to improve the readability of the paper by following reviewer's suggestions. Please see the overview of major changes.

**Reviewer comment R#1-2** I am also not convinced by the usefulness and the novelty of section 3, at least in its current form. To me, the most interesting part of the study lies in section 5, but it is overshadowed by the lack of clarity of the previous sections. It would be worth shifting the emphasis of the paper to this section more quickly.

**Author response** We appreciate the critical comment regarding the Section 3 content. Please see the overview of major changes and the modified Section 3 starting at P7L205 of the revised manuscript.

**Reviewer #1 Specific comments**

**Reviewer comment R#1-3 : P1L22.** "model uncertainties...", which models are the authors talking about here?

**Author response** We made a change to this passage. Please see the revised manuscript P1L22–P2L27.

**Reviewer comment R#1-4 : P2L25.** The authors should consider giving a definition of the MIZ.

**Author response** The MIZ definition is added. Please see the revised manuscript P2L29–31.

**Reviewer comment R#1-5 : P2L36.** "wave-ice interaction source term": the authors haven't introduced the principle of spectral wave models yet, so it is not clear for everyone what the wave-ice interaction source term is.

**Author response** Changed the object from "the wave-ice interaction source term" to "the wave-ice interaction parameterisation". Please see the revised manuscript P2L37.

**Reviewer comment R#1-6 : P3L64.** Here the authors introduce the methods used for the measurements on board on the R/V Mirai. But the way the so called "sea truth" images were taken is only described in section 3.1 (P7L212), lost in comments on the results. Similarly, before the end of section 3.1, a definition of how the uncertainty of a given quantity is computed is suddenly introduced, without even a proper transition (P8L225). This mix between comments on the results and details of the methods makes section 3.1 very confusing and much longer than it should be.

**Author response** Please see author's response to R#1-2. Regarding the uncertainty definition, it is moved to Section 2.2 and introduced where the eight SIC products are described. Please see the revised manuscript P4L113.

**Reviewer comment R#1-7 : P4L88.** I think the word "translated" is not appropriate here (and in some other places). The authors could consider using "interpreted", "inferred".

**Author response** SIC is a variable that is calculated from brightness temperatures, so we would prefer to have a quantitative phrase rather than being qualitative. In that sense, "calculate" is more appropriate here. Please see the revised manuscript P4L101. Regarding the use of "translate" in other places, we have replaced them with more straightforward expressions.

**Reviewer comment R#1-8 : P4L118.** The sentences about the different grids the authors could have used and the one they are actually using are really confusing. Maybe they should try to cut them into more but shorter sentences, each dealing with one region and one resolution.

**Author response** Changed to a shorter sentence as suggested. Please see the revised manuscript P5L130–133.

**Reviewer comment R#1-9 : P5L143.** "A curvilinear grid [...] sea point cells." I found this whole paragraph very confusing. As a reader I found very difficult to understand the links between each sentence, and the expression "The grid" seems to be applied to different things. As an example, they first refer to the model "geographical" grid, then to the spectral grid, then they use again the expression "the grid" to give details about the bathymetry. A quick reminder of the region they are focusing on would also be welcome, especially as they refer to the "other seas" at the border of "the domain".

**Author response** Adding appropriate modifiers to "grid" was necessary, and the spectral grid description was mixed up in the regional grid description, which was confusing. The paragraph is clarified and shortened. Please see the revised manuscript P7L185–189.

**Reviewer comment R#1-10 : P6L152.** "During the version upgrade of TodaiWW3-ArCS...": which version upgrade are the authors referring to? Are they sure it is relevant for the paper? I think the authors could just state that they are using the ST6 parameterization for the non-ice source terms as it was previously shown to give the best results for the case discussed in Nose et al. (2018) with the model being forced by ERA5 winds.

**Author response** Corrected as suggested. Please see the revised manuscript P6L179–184.

**Reviewer comment R#1-11 : P6L157.** "The s_ice term is composed": I think the use of composed is misleading here. The attenuation terms the authors mention are included in different parameterizations (ISX, ICX), and they are not all compatible with each other. I would suggest an expression like "The s_ice term represents wave-in-ice attenuation processes such as..." for instance.

**Author response** The passage relating to the WW3 wave-ice interaction term is one of the major changes. Please see the revised manuscript P6L154–171.

**Reviewer comment R#1-12: P6L160.** "The dominant floe size [...] IS0 switch.": Here I have the feeling that the authors want to justify why they did not include scattering terms in their wave-in-ice source terms. I think this justification is very long and with unnecessary information (the way scattering terms work in WW3 for instance). I think it would be much clearer simply stating that during the cruise, sea ice in the MIZ was mainly made of grease, nilas and pancake ice, for which scattering is not expected to be the dominant process (Montiel et al., 2018), and therefore scattering was not considered here. Specifying the WW3 switch IS0 is also unnecessary.

**Author response** Thank you for this suggestion. The suggested text is succinct and used in the manuscript. Please see the revised manuscript P6L174.

**Reviewer comment R#1-13: P6L170.** "The underlying principle of sea ice models is that sea ice is treated a continuum." Firstly, there is a small typo, it should be "treated as...". Secondly, this statement might be true for the sea ice models used in climate models, but it ignores discrete elements

sea ice models, often used for sea ice-structure interactions. They can also be used to study wave-ice interactions (Herman et al., 2015). Actually I think the sentences between P6L170 and P6L173 could be shortened. The reason the authors choose this parameterization is because it has been developed to represent similar ice conditions to the ones encountered by the R/V Mirai, which is not the case for the other parameterizations.

**Author response** We have incorporated the reviewer's comment. Please see the revised manuscript P6L172.

**Reviewer comment R#1-14: P6L175.** "... the treatment of independent SIC and sea ice thickness data sets is not a trivial matter." I am not sure I understand this statement. Would it be possible to develop this idea a bit more?

**Author response** It is much clearer just to say the constant thickness was applied, so we can evaluate solely the SIC uncertainty on wave modelling. The text is modified. Please see the revised manuscript P6L175–178.

**Reviewer comment R#1-15: P6L179.** "the former [...] experiment." The first part of the sentence is unnecessary in my opinion. I would also recommend avoiding using the word "domain" for the the SIC product, as it usually refers to the study region.

**Author response** Corrected as suggested. Please see the revised manuscript P7L190–192.

**Reviewer comment R#1-16: P6L184 .** "By doing so [...] in atmospheric models." This passage is very confusing, I do not understand the point the authors are trying to make. They should either rewrite it, if they think it is important, or remove it.

**Author response** The passage is rewritten. Please see the revised manuscript P7L198–203.

**Reviewer comment R#1-17: P7L189.** "WW3 is a standalone wave model...": In this case, it is indeed used in standalone mode, but WW3 can be coupled.

**Author response** This is true the modifier "standalone" was unnecessary. Further, the entire sentence turned out to be superfluous when we addressed R#1-16, and as such, the sentence is removed.

**Reviewer comment R#1-18: P7L190.** "numerical stability is unaffected": By what?

**Author response** This sentence is removed. Please see author's responses to R#-16 and R#-17.

**Reviewer comment R#1-19: P7L195, Section 3.** I am not particularly convinced by the major interest of this section, and particularly by the interest of comparing pictures taken from the boat to the sea ice concentration products in section 3.1. Which angle does the pictures cover? Which surface area are they representative of? As the authors say, sea ice tends to cluster in the MIZ, and the fact that the sea ice concentration is not uniformly distributed spatially is well known by anyone who has had the opportunity to go in sea ice covered places. As I understand it, these observations motivated the study, but to me the interest of this paper does not lie here, and I actually think the removal of section 3.1 could potentially improve the clarity of the paper. If the authors want to keep it, they must make clearer the novelty of these observations and their interest. Moreover, I find the writing style very confusing in section 3.1. Section 3.2 is more convincing and clearer, but it is hard to see any novelty in it. It could maybe bring more to the study by linking it more closely to the results of section 4 and 5.

**Author response** Please see the overview of major changes and author's response to R#1-2.

**Reviewer comment R#1-20: P7L197.** I don't understand the use of "respectively" here.

**Author response** This passage is moved to an appendix, and the "respectively" is removed there. Please see the revised manuscript P16L482.

**Reviewer comment R#1-21: P9L270.** "... the sea ice cut-off criterion is not clear in the documentation." This is not a very satisfying statement. Have the authors considered contacting the people in charge of ArcMFC to get more information about this criterion?

**Author response** Thank you for this comment: the statement was unsatisfactory. We found out that the wave-ice interaction of Sutherland et al. [2019] was implemented in December, 2019: the month after our Arctic Ocean observation. We still find value in reporting the ARCMFC model data in our results, but we have amended the ARCMFC model description. Please see the revised manuscript P8L233.

**Reviewer comment R#1-22: P9L275.** "... but the Piper [...] did not reflect the sea state.": It is quite confusing, please rephrase.

**Author response** Waves in the Chukchi Sea during the study period was dominated by wind seas, which have shorter wavelength relative to the ship dimensions. These short waves are impeded by R/V Mirai's hull, so the shipboard Piper#15 has limitations on measuring wind seas. As such, most of the Piper#15 data did not reflect the true wave field. We have rephrased the paragraph, and the explanation of this point is moved to an appendix. So it is removed from the main text. Please refer to author's response to R#1-34 for how we addressed this comment.

**Reviewer comment R#1-23: P9L283.** "Furthermore [...] an important role." This is very noticeable indeed. It would be very interesting to give an estimate of the spatial attenuation coefficient at the ice edge assuming an exponential wave attenuation, in order to show how it compares with the models and other reported observations (for instance Kohout et al., 2015).

**Author response** Thank you for the comment. We realised we used incorrect expressions to describe the wave height uncertainty earlier in this paragraph, which likely led the reviewer to inquire about the attenuation rates. In P9L278 of the original manuscript, we stated "waves decay with varying attenuation rates"; however, this is changed to "waves decay at different timing depending on the sea ice edge location of the respective SIC forcing used" (P8L244 in the revised manuscript). The whole point of the experiment is testing different SIC forcing using the consistent wave-ice interaction source term setting, i.e., the same attenuation coefficient. We apologise for the misleading sentence.

We are also mindful that a successful study of wave attenuation rates depends strictly on the knowledge of the ice edge or availability of two and more buoys along the fetch. As such, we are unable to repeat the novel analysis of Kohout et al. [2016] because we did not have enough measurements along the fetch.

**Reviewer comment R#1-24: P11L343.** I don't think that one can write that the MIZ is aligned with the wind. The MIZ is an area. Maybe the authors could substitute MIZ by "the ice edge".

**Author response** The sentence is corrected by replacing "MIZs" with "ice edges" (P11L320 in the revised manuscript).

**Reviewer comment R#1-25: P12L352.** "The figure only comprising...": I don't understand this sentence. What does "highly forced waves" mean?

**Author response** Highly forced waves imply a wave field, specifically wind seas, that is rapidly growing under the wind forcing. The sentence is rewritten to clarify the point. Please see the revised manuscript P11L329.

**Reviewer comment R#1-26: P12L362.** "this can occur": What is "this"?

**Author response** "this" is specified as "high $\Delta H_{m0}$". Please see the revised manuscript P11L338.

**Reviewer comment R#1-27: P12L365.** "Here, ..." I don't understand this sentence either, please consider rephrasing it.

**Author response** In this sentence, "highly forced" was unnecessary and is removed. The subject of the sentence is also clarified instead of using "Here". Please see the revised manuscript P11L341-343.

**Reviewer comment R#1-28: P12L371.** "The off-ice [...] of Appendix D." These two paragraphs are very confusing in my opinion, mostly because they are not well structured. It makes it very hard for the reader to understand the problem the authors are trying to address. They should be entirely rewritten.

**Author response** The repeat introduction of the equation in both paragraphs made them seem unstructured. We also made the second paragraph more succinct. The two paragraphs are modified. Please see the revised manuscript P11L346–P12L360.

**Reviewer comment R#1-29: P13L405.** "Three principal parameters that form the sea ice forcing are": This formulation is misleading. I would instead say: "The three main parameters used to tune the wave-in-ice attenuation in the IC2,IC3 and IC5 parameterizations are"

**Author response** The sentence is changed as suggested. Please see the revised manuscript P12L370.

**Reviewer comment R#1-30: P14L421.** "The values here [...] the adopted default source term parameters." I am not sure I understand this sentence, it should be rewritten.

**Author response** This sentence was unnecessary and is removed.

**Reviewer comment R#1-31: P14L422 .** "Our analysis demonstrates..." This statement should be at least discussed a bit more. For instance, the authors have used a limited number of wave-in-ice attenuation parameterizations, and none of them represent the wave scattering. Also, could these results change in a MIZ made of large floes and thicker ice for example? In addition, the authors have assumed a constant sea ice thickness of 10cm, and it is known that the behaviour of attenuation processes can change significantly depending on the sea ice thickness (see for instance Boutin et al., 2018). The sensitivity of these results to the sea ice thickness should be explored and discussed, for example by setting it to 20/30cm instead of 10cm.

**Author response** Agreed that further discussion is warranted. We tested robustness of the study finding by checking sensitivity to SIT and scattering. Please see the revised manuscript P12L383–P13L398.

**Reviewer comment R#1-32: P14L422 .** The conclusion is, in my opinion, much longer than it should be. I think it would have more impact if the results were synthesized in a few sentences only, and if it was ending with a discussion on the perspestives and the consequences of the findings presented here.

**Author response** The conclusion section has been made more succinct. Please see the revised manuscript P13L403–P14L420.

**Reviewer comment R#1-33: P14L443.** "Reliable modelling [...] melt the Arctic Ocean sea ice." I don't really see the cause/consequence link in this sentence.

**Author response** The sentence is removed as it was unnecessary. See author's changes in manuscript to R#1-32.

**Reviewer comment R#1-34: P16L490.** "Reliable shipboard [...] was slow." I find the formulations used in these sentences a bit ambiguous. For instance, what do the authors mean by "seemingly sensible"? I think I get the idea, but it is not very clearly expressed. It is also not clear to me what is validated in the first sentence.

**Author response** How these data were used in the study is described with more clarity. We have rewritten this passage in Appendix A explaining how the shipboard data were used. Please see the

revised manuscript P14L426–442.

**Reviewer comment R#1-35: P25 Figure 4.** "showing considerable uncertainty": This comment should be in the main text, not in the caption. The font size of the legend is also too small.

**Author response** The figure is removed as described in the overview of major changes.

**Reviewer #1 Technical corrections**

**Reviewer comment R#1-36: General.** I would recommend using a roman text font for the subscripts that are made of more than one character in the equations (low in f_low) for instance (If you are using latex, it means that you should add "\rm" or "\mathrmmy_subscript" in your equations). It would improve the readability of the paper.

**Author response** Subscripts are changed to roman text as recommended.

**Reviewer comment R#1-37: P1L21.** "encountering high winds...": encounters is already the verb of the sentence, so no need to repeat.

**Author response** Removed the verb phrase. Please see author's changes to manuscript in R#1-3.

**Reviewer comment R#1-38: P6L174.** "Sea thickness"→"Sea ice thickness"

**Author response** Corrected.

**Reviewer comment R#1-39: P14L440.** "have"→"has".

**Author response** Corrected.

**Reviewer comment R#1-40: P15L478.** ":microwave..."→":a microwave...".

**Author response** Corrected.

**Reviewer comment R#1-41: P15L480.** "a variant but similar device of Kohout et al..."→"a device similar to the one used by Kohout et al. (2015)".

**Author response** Changed as suggested.

**Reviewer #2 General comments**

**Reviewer comment R#2-1** The manuscript examines the impact of observational uncertainty in sea ice concentration on numerical modelling of ocean surface waves in sea ice, using Wavewatch III. The most interesting finding, in my opinion, is that the effect of this uncertainty on wave modelling is large, and larger than the differences between various wave-ice interaction source terms currently included in WW3.

In general, I find the results to be well supported by thorough analysis. The manuscript is a little hard to follow in places (some instances of this noted below in 'presentational comments'). As currently presented, my sense is that this work is of interest to a narrow section of the community (i.e. those working on spectral models of wave-ice interactions) and may be better suited to a more targeted journal. However, this could be addressed by revising the text to stress the aspects that could be interesting to a more general audience e.g. the differences between SIC products and their accuracy near the ice edge; that the sea ice coverage may be more important in determining ocean surface waves in ice than a detailed understanding of wave-ice interaction physics; the level to which waves in the open ocean are influenced by sea ice.

**Author response** We are pleased the significance of the SIC uncertainty on wave-ice models has been conveyed to the reviewer. Regarding the suggested approach to broaden the audience (*"e.g. the differences between SIC products and their accuracy near the ice edge; that the sea ice coverage may be more important in determining ocean surface waves in ice than a detailed understanding of wave-ice interaction physics; the level to which waves in the open ocean are influenced by sea ice"*), our analysis aims to convey that wave-ice model tuning may not be as effective at this time when the knowledge of the true SIC field is too uncertain; however, our work was not intended to suggest that SIC forcing accuracy takes precedence over the detailed understanding of wave-ice interaction physics. Our view is that advancing the knowledge of wave-ice interactions and its implementation in WW3 (or other wave model platforms) constitute the foundation to improving the predictability of ocean waves in ice-covered waters.

The research of ocean waves are becoming an important polar region research topic; waves feedback to the dynamics of atmosphere, ocean, and sea ice, and as such, we believe the manuscript contents are of interest to a general audience of The Cryosphere (TC). The recent acceptance of a coupled ice-ocean-wave model paper [Boutin et al., 2019] in TC reflects the increasing interest of the audience to the topic of ocean waves. We also cited the coupled wave-ice model references in the text to address R#2-10, which should help attract a wider audience.

**Reviewer #2 Major comments**

**Reviewer#2 comment R#2-2** 'Analysis of significant wave height uncertainty distributions for SIC forcing and wave-ice interaction source terms reveals that they are both sizeable; however, the study concludes the more dominant uncertainty source of modelling wave-ice interactions is the accuracy of satellite retrieved SIC estimates that are used as model forcing.' The authors need to clarify that uncertainty in wave-ice interaction terms is an estimate based only on the various wave-ice interaction terms included in WW3, which as the authors note later in the manuscript are based on a limited number of field observations. It does not sample all possible wave-ice interaction terms, nor does it sample different parameter values.

**Author response** Abstract has been rewritten as mentioned in the overview of major changes. Please see the revised manuscript P1L1–14.

**Reviewer comment R#2-3** It is well-known that SIC satellite observations underestimate concentrations for low thicknesses, less than 35 cm (Ivanova et al. 2015). This should be stated in the Introduction. I think the newer result here is how varied the satellite products are for these low thicknesses. This should be made clearer in the text.

**Author response** The long-known deficiency of satellite derived SIC accuracy in thin ice was indeed missing. We have incorporated the comment in Section 1 Introduction. Please see the revised manuscript P2L56.

**Reviewer comment R#2-4** Sec 2.3 This section needs more explanation to make it suitable for a more general audience - i.e. those not familiar with WW3 wave-ice interaction terms. What is the IS0 switch?

**Author response** We have modified Section 2.3 to be more readable to a broader audience. Please see the overview of major changes and the modified Section 2.3 starting at P5L142 of the revised manuscript.

**Reviewer comment R#2-5** What is the estimated of spatial domain of the sea-truth concentration observations? Are these comparable to the resolution of the satellite products?

**Author response** Here, we infer the comment pertains to Section 3.1. This section has been rewritten to address Reviewer#1's comments (refer to R#1-1, R#1-2, and R#1-19). Please see the modified Section 3 starting at P7L205 of the revised manuscript.

**Reviewer comment R#2-6** Sec 3.2 - why are only three products considered here? Conclusions on which products perform best (or worst) for this region would be useful and of broader interest to the cryospheric community.

**Author response** The three products were selected to convey the magnitude of uncertainty. Since the uncertainty was defined as $\Delta c_i = max(c_i) - min(c_i)$, three products are sufficient to convey this point. Showing all eight products clutters the figure and potentially distract readers from the main point. Note that this sub-section is moved to Section 5 (P10L284 of the revised manuscript).

In addition, the revised manuscript is presented such that the focus is apparent, which is the wave-ice models and SIC forcing and not about the SIC data. We anticipate this will be apparent when the revised manuscript is read. Nevertheless, regarding ranking the product performance, we agree such a conclusion would be a significant contribution. We have attempted to rank the SIC products, but the result is inconclusive primarily due to the number of days comprehensive mosaic can be produced and inconsistent variability of SIC estimates. We expect such an analysis warrants a separate dedicated study to produce noteworthy outcomes, especially considering the previous intercomparison studies, such as Ivanova et al. [2015] and Comiso et al. [2017], concluded there is no one product that is superior. They state that the choice of SIC products are largely dictated by the SIC data application, and we concur with this statement based on our field observation and analysis conducted.

**Reviewer #2 Minor comments**

**Reviewer comment R#2-7 :** **L22.** 'which are greater in the Arctic regions.' As compared to what?

**Author response** We made a change to this passage. Please see the revised manuscript P1L22–P2L27.

**Reviewer comment R#2-8 :** **L23.** 'sustainable developments of the Arctic Ocean' - this statement sounds strange.

**Author response** Please see author response to R#2-7.

**Reviewer comment R#2-9 :** **L28.** 'dispersion relation'→'the dispersion relation'.

**Author response** Corrected.

**Reviewer comment R#2-10:** **L32.** Other recent work on wave-ice interactions could be cited here: Zhang et al. 2019 (https://doi.org/10.1016/j.ocemod.2019.101532), Roach et al. 2019

(https://doi.org/10.1029/2019MS001836).

**Author response** This passage pertains specifically to the standalone WW3 wave-ice interaction source term developments, so the suggested references are unrelated. We added new sentences to address this comment at the end of the paragraph. This way, we can relate the study to a broader audience, which also helps to address R#2-1. Please see the revised manuscript P2L42–L45.

**Reviewer comment R#2-11 : L37.** 'Besides the model interior' What does this mean?

**Author response** We are referring to the inner part of numerical models. We added "wave-ice interaction parameterisations" as an example. Please see the revised manuscript P2L46.

**Reviewer comment R#2-12: L37.** 'like SIC'→'in products such as SIC'?

**Author response** Changed as suggested.

**Reviewer comment R#2-13: L69.** 'generally'→'generally between'.

**Author response** Corrected.

**Reviewer comment R#2-14: L70.** 'Albeit the MIZ coverage..' - this sentence is unclear.

**Author response** We have attempted to make it clearer. Please see the revised manuscript P3L84–86.

**Reviewer comment R#2-15: L82.** 'mostly same'→'mostly the same'.

**Author response** Corrected.

**Reviewer comment R#2-16 : L96.** remove 'exhaustive' (this is an opinion).

**Author response** "exhaustive" changed to "large".

**Reviewer comment R#2-17 : L102.** 'most frequent'→'most frequently'.

**Author response** Changed as suggested.

**Reviewer comment R#2-18 : L102.** 'leading algorithms' leading in what? Perhaps you mean the most commonly used?.

**Author response** We realised the phrase is unnecessary, so it is removed.

**Reviewer comment R#2-19 : L152.** 'sanity checked' inappropriate language for a publication.

**Author response** Changed to "tested". Please see the revised manuscript P6L179.

**Reviewer comment R#2-20 : L153.** 'leading packages' again, leading in what way? Most commonly used? Most skill?.

**Author response** Changed to "commonly used".

**Reviewer comment R#2-21 : L211.** change to 'SSTs exceeded 0oC'.

**Author response** Corrected. Please see the revised manuscript P16L495.

**R#2-22 : L211.** At what depth were the shipboard SSTs measured? Are they affected by the ship itself?.

**Author response** The technical details of the instruments were given in Appendix A, which was introduced in L76 of the original version: "$-1$ m below the sea surface with further 5 m inlet to the gauge" (L458). We are not aware of any ship effects on the SST measurement. During the cruise,

we note that sea ice, even grease ice, was observed only when shipboard SSTs were less than around $-1.5$ ℃. Accordingly, the gauge was sufficiently accurate for this discussion.

**Reviewer comment R#2-23 : L212.** No s after MIZ.

**Author response** "s" removed.

**Reviewer comment R#2-24 : L218.** 'forseeable'→'visible'.

**Author response** This section is modified and the sentence is removed. Please see author's response to R#2-5.

**Reviewer comment R#2-25 : L223.** 'The sea ice tends to cluster, so it appears dense where the ice exists.' - reword.

**Author response** Please see author's responses to R#2-5 R#2-24.

**Reviewer comment R#2-26 : L230.** 'depict the'→'show that the' ?

**Author response** Changed as suggested.

**Reviewer comment R#2-27 :  L264.** What does 'best practice wave models' mean? Perhaps replace with 'commonly used'?

**Author response** Changed to "high quality". Please see the revised manuscript P8L230.

**Reviewer comment R#2-28 : L266.** 'a bit over' - informal language.

**Author response** Changed to "slightly".

**Reviewer comment R#2-29 : L277.** 'as uncertainty generally increased in the MIZs.' - unclear. No s after MIZ.

**Author response** "in the MIZs" removed as it was unnecessary. Please see the revised manuscript P8L243.

**Reviewer comment R#2-30 : L299.** remove the rest of the sentence after 'ongoing work' - not necessary.

**Author response** Removed the sentence as it was unnecessary.

**Reviewer comment R#2-31 : L323.** add 'that' - 'demonstrated that'.

**Author response** Added "that".

**Reviewer comment R#2-32 : L338.** 'like a'→'as a'.

**Author response** Changed as suggested.

**Reviewer comment R#2-33 : L339.** 'effects. . .are' (not is).

**Author response** Corrected.

**Reviewer comment R#2-34 : L443.** Reword.

**Author response** The sentence is removed as it was unnecessary.

**Marked-up manuscript**

The marked-up manuscript with major changes indicated by the magenta text follows the references.

[revised manuscript text omitted]

---

## Referee Report (RR1)

The authors' revisions have greatly improved the quality of the manuscript. However, some elements of the manuscript still need to be clarified before publication. Most of them are very minors, I added a * in front of the ones that seem more important to me.

P1L1: *"Waves are known to decay exponentially..."*
It is often assumed that wave decay is exponential, but it is not found in all the observations of waves in ice (eg. Kohout et al., 2014). I would recommend to simply remove "exponentially" from the sentence.

P8L220 *"On the other hand"*
I am a bit confused by the use of "On the other hand" here. How does this sentence relate to the previous ones ? As I understand it, I would rather write something like "In existing parameterizations...".

P8L220 "Figure 2 shows" -> "Figure 2 also shows"

*P9L279 *"whereas the TodaiWW3-ArCS [...] Piper#13 location."*
I find this interesting, do the authors have an idea why this is the case?

P10L291 *"contours can be some 200km apart"*
It would be actually really nice to have some spatial scale in km on Figure 5 if possible.

P10L297 *"The above suggest [...] observations sites"*
I found the transition between section 4 and section 5 pretty confusing before reaching this paragraph, maybe consider putting it at the very beginning of section 5.

P13L386 *"For observational evidence [...] and selected 50cm."*
Using 50cm to study sensitivity of results to SIT makes sense to me, but I don't really understand this sentence. Ardhuin et al. (2018) do not use thickness observations, and investigate the sensitivity of their results to the constant thickness they provide by using SIT=15cm and SIT=30cm, not 50cm.

P13L396 Please mention that the results for the cases with scattering are not shown.

*P13L403 *"[...] models represent the exponential decay…"*
Here again I disagree with the fact that models in WW3 represent an exponential decay. This may be the case in the selection presented in this manuscript, but it is not the case in general. For instance, some of the processes introduced by Boutin et al. (2018) are non-linear.

P13L409 Please remind the reader  what is $s_{ice}$

*P13L415 *"It is worthy [...] to be resolved"*.
Swells that are long enough to propagate far in the ice cover (O(100km)) are likely to be unaffected by scattering. Scattering is only efficient for short waves (< 10s) when sea ice is made of consolidated floes.

---

## Author Response (AR2)

**Rebuttal 2 for tc-2019-285**

Takehiko Nose on behalf of all authors

We sincerely thank Editor David Schroeder for overseeing the review and both referees for the time spent on reviewing the revised manuscript. In this minor revision, main changes include the following:

- We revised sentences that assumed wave-ice decay is exponential.

- We clarified the "subgrid scale physics" missing in the wave-ice models to address several comments made by Referee #2.

- We elaborated the conclusion text so that general findings of the study can be understood as suggested by the editor and Referee #2.

- Some of the abstract was also rewritten to improve readability as suggested by Referee #2.

**Reviewer #1 Comments**

**Reviewer comment R#1-1: P1L1.** "Waves are known to decay exponentially..." It is often assumed that wave decay is exponential, but it is not found in all the observations of waves in ice (eg. Kohout et al., 2014). I would recommend to simply remove "exponentially" from the sentence.

**Author response** "Exponentially" removed as suggested in P1L1.

**Reviewer comment R#1-2: P8L220.** "On the other hand" I am a bit confused by the use of "On the other hand" here. How does this sentence relate to the previous ones ? As I understand it, I would rather write something like "In existing parameterizations...".

**Author response** Changed as suggested. Please see P8L227.

**Reviewer comment R#1-3: P8L220.** "Figure 2 shows" -> "Figure 2 also shows".

**Author response** Changed to "Figure 2 snapshot images are accompanied with", which fits better with the revised text addressing Referee #2's comments to clarify model subgrid scale physics. Please see P8L228.

**Reviewer comment R#1-4: P9L279.** "whereas the TodaiWW3-ArCS [...] Piper#13 location." I find this interesting, do the authors have an idea why this is the case?

**Author response** Thank you for pointing this out. Based on the close inspection of this event, we see that the fetch distance is changed depending on how models treat sea ice: ice mask (ERA5 & ARCMFC) and satellite SIC (TodaiWW3). We added a passage to this effect in P10L287–291.

**Reviewer comment R#1-5: P10L291.** "contours can be some 200km apart" It would be actually really nice to have some spatial scale in km on Figure 5 if possible.

**Author response** Added indicative spatial reference in km. Please see new Figure 5.

**Reviewer comment R#1-6: P10L297.** "The above suggest [...] observations sites" I found the transition between section 4 and section 5 pretty confusing before reaching this paragraph, maybe consider putting it at the very beginning of section 5.

**Author response** The paragraph was moved to the end of Section 4 to make the section transition more fluent as suggested. Please see P10L295.

**Reviewer comment R#1-7: P13L386.** *"For observational evidence [...] and selected 50cm."* Using 50cm to study sensitivity of results to SIT makes sense to me, but I don't really understand this sentence. Ardhuin et al. (2018) do not use thickness observations, and investigate the sensitivity of their results to the constant thickness they provide by using SIT=15cm and SIT=30cm, not 50cm.

**Author response** Thank you for pointing this out. Actually, we initially tested 30 cm thickness as guided by Ardhuin et al. [2018]. We then tested 50 cm to consolidate our sensitivity test results. Reporting on the 50 cm result (without the 30 cm result) seems sufficient to address the thickness sensitivity, and it also makes the text more succinct. The sentences with inaccuracies pointed out are removed. Please see P13L401.

**Reviewer comment R#1-8: P13L396.** Please mention that the results for the cases with scattering are not shown.

**Author response** Added the suggested in P13L410.

**Reviewer comment R#1-9: P13L403.** *"[...] models represent the exponential decay..."* Here again I disagree with the fact that models in WW3 represent an exponential decay. This may be the case in the selection presented in this manuscript, but it is not the case in general. For instance, some of the processes introduced by Boutin et al. (2018) are non-linear.

**Author response** Thank you. We incorporated this comment. Please see P14L418–421.

**Reviewer comment R#1-10: P13L409.** Please remind the reader what is $s_{ice}$.

**Author response** "Wave-ice interaction term" added to the sentence as well as including more explanations the experiment. Please see P14L426–428.

**Reviewer comment R#1-11: P13L415.** *"It is worthy [...] to be resolved".* Swells that are long enough to propagate far in the ice cover (O(100km)) are likely to be unaffected by scattering. Scattering is only efficient for short waves ($< 10s$) when sea ice is made of consolidated floes.

**Author response** We simplified the sentence here to say that the significance of this study outcome for scattering dominated wave conditions remains to be resolved in P14L434.

We sincerely thank Referee #1 for providing invaluable insights that have improved this manuscript.

**Reviewer #2 General comments**

**Reviewer comment R#2-1: L1.** 'Ocean waves are known to decay exponentially when they interact with sea ice.' My understanding from the literature is that there is some uncertainty around this. Suggest softening the sentence.

**Author response** "Exponentially" removed as suggested. Please see P1L1.

**Reviewer#2 comment R#2-2: L4.** 'this implies that the subgrid scale physics is missing' - how? I don't understand this sentence.

**Author response** There were several comments regarding "subgrid scale physics". The subgrid scale processes relate to the SIC and sea ice types $s_{\mathrm{ice}}$ heterogeneity. As an example to account for SIC heterogeneity, we may write the subgrid scale SIC as $c_{i,\mathrm{subgrid}} = \langle c_{i,\mathrm{subgrid}} \rangle + c_{i,\mathrm{subgrid}}'$ where $\langle c_{i,\mathrm{subgrid}} \rangle$ is the grid scale. If we let the attenuation rate $\alpha$ be a function of SIC, the subgrid scale attenuation may be expressed as $\alpha(c_{i,\mathrm{subgrid}}) = \alpha(\langle c_{i,\mathrm{subgrid}} \rangle + c_{i,\mathrm{subgrid}}')$; however, $\alpha(\langle c_{i,\mathrm{subgrid}} \rangle + c_{i,\mathrm{subgrid}}') \neq \alpha(\langle c_{i,\mathrm{subgrid}} \rangle) + \alpha(c_{i,\mathrm{subgrid}}')$. If the attenuation rate is nonlinear, then $\langle \alpha(c_{i,\mathrm{subgrid}}') \rangle \neq 0$ so $\langle \alpha(c_{i,\mathrm{subgrid}}) \rangle \neq \alpha(\langle c_{i,\mathrm{subgrid}} \rangle)$. The same logic may also apply to the sea ice types. Instead of saying "it implies", we changed this phrase to "we show that" in P1L4.

**Reviewer comment R#2-3: L12.** 'model ice types, i.e., wave-ice interaction parameterisations. Despite these parameterisations being derived from different concepts and missing the subgrid scale physics relating to sea ice field heterogeneity.' This is not clearly written. Please edit.

**Author response** The sentence is modified as suggested in P1L14.

**Reviewer comment R#2-4: L24.** 'describe that' -> 'state that'

**Author response** Changed as suggested in P2L25.

**Reviewer comment R#2-5: L29-31.** There is a bit of repetition here.

**Author response** They are not a repetition as we first provide the WMO definition of an MIZ, which is followed by the text to specify the region of the MIZ that is of primary interest in this study.

**Reviewer comment R#2-6: L40.** 'qualify' >'caution that'.

**Author response** Changed as suggested in P2L41.

**Reviewer comment R#2-7: L46.** 'Besides the model interior, e.g., wave-ice interaction parameterisations, the 0th order uncertainty pertains to sea ice forcing accuracy such as SIC and sea ice thickness (SIT).' -> Suggest rewording as follows 'Uncertainty in modelling waves in ice arises both from parametrisation of wave-ice interactions and from uncertainty in sea ice variables used as forcing, specifically SIC and sea ice thickness (SIT).' or similar. 'Model interior' sounds a little strange to me.

**Author response** Changed as suggested in P2L47.

**Reviewer comment R#2-8: L65.** 'observation' -> 'observations'

**Author response** Changed. Please see P3L66.

**Reviewer comment R#2-9: L67.** 'which led to the subject of satellite retrieved SIC.' - unnecessary, suggest removing.

**Author response** Removed as suggested.

**Reviewer comment R#2-10: L85.** 'was less expansive' - than what?

**Author response** Removed the first clause of the sentence.

**Reviewer comment R#2-11: L116.** 'is' -> 'denote'.

**Author response** Changed as suggested in P4L116.

**Reviewer comment R#2-12: L117.** 'Four algorithms that appear most frequently in literature were considered, and the following algorithms were selected' -> 'Four algorithms that appear most frequently in literature were selected'

**Author response** Corrected as suggested in P4L117.

**Reviewer comment R#2-13: L180.** Briefly note what 'ST4' and 'ST6' are.

**Author response** This paragraph is about the WW3 $s_{\mathrm{wind}}$ and $s_{\mathrm{dissipation}}$ parameterisations: ST4 and ST6 are the most commonly used parameterisations. We edited this paragraph to make sure this is clear. Please see P6L179.

**Reviewer comment R#2-14: L204.** 'and the use in wave-ice models' -> 'and use in wave-ice models'.

**Author response** Corrected as suggested in P7L205.

**Reviewer comment R#2-15: L207.** 'the wave' -> 'waves'?

**Author response** Changed to "waves that provide the scale to SIC". Please see P7L208.

**Reviewer comment R#2-16: L218.** 'the subgrid scale physics is completely missing' -> 'the sub grid scale distribution of sea ice and ice types are missing' ?

**Author response** We clarified model subgrid scale physics. Text similar to R#2-2 is added to address this comment. Please see P8L219–225.

**Reviewer comment R#2-17: L218.** The two sentences beginning 'It is plausible' I don't really understand. Please clarify.

**Author response** The additional text addressing R#2-16 should make this clear. Please see P8L225–227.

**Reviewer comment R#2-18: L222.** Sentence beginning 'Moreover' doesn't quite make sense. Please check.

**Author response** "of" was not the most appropriate preposition, which is changed to "during". Please see P8L230.

**Reviewer comment R#2-19: L233.** 'The ARCMFC wave model has a horizontal resolution of around 8 km and also used an ice mask; from December 2019, the month after our observation, the model was upgraded to simulate waves under sea ice cover based on Sutherland 235 et al. (2019)'s two-layer sea ice model.' So which version did you use? If the latter, no need to mention the previous model version.

**Author response** Please accept our apology as there was a typo here. It should be 2018. The data we obtained was before the 2-layer wave-ice model implementation as the observation was before their model upgrade. The sentence is amended as shown in P8L240.

**Reviewer comment R#2-20: L243.** 'The time series figure depicts the effect of sea ice on waves each time R/V Mirai sailed in the ice cover as the uncertainty generally increased.' > 'The figure shows that each time R/V Mirai sailed into ice cover, uncertainty in wave height generally increased.'

**Author response** Changed as suggested in P9L250.

**Reviewer comment R#2-21: L250.** I don't quite follow this sentence.

**Author response** We rephrased the last clause to clarify the point better. Please see P9L257.

**Reviewer comment R#2-22: L264.** 'having open water to 0.30 SIC' - it is a little unclear what this means.

**Author response** We added "as it uses the ice mask" in P9L271.

**Reviewer comment R#2-23: L269.** 'at the finals hours on 6 Nov' - final hour? 23:00?

**Author response** Removed "at the finals hours".

**Reviewer comment R#2-24: L271.** 'indicate instrument noise' -> 'indicate instrument noise only'?

**Author response** Removed "instrument".

**Reviewer comment R#2-25: L275.** 'mean(Hm0)' - I think this is the mean across the six SIC forcing simulations, but it is not defined. Similarly for mean(c).

**Author response** Added (i.e., mean of the TodaiWW3-ArCS simulations). Please see P9L282. Also see P11L329 for mean(c).

**Reviewer comment R#2-26: L285.** Perhaps, 'NRCSs provide an alternative indication of true sea ice fields'? The passive microwave radiometer products do also provide indication of true sea ice fields, although they have uncertainties.

**Author response** Added "at a higher resolution" in P10L301.

**Reviewer comment R#2-27: L287.** 'depicts sea ice edges have a wavy' > 'depicts sea ice edges that have a wavy'

**Author response** Changed in P10L302.

**Reviewer comment R#2-28: L292.** 'appears' -> 'appear'.

**Author response** Corrected as suggested P10L308.

**Reviewer comment R#2-29: L293.** 'the waters in this area were not high SIC' -> 'SIC was not high in this area'.

**Author response** Changed as recommended in P10L308.

**Reviewer comment R#2-30: L310.** 'fetch on' -> 'fetch over'.

**Author response** Changed in P11L323.

**Reviewer comment R#2-31: L312.** 'non-homogeneous nature of wind that generates waves and the SIC field heterogeneity' -> 'the heterogeneity of both the nature of wind that generates waves and the SIC field'

**Author response** Changed as suggested in P11L325.

**Reviewer comment R#2-32: L348.** The two sentence beginning on this line are a little unclear. I think they would benefit from some further explanation.

**Author response** We simplified this sentence using observational evidence. Please see P12L361–364.

**Reviewer comment R#2-33: L360.** Fig. 9b merits a sentence or two description here.

**Author response** Added a short description as suggested in P12L375.

**Reviewer comment R#2-34: L365.** 'observation of' -> 'observed'.

**Author response** Corrected as suggested in P13L380.

**Reviewer comment R#2-35: L368.** 'experiment was conducted' -> 'experiments were conducted'.

**Author response** We only carried out one $s_{ice}$ uncertainty experiment.

**Reviewer comment R#2-36: L390.** 'sensitivity on' -> 'sensitivity to'.

**Author response** Corrected in P13L405.

**Reviewer comment R#2-37: L397.** 'The results remained robust' - which results exactly?.

**Author response** The result is described clearly as suggested in P13L411.

**Reviewer comment R#2-38: L398.** 'same outcome' - suggest being more specific here.

**Author response** The outcome is described clearly as suggested in P13L413.

**Reviewer comment R#2-39: L403.** 'models represent' -> 'model represents'.

**Author response** Changed as suggested P14L418.

**Reviewer comment R#2-40: L409.** 'the sice uncertainty experiment' - this experiment should be briefly described here, so that a reader can understand the Conclusions without reading the text.

**Author response** This is true, and we elaborated the conclusion text. Please see P14L426–433.

**Reviewer comment R#2-41: L413.** completely missing the subgrid scale physics relating to sea ice field heterogeneity' - suggest explaining what exactly is missing here.

**Author response** Addressed similarly to R#2-3. Please see P14L431.

**Reviewer comment R#2-42: L418.** Perhaps use 'approaches' rather than 'ends'.

**Author response** Changed as suggested in P14L437.

**Reviewer comment R#2-43: L471.** 'These estimates are noise because SSTs were too warm for new and young ice to form.' Is this from SSTs captured by the ship? The ship-measured sea temperatures are taken 1m below the surface, and could be warmer than the the true surface temperature.

**Author response** Removed the SST comment and simply noted that the R/V Mirai crew and ice navigator did not log any sea ice. P16L490.

**Reviewer comment R#2-44: L491.** 'We've' -> 'We have'.

**Author response** Thank you for pointing this out. Corrected in P17L508.

**Reviewer comment R#2-45: Fig.2.** should add a legend for the line plot for colour-blind readers.

**Author response** Ideally we prefer not to compromise the figure legibility. Since some appendix SIC figures use the same figure schematics, we cross referenced these figures for those who require viewing the legend.

**Reviewer comment R#2-46: Fig3-4.** remove 'Model' from the y-axis label, as these figures also include observations.

**Author response** Removed as suggested.

We sincerely thank Referee #2 for reviewing this manuscript in such great detail, which improved this manuscript.

**Marked-up manuscript**

The marked-up manuscript with changes indicated by the magenta text follows the references.

**References**

Fabrice Ardhuin, Guillaume Boutin, Justin Stopa, Fanny Girard-Ardhuin, Christian Melsheimer, Jim Thomson, Alison Kohout, Martin Doble, and Peter Wadhams. 
[revised manuscript text omitted]